# Base editing screens define the genetic landscape of cancer drug resistance mechanisms

Matthew A. Coelho [1,2,3] ✉, Magdalena E. Strauss [4,5,6,13], Alex Watterson[1], Sarah Cooper[6], Shriram Bhosle [1], Giuditta Illuzzi[7], Emre Karakoc[1,3], Cansu Dinçer[1], Sara F. Vieira [1,3], Mamta Sharma[1], Marie Moullet[1], Daniela Conticelli [8,9], Jonas Koeppel [10], Katrina McCarten[1], Chiara M. Cattaneo [11,12,14], Vivien Veninga[11,12,15], Gabriele Picco[1,3], Leopold Parts[10], Josep V. Forment[7], Emile E. Voest [11,12], John C. Marioni [4,5,16], Andrew Bassett [6] & Mathew J. Garnett [1,3] ✉

Drug resistance is a principal limitation to the long-term efficacy of cancer therapies. Cancer genome sequencing can retrospectively delineate the genetic basis of drug resistance, but this requires large numbers of post-treatment samples to nominate causal variants. Here we prospectively identify genetic mechanisms of resistance to ten oncology drugs from CRISPR base editing mutagenesis screens in four cancer cell lines using a guide RNA library predicted to install 32,476 variants in 11 cancer genes. We identify four functional classes of protein variants modulating drug sensitivity and use single-cell transcriptomics to reveal how these variants operate through distinct mechanisms, including eliciting a drug-addicted cell state. We identify variants that can be targeted with alternative inhibitors to overcome resistance and functionally validate an epidermal growth factor receptor (EGFR) variant that sensitizes lung cancer cells to EGFR inhibitors. Our variant-to-function map has implications for patient stratification, therapy combinations and drug scheduling in cancer treatment.

Resistance to molecularly targeted anti-cancer treatments remains a major clinical challenge[1]. Drug resistance is frequently caused by DNA single nucleotide variants (SNVs) in the cancer genome[2], leading to point mutations in the drug target or proteins within the same signaling pathway[3]. The study of drug resistance can inform drug mechanism of action, the design of second-generation inhibitors targeting drug-resistant proteins, the development of combination therapies and patient stratification for second-line therapies. Current approaches often depend on sequencing tumor biopsies from patients that relapse on treatment. These are challenging samples to acquire, meaning it can take years to accrue enough to infer variant function. These analyses are generally restricted to frequently observed variants and must

be individually experimentally validated to establish a causal link to drug resistance. This is a slow process that does not allow for the direct comparison of different variant effects. These challenges limit the interpretation of cancer biopsy and circulating tumor DNA sequencing data. Rapid, prospective and systematic functional annotation of variants would accelerate the discovery of drug resistance mechanisms.

CRISPR-based gene editing approaches such as base editing can be used to directly interpret the function of variants of unknown significance (VUS)[4–12] and study genetic mechanisms of resistance to cytokines and inhibitors[8,13,14]. Cytidine and adenine base editors use a Cas9 nickase fused to a deaminase, facilitating the programmed installation of C>T and A>G SNVs in the genome at high efficiency in

**Fig. 1 | Base editing screens map functional domains in oncogenes.**
**a**, Overview of base editing screens to identify drug resistance variants in cancer cell models. **b**, Base editor screens in HT-29 cells across 11 cancer genes show depletion of gRNAs targeting essential genes demonstrating base editing activity. Unpaired, two-tailed Student's *t*-test comparing NT gRNAs ($n = 114$) with gRNAs targeting essential gene splice sites ($n = 632$) in CBE and ABE screens. Boxplots represent the median, interquartile range (IQR) and whiskers are the lowest and highest values within 1.5 × IQR. **c**, Comparison of gRNA *z*-scores from base editor screens in PC9 (*EGFR*-mutant, *MYC*-dependent) and HT-29 (*BRAF*-mutant, *MYC*-dependent) reveals shared and disparate oncogene dependencies. **d**, Base editing mutagenesis screens of the driving oncogene, *BRAF*, in HT-29 cells reveal functional protein domains, sites of post-translational modification and driver variants. Data are the average of two independent experiments. See also Extended Data Fig. 1. Schematic in **a** created with BioRender.com.

physiologically relevant cell types[9–12,15–18]. Here we use base editing at scale to investigate genetic mechanisms of acquired resistance to molecularly targeted cancer therapies, identifying VUS conferring drug resistance and drug sensitization in cancer cells. We classify cancer variants modulating drug sensitivity into four functional classes, thus providing a systematic framework for interpreting drug resistance mechanisms.

## Results

### Base editing screens map functional domains in cancer genes

We investigated drug resistance to ten molecularly targeted cancer drugs that are currently approved by the United States Food and Drug Administration (FDA) or are under clinical investigation (Fig. 1a). We selected four cancer cell lines that are sensitive to these agents[19] and harbor diverse oncogenic drivers[20]: H23 (lung; KRAS-G12C), PC9 (lung; EGFR amplification and exon19 deletion), HT-29 (colon; BRAF V600E) and MHH-ES-1 (Ewing sarcoma; EWS-FLI1 fusion). We mutagenized 11 cancer genes that encode common drug targets or genes within the same signaling pathway for interrogation with a guide RNA (gRNA) library ($n = 22,816$), tiling these genes and their 5′ and 3′ untranslated regions (UTRs). As controls, we included nontargeting gRNAs (NT gRNAs) ($n = 57$), intergenic-targeting gRNAs ($n = 168$) and gRNAs predicted to introduce splice variants[21] in nonessential ($n = 87$) and essential ($n = 316$) genes[22,23] (Supplementary Table 1). To maximize the saturation of targeted mutagenesis, the gRNA library was introduced into cancer cell lines expressing doxycycline-inducible cytidine base editor (CBE) or adenine base editor (ABE)[8] with relaxed PAM requirements (Cas9–NGN)[24]. We analyzed the potential functional effects of thousands of gene variants on drug resistance in parallel by performing base editing screens with a proliferation read-out in the presence of targeted anti-cancer drugs from 46 independent pooled genetic screens (Fig. 1a and Supplementary Table 2). Base editing screen replicates were highly correlated (Extended Data Fig. 1), and control gRNAs targeting essential genes were depleted, indicating efficient base editing (Fig. 1b and Extended Data Fig. 2a). gRNAs with a high off-target score were excluded from further analysis (1.2% of gRNAs; Extended Data Fig. 2b; Methods).

Cancer cell models had enhanced sensitivity to gRNAs targeting their specific driver oncogenes. PC9 cells were most sensitive to deleterious edits in *EGFR*, whereas HT-29 were most sensitive to deleterious

edits in *BRAF*, and both were sensitive to targeting *MYC*, consistent with reported oncogene addiction[25] (Fig. 1c). Specifically, base editing mutagenesis across *BRAF* in HT-29 cells revealed enrichment of depleted gRNAs predicted to install missense variants in crucial functional domains, such as the RAS-binding domain and protein kinase domain (10.2% and 59.2% of deleterious BRAF missense variants with *z*-scores < −2, respectively) (Fig. 1d). We identified important sites of post-translational modification, such as the phosphorylation sites[26] S446, S365 and T753, as well as a predicted gain-of-function missense variant at residue L505 (Fig. 1d), which has been reported to co-occur with BRAF V600E, cause resistance to the BRAF inhibitor vemurafenib and increase MAPK signaling[27]. These data highlight that base editing can provide insights into protein structure–function relationships, including critical domains and residues, which could be valuable in drug discovery campaigns.

## Four classes of variants modulating drug sensitivity

The integration of 46 mutagenesis screens led to the identification of four functional classes of variants modulating drug sensitivity. We classified these as: (1) 'drug addiction variants' that confer a proliferation advantage in the presence of drug but are deleterious in the absence of drug; (2) 'canonical drug resistance variants' conferring a proliferation advantage only in the presence of drug; (3) 'driver variants' conferring a proliferation advantage in the presence and absence of drug; and (4) 'drug-sensitizing variants', which are deleterious only in the presence of drug.

As an example, we observed all four classes of variants modulating drug sensitivity to the allosteric MEK1/2 inhibitor trametinib[28,29] in HT-29 cells (Fig. 2a); drug addiction variants (*n* = 10 gRNAs), canonical drug resistance variants (*n* = 30 gRNAs), driver variants (*n* = 24 gRNAs) and drug-sensitizing variants (*n* = 111 gRNAs). Of the 175 hit-scoring gRNAs from the trametinib screens, 0 were control gRNAs, implying a high signal-to-noise ratio (summarized in Supplementary Note 1 for all screens). We identified trametinib sensitizing variants as causing loss-of-function in EGFR (Fig. 2a, z-score < −2; Methods), indicating that combination therapies targeting RAF-MEK and EGFR are effective in BRAF-mutant colorectal cancer (CRC), and consistent with the clinical approval EGFR and BRAF inhibitors in CRC[30–32]. We confirmed this genetic interaction in HT-29 cells in genome-wide CRISPR–Cas9 knockout screens in the presence of the BRAF inhibitor, dabrafenib, where knockout of EGFR was the strongest sensitizing hit[33] (Extended Data Fig. 2c).

We subsequently investigated resistance to the combination of BRAF and EGFR inhibitors by performing a base editing screen in the presence of dabrafenib and cetuximab (Fig. 2b). As expected, we detected three main classes of drug resistance variants, with drug-sensitizing variants now largely absent for the combination therapy (8 with the combination therapy versus 111 with trametinib). Collectively, these data demonstrate that base editing screens can reveal functionally distinct variants modulating drug sensitivity and highlight effective drug combinations.

## Drug addiction and canonical drug resistance variants

We identified drug addiction and canonical drug resistance variants for several drugs. Canonical drug resistance variants such as MEK1 L115P[34] (Fig. 2a) and EGFR S464L[35] (Fig. 2b) were within the drug-binding pockets for trametinib and cetuximab, respectively, consistent with disruption of drug binding (Fig. 2c). In contrast, drug addiction variants were predominantly activating mutations in oncogenes within the MAPK signaling pathway (for example KRAS Q61R/E62G, MEK2 Y134H)[36]. The deleterious effects of these variants in the absence of inhibition with trametinib is consistent with overactivation of MAPK signaling leading to oncogene-induced senescence[37,38]. This phenotype is concordant with the mutual exclusivity of activating mutations within KRAS and BRAF in patient CRC samples (*P* < 0.001; Extended Data Fig. 2d), highlighting that the drug addiction phenotype is dependent

on pre-existing mutations in the cancer cell. Notably, MEK1 Q56R/ K57E/R and F53L/S drug addiction variants confer resistance to cetuximab in CRC patients[39–41] (Fig. 2a,b).

We validated the effect of BRAF and EGFR combination and MEK1/2 inhibitor drug resistance variants using arrayed proliferation assays and by analyzing cell signaling. Overall, we set out to validate 4 of 13 drug addiction and 3 of 36 canonical drug resistance gRNAs for dabrafenib and cetuximab, and 4 of 10 drug addiction and 2 of 30 canonical drug resistance gRNAs for trametinib. Consistent with our high-throughput screening data, all of the variants analyzed variants led to robust drug resistance (Fig. 2d). Canonical drug resistance variants did not have adverse effects on cell growth in the absence of drug; however, the drug addiction variants grew more slowly than controls in the absence of drug treatment. HT-29 cells with drug addiction variants had elevated basal MAPK signaling and elevated p21 protein expression, which were reduced to near WT levels by dabrafenib and cetuximab (Fig. 2e and Extended Data Fig. 3a). HT-29 cells harboring drug addiction variants showed an altered cell morphology and increased β-galactosidase staining in the absence of drug, indicating increased senescence, which was also reversed with drug treatment (Fig. 2f and Extended Data Fig. 3b). These data support the hypothesis that cancer cell clones harboring drug addiction variants may be eliminated by implementing intermittent drug scheduling—so-called drug holidays[42–44].

## Driver variants conferring drug resistance

Driver variants in orthogonal signaling pathways or in kinases downstream of the drug target can give rise to drug resistance in cancer. We observed rare gain-of-function variants arising from gRNAs predicted to install missense variants in *AKT1* and *PIK3CA* (Fig. 3a). These sites resided in known hotspots; AKT1 E17K and PIK3CA E545K/ E542K[45]. We also detected rarer driver variants at residues AKT1 D323 and PIK3CA C378 and E365 (ref. 45), although PIK3CA H1047 driver variants were not detected due to the editing and saturation constraints of the CBE and ABE NGN base editors.

Variants conferring resistance to pictilisib, a pan-PI3K inhibitor under clinical development for solid tumors[46], were rare in HT-29 base editing screens. Only driver variants in the downstream kinase, AKT1, conferred effective drug resistance (Fig. 3b). In contrast, driver variants in the drug target itself, PIK3CA, conferred a proliferation advantage only in the absence of drug, suggesting that PIK3CA activating variants remain sensitive to pictilisib.

We also compared the genetic mechanisms of acquired resistance with the recently FDA-approved KRAS-G12C inhibitors sotorasib[47] and adagrasib[48] in H23 KRAS^G12C lung cancer cells. KRAS variants proximal to the drug binding pocket (for example, R68G and D69G), gave cross-resistance to sotorasib and adagrasib, consistent with clinical findings[49]. Other known resistance variants, including alternative G12 alleles[49], were not detected in our analysis, highlighting that base editing screens are not saturating and cannot install all variants. Our pathway-wide approach allowed us to detect activating variants in downstream kinases that confer resistance to sotorasib and adagrasib (Fig. 3c). Many of these activating variants in downstream kinases MEK1 (gene *MAP2K1*—Y130H, F129S) and MEK2 (gene *MAP2K2* Y134H and Q60R, K61R/E) conferred resistance to both inhibitors.

## Drug resistance and drug-sensitizing variants in PARP1

Mechanisms of resistance to PARP inhibitors in cancer include ontarget drug-resistance variants[50] and reactivation of DNA damage repair pathways, such as the reversion of *BRCA2* mutations[51,52]. We investigated genetic resistance mechanisms to olaparib[53] and niraparib[54] in the Ewing sarcoma cell line MHH-ES-1, which harbors an *EWSR1* fusion conferring sensitivity to PARP inhibitors[55] (Fig. 4a). Drug resistance was predominantly mediated by ontarget mutations in PARP1 rather than PARP2 or other target genes (Extended Data Fig. 4a), in line with data showing that inhibition of PARP1 is the main driver of PARP inhibitor efficacy[50,56,57].

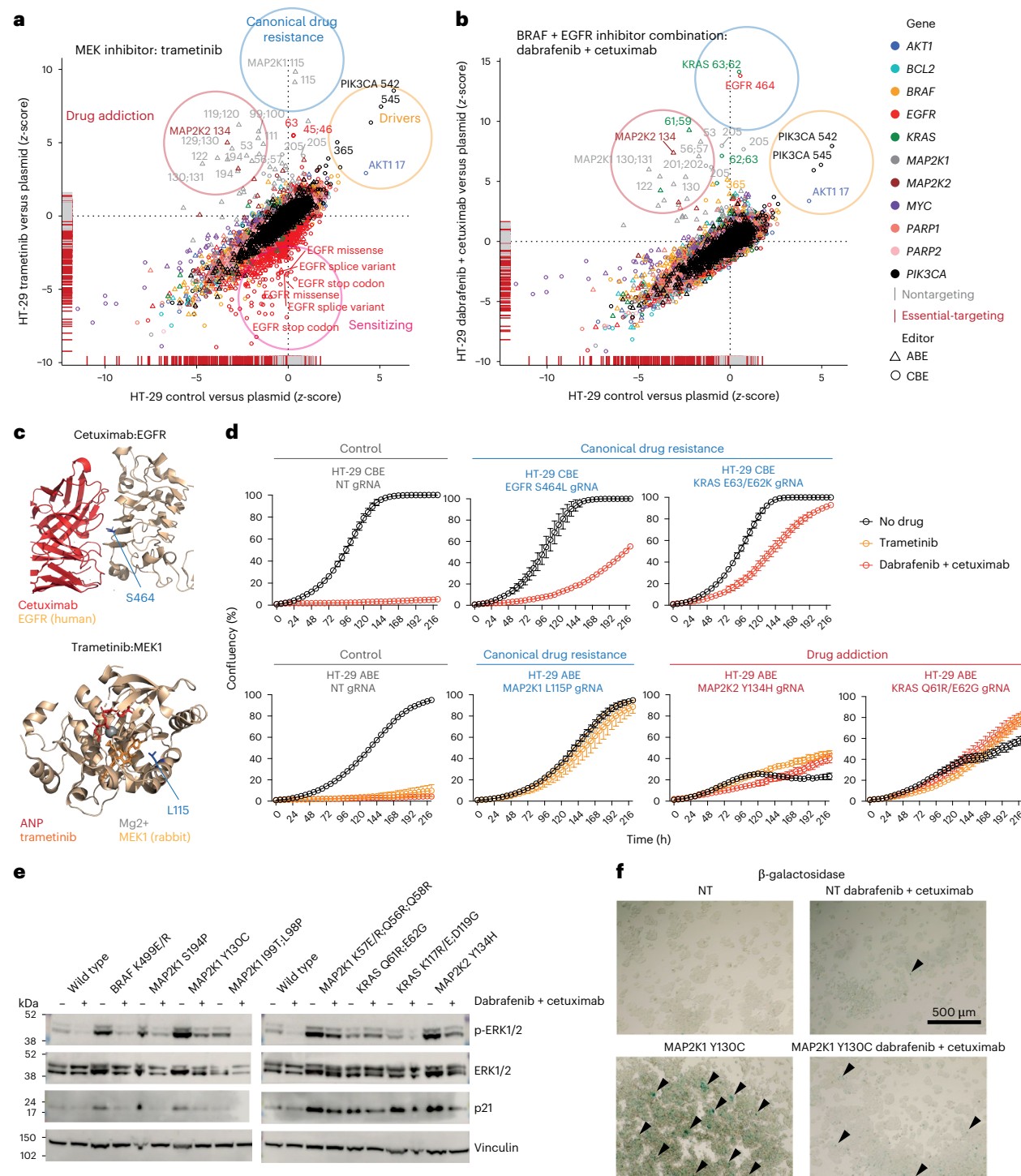

**Fig. 2 | Variants modulating drug sensitivity cluster into four functional classes. a**, Variants conferring resistance or sensitivity to the MEK inhibitor, trametinib, in HT-29 cells. Comparison of gRNA z-scores for the control treated arm versus plasmid library, and the drug-treated arm versus plasmid library is shown. **b**, Variants conferring resistance to the combination of BRAF and EGFR inhibitors, dabrafenib and cetuximab, in HT-29 cells. Comparison of gRNA z-scores for the control treated arm versus plasmid library, and the drug-treated arm versus plasmid library is shown. **c**, Crystal structure of the complex of EGFR and cetuximab (PDB 1yy9)[82], and MEK1 and trametinib (PDB 7jur)[83], highlights canonical drug resistance variants discovered in base editor screens predicted to disrupt drug binding. **d**, Cell growth of base-edited HT-29 cells harboring canonical and drug-addiction drug-resistance variants. Cells were left untreated or treated with trametinib (3 nM) or the combination of dabrafenib (80 nM) and cetuximab (1 μg ml⁻¹), and cell proliferation was

monitored using an incucyte. Data represent the mean ± s.d. of biological triplicates and are representative of two independent experiments. **e**, Western blotting of WT HT-29 ABE cells and cells harboring drug-resistance mutations activating the MAPK signaling pathway. Cells were treated with the combination of dabrafenib (80 nM) and cetuximab (1 μg ml⁻¹) or DMSO as a control for 24 h before analysis. **f**, β-galactosidase staining for senescent cells; β-galactosidase positive senescent foci (blue) are indicated with arrows. HT-29 cells were treated with the combination of dabrafenib (80 nM) and cetuximab (1 μg ml⁻¹) or DMSO as a control for 48 h before analysis. Representative images are shown for the drug addiction variant MAP2K1 Y130C. Scale bar, 500 μm. Predicted amino acid editing consequences are labeled for drug resistance screens and genotyped edits are shown in **d**, **e** and **f**. Data are the average of two independent experiments performed on separate days, or representative of two independent experiments (**e** and **f**). See also Extended Data Figs. 2 and 3.

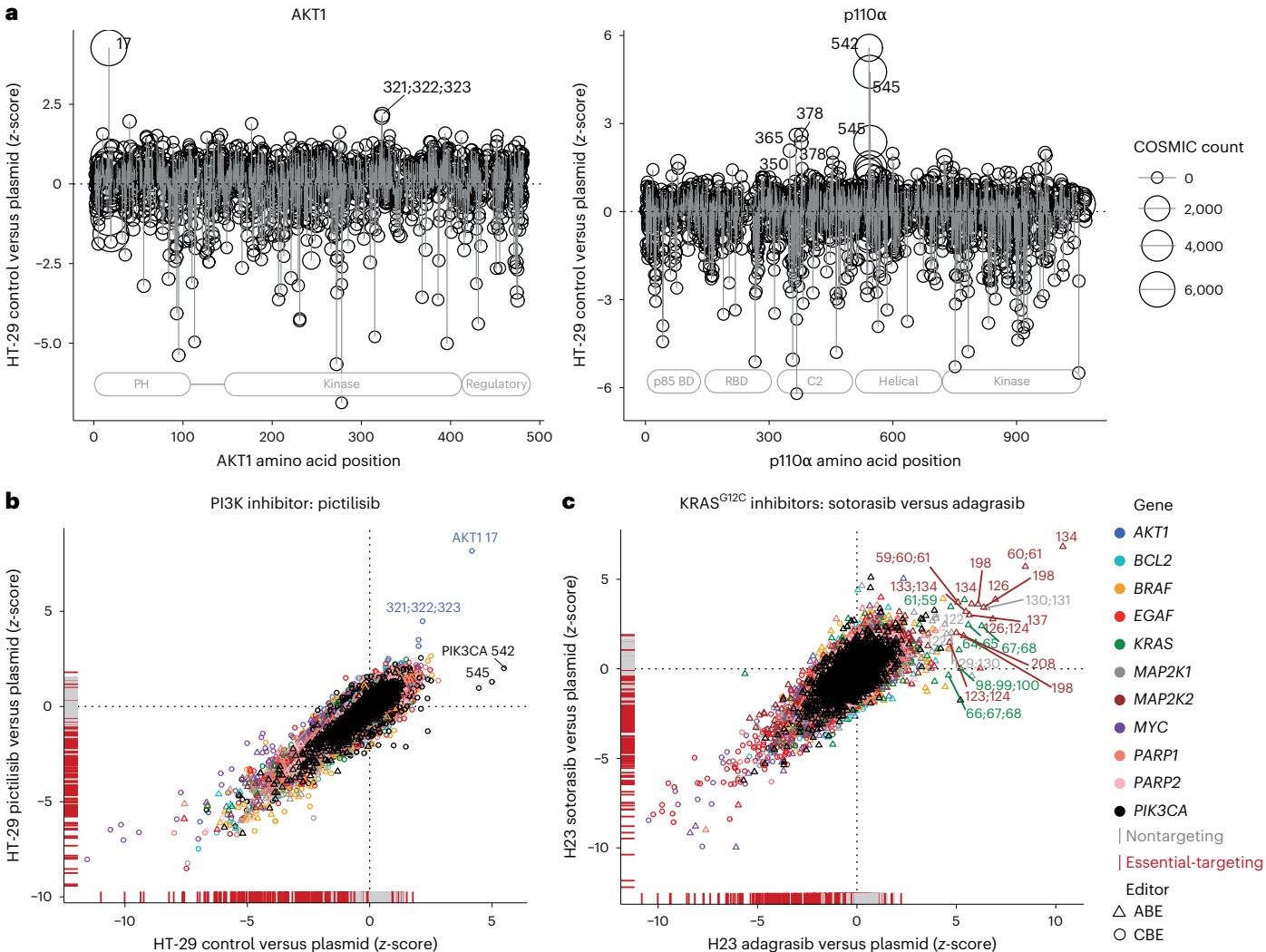

**Fig. 3 | Driver variants conferring drug resistance. a**, Base editing screens reveal clinically apparent hotspot mutations in oncogenes. Comparison of total COSMIC mutation counts for amino acid positions were compared with the *z*-score of gRNAs tiling across *AKT1* and *PIK3CA* in HT-29 cells. **b**, Drug resistance variants to the PI3K inhibitor, pictilisib, in HT-29 cells. Comparison of gRNA *z*-scores for the control treated arm versus plasmid library, and the drug-treated arm versus plasmid library is shown. **c**, Drug resistance variants to the KRAS[G12C] inhibitors, sotorasib and adagrasib, in H23 lung cancer cells. Comparison of gRNA *z*-scores for the sotorasib treated arm versus plasmid library, and the adagrasib treated arm versus plasmid library is shown. Predicted amino acid editing consequences are labeled for drug resistance variants. Data are the average of two independent screens performed on separate days.

PARP1 resistance variants included start lost variants (M1V) (Fig. 4a,b and Extended Data Fig. 4b), which presumably decrease PARP1 expression and reduce PARP trapping on DNA[50,58]. In addition, we observed rare missense variants that confered resistance to both inhibitors (for example, L390S/S391P; Fig. 4b and Extended Data Fig. 4b).

The identification of drug-sensitizing variants enables the stratification of patients for effective treatments and can identify combination therapies. Unlike other drugs tested in this study, we discovered a high number of drug-sensitizing variants in PARP1/2 relative to drug resistance variants, consistent with a previous base editing screen of PARP1 (ref. 5). We identified several predicted missense variants that sensitized to both olaparib and niraparib in PARP1. These drug-sensitizing variants were exclusively within the catalytic domain (ten out of ten) and predominantly within the helical subdomain (HD) (six out of ten; Fig. 4a), which has an autoinhibitory role through blocking NAD[+] binding[59]. For example, the helical subdomain missense variant PARP1 I691T, sensitized to both PARP inhibitors in base editing screens (Fig. 4a), in validation experiments using cell competition assays (Fig. 4b) and in proliferation assays (Extended Data Fig. 4c). In contrast,

PARP1 Y889C within the ART (ADP-ribosyl transferase fold) subdomain, conferred resistance to niraparib, but sensitized to olaparib (Fig. 4a–c). Consistently, PARP1 Y889C increased cytotoxic PARP trapping on DNA following DNA damage in the presence of low doses of olaparib and, conversely, reduced PARP trapping in the presence of niraparib relative to control cells in immunofluorescence and chromatin fractionation assays (Extended Data Fig. 4d,e). PARP1 Y889 is within the drug-binding pocket in the catalytic domain (Fig. 4d) and has been shown to be a critical residue for determining inhibitor specificity over related proteins such as tankryase 1 through van der Waals and pi-interactions with PARP inhibitors[60]. This provides a potential explanation for the disparate effects of the Y889C variant on sensitivity to different PARP inhibitors. Overall, these data imply that olaparib could be effective in treating niraparib-resistant cells harboring this variant.

## Drug resistance variants in EGFR
We investigated drug resistance to the EGFR tyrosine kinase inhibitors gefitinib and osimertinib. The archetypal EGFR T790M gatekeeper variant conferred the strongest resistance to gefitinib, consistent with

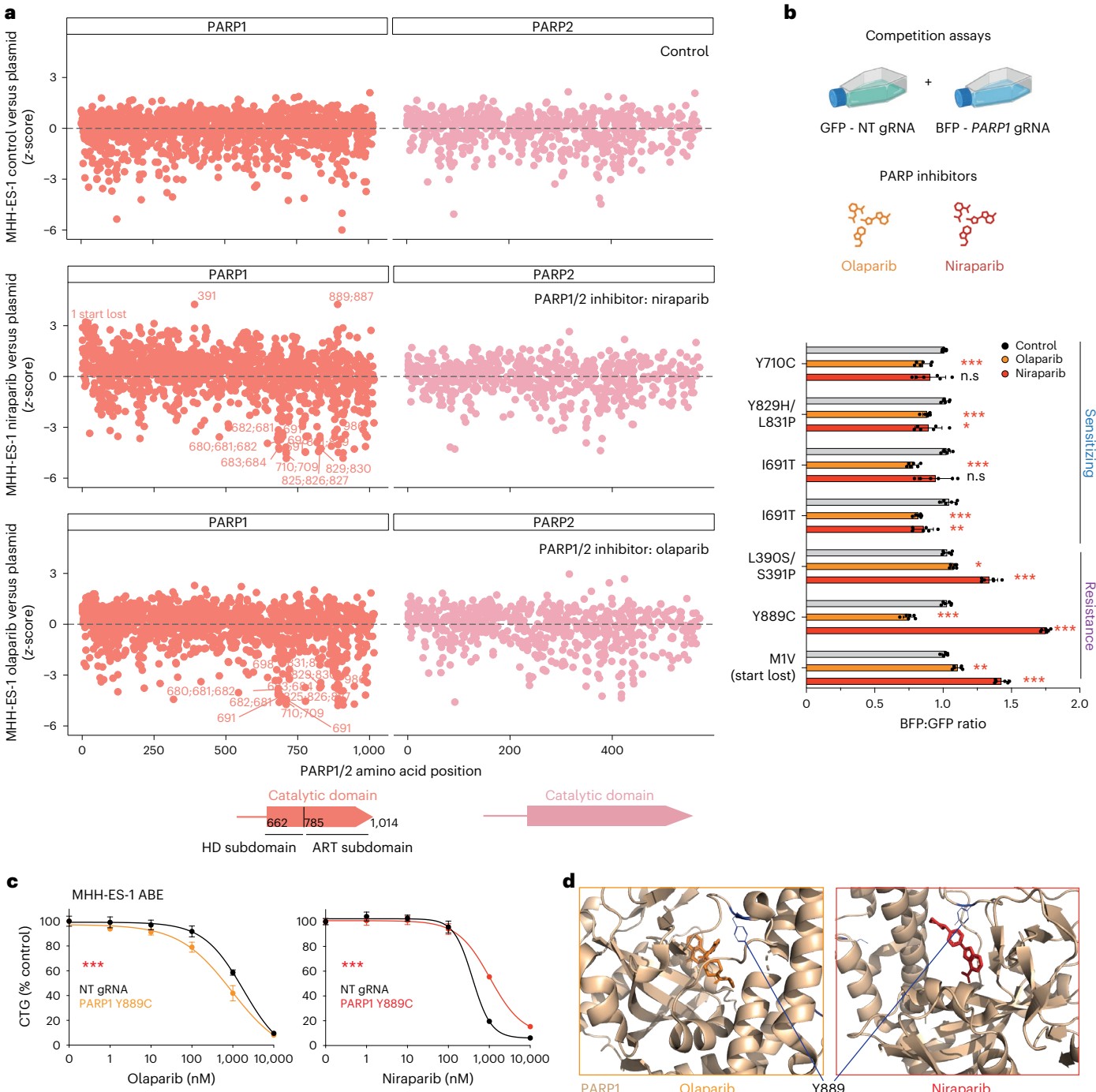

**Fig. 4 | Drug resistance and drug-sensitizing variants in PARP1. a,** Base editing screens of *PARP1* and *PARP2* in the presence of olaparib or niraparib reveal drug-sensitizing and drug resistance variants. Comparison of gRNA z-scores for the control treated arm, olaparib-treated or niraparib-treated arm versus plasmid library is plotted against the amino acid position. Predicted amino acid editing consequences are labeled for drug resistance variants. The position of the catalytic domain of PARP1 and PARP2 is shown. Screening data are the average of two independent screens performed on separate days. **b,** Competition assays comparing variants in PARP1 that modulate response to olaparib and niraparib. GFP NT gRNA cells were grown in competition with PARP1-edited cells expressing BFP and quantified with flow cytometry after 72 h in the presence or absence of PARP inhibitors (IC$_{50}$ concentrations; olaparib 510 nM, niraparib

330 nM). Two independent gRNAs installing the I691T variant were tested. Data represent the mean ± s.d. of two independent experiments, each performed in biological triplicate. Unpaired, two-sided Student's *t*-test comparing NT gRNAs with gRNAs targeting *PARP1*; ***P < 0.0001, **P = 0.0004 (I691T) or 0.0002 (M1V), *P = 0.014 (L390S/S391P) or 0.013 (Y829H/L831P). **c,** Dose response proliferation assay comparing the growth of MHH-ES-1 ABE cells harboring a NT control gRNA or a gRNA installing the PARP1 Y889C variant. Data are the average of two independent experiments, each performed in triplicate. Two-way ANOVA (analysis of variance); ***P < 0.0001. CTG; CellTiter-Glo. **d,** Crystal structures of PARP1 bound to olaparib (PDB 7AAD)[84] or niraparib (PDB 7KK5)[60] comparing the two binding modes of the inhibitors with respect to the Y889 residue. See also Extended Data Fig. 4. Schematic in **b** created with BioRender.com.

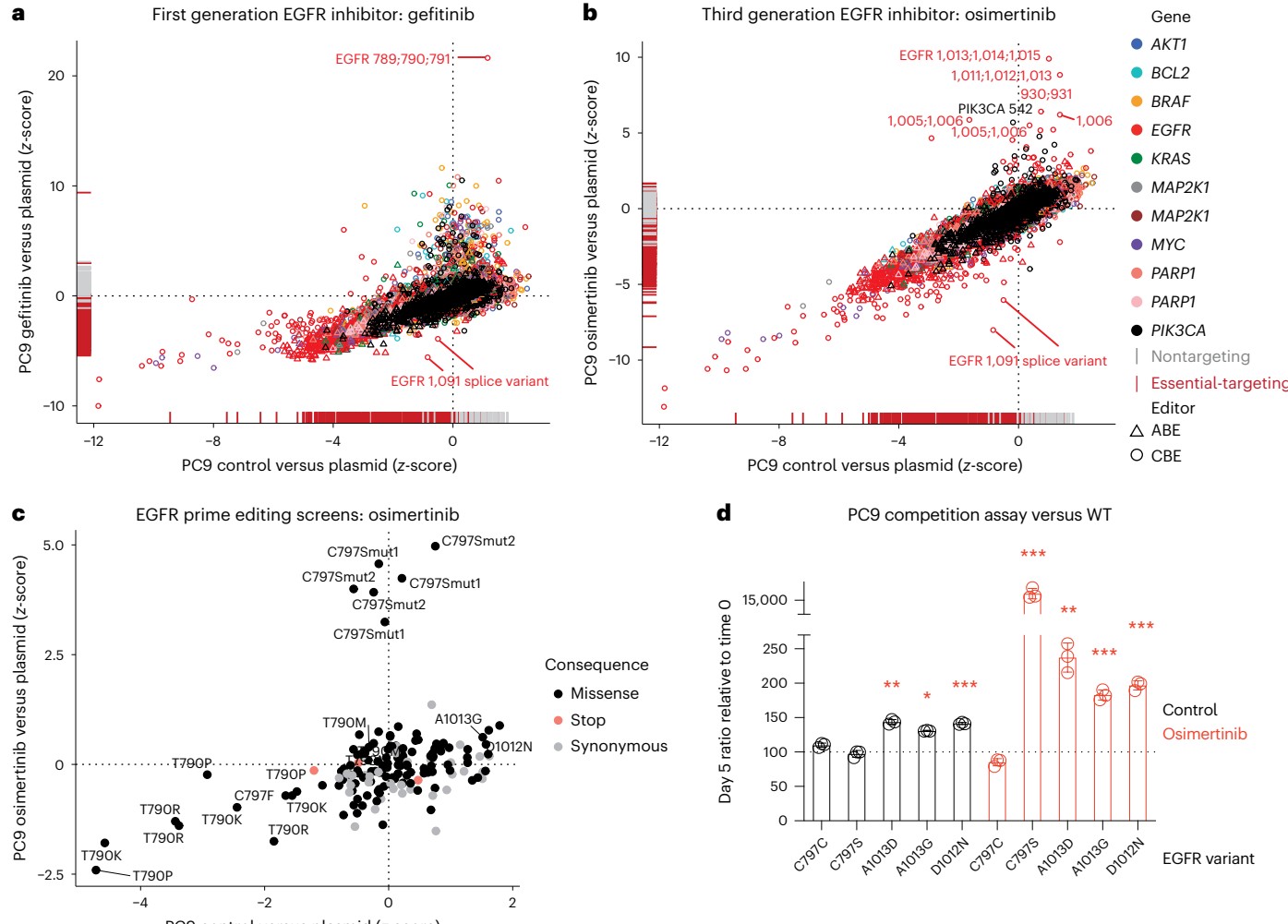

**Fig. 5 | Drug resistance and drug-sensitizing variants in EGFR. a**, Drug resistance variants to the EGFR inhibitor gefitinib, profiled with CBE and ABE base editors in PC9 lung cancer cells. Comparison of gRNA $z$-scores for the control treated arm versus plasmid library, and the drug-treated arm versus plasmid library is shown. **b**, Drug resistance variants to the EGFR inhibitor, osimertinib, profiled with CBE and ABE base editors in PC9 lung cancer cells. Comparison of gRNA $z$-scores for the control treated arm versus plasmid library, and the drug-treated arm versus plasmid library is shown. Data represent the average of two independent screens performed on separate days. **c**, Prime editing mutagenesis screens of EGFR in the presence and absence of osimertinib. PC9 $\Delta MLH1$ cells were prime edited for 7 days with doxycycline (1 μg ml[−1]) before growth for 10 days in DMSO (control) or osimertinib (75 nM). Data represent the $z$-score for each pegRNA derived from the average of two independent screens performed on separate days. Samples were compared with the plasmid library. **d**, Competition flow cytometry assays in PC9 $\Delta MLH1$ cells comparing the growth of NT gRNA GFP cells with epegRNA BFP cells harboring different EGFR variants in the presence and absence of osimertinib (75 nM) for 5 days. Data are normalized to day 0 ratios and represent the mean ± s.d. of biological triplicates. Unpaired, two-tailed Student's $t$-test comparing with the EGFR C797C synonymous variant control; *$P = 0.0003$, **$P = 0.0002$, ***$p < 0.0001$. Predicted amino acid editing consequences are labeled for drug resistance variant screens. See also Extended Data Fig. 5.

clinical data[2,61] (Fig. 5a). The second-generation EGFR inhibitor, osimertinib, is designed to target EGFR T790M[62]. For osimertinib, drug resistance conferring gRNAs were predicted to introduce EGFR missense variants surrounding residues D1006 and D1012, implicating these VUS in drug resistance (Fig. 5b). The activating PIK3CA E542K variant also conferred resistance to osimertinib, in line with clinical findings[63]. Notably, the clinically observed C797S resistance mutation[63–65] was absent as base editing was unable to introduce this specific T > A transversion mutation. Therefore, we used prime editing[15] in $\Delta MLH1$ PC9 cells as a complementary gene editing approach to screen variant function at six EGFR residues of interest identified from base editing screens and including C797 (Supplementary Note 2, Extended Data Fig. 5a,b and Supplementary Table 3). pegRNA sequencing confirmed a strong correlation between independent biological replicates (Extended Data Fig. 5c and Supplementary Table 4). Six out of six pegRNAs installing C797S were significantly enriched in the presence of osimertinib

(Fig. 5c). EGFR C797 and T790 substitutions to chemically distinct residues were depleted (Fig. 5c and Extended Data Fig. 5c), suggesting that particular variants of these key drug resistance residues can disrupt EGFR function and potentially explains why they are not observed clinically. Although we achieved the sensitivity required to detect loss-of-function effects, pegRNAs designed to install stop codons in EGFR were not depleted (zero of three; Fig. 5c), indicating low prime editing efficiencies for some pegRNAs, potentially relating to the high copy number of EGFR in PC9 cells[20]. In competition assays, EGFR C-terminal regulatory region prime edited variants D1012N, A1013D and A1013G had a modest growth advantage over wild type (WT) and C797C synonymous-variant-harboring cells, but this was significantly enhanced in the presence of osimertinib, suggesting these variants confer resistance, albeit less effectively than EGFR C797S (Fig. 5d). Overall, this highlights the complementarity of base editing screens to capture sites functionally involved in drug sensitivity across entire

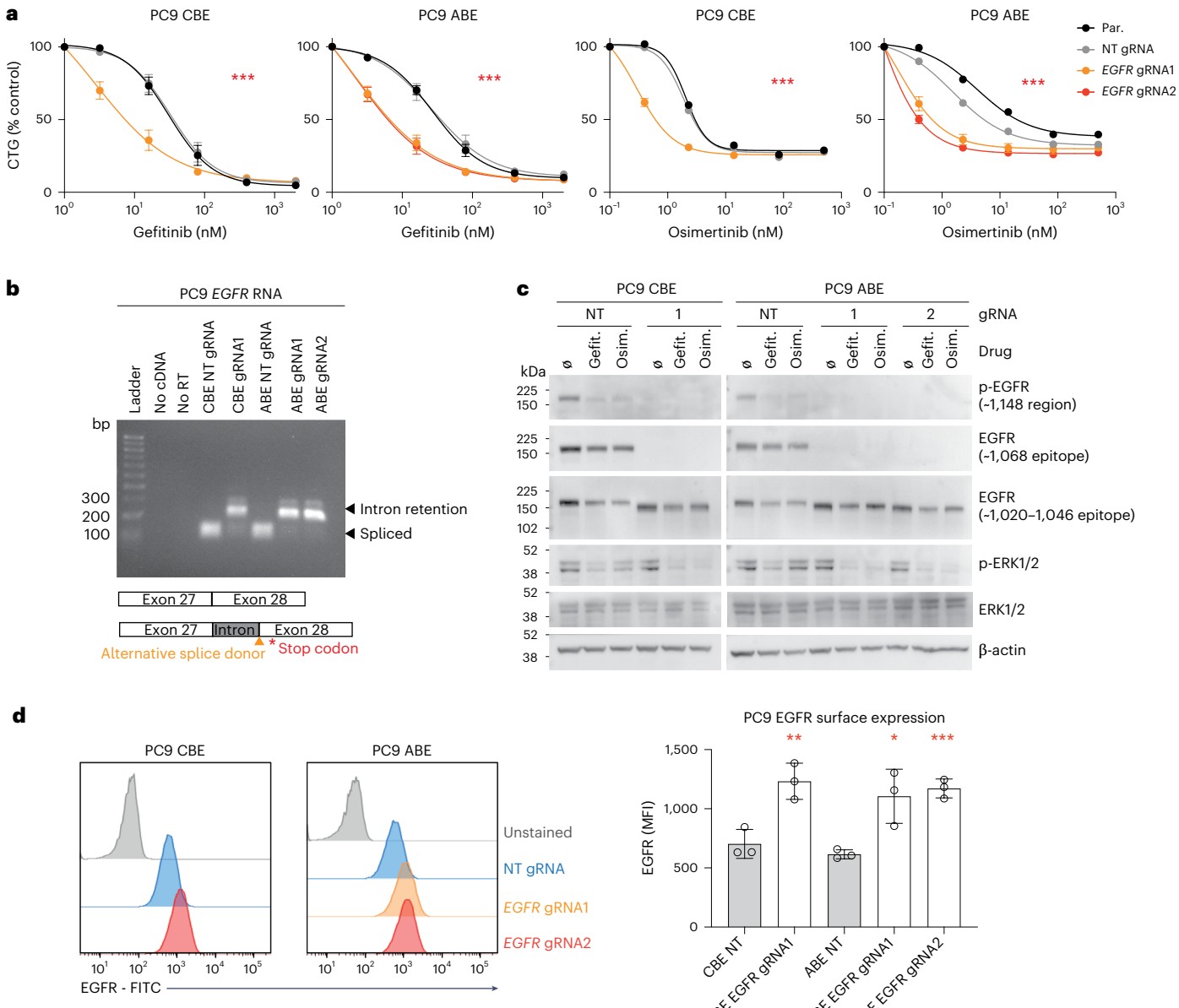

**Fig. 6 | EGFR C-terminal truncating variants sensitize to EGFR inhibitors.**
**a**, Increased gefitinib and osimertinib sensitivity in PC9 cells with EGFR C-terminal truncating mutations. Data represent the mean ± s.e.m. of two independent experiments, each performed in biological triplicate. Two-way ANOVA (analysis of variance) comparing with parental (Par.) response; ***$P$ < 0.0001. CTG; CellTiter-Glo. **b**, A drug-sensitizing base edit in *EGFR* causes loss of a splice donor site. The *EGFR* RNA splice variants are shown by migration of PCR products from cDNA. The larger PCR product in the mutant samples is due to retention of a short region of a downstream intronic sequence after exon 27, where an alternative splice donor is used. EGFR protein after residue 1,091 is not translated due to a frameshift leading to a stop codon. **c**, Western blotting of drug-sensitizing mutants reveals a C-terminal truncation in EGFR and confirms drug sensitization. PC9 CBE or ABE control cells (NT gRNA) or cells mutant for EGFR were treated with gefitinib (gefit.), osimertinib (osim.) or DMSO vehicle control (ø) for 24 h before analysis. Data are representative of two independent experiments. **d**, Flow cytometry analysis of EGFR protein surface expression in PC9 cells with WT EGFR or base-edited EGFR. Data are represented as a histogram or quantified as EGFR-FITC mean fluorescence intensity (MFI), and represent the mean of three independent experiments ± s.d. Unpaired, two-tailed Student's $t$-test; ***$P$ = 0.0004, **$P$ = 0.0096, *$P$ = 0.0217. See also Extended Data Fig. 6.

proteins, and for directing subsequent mutational scans at higher resolution using prime editing[15].

## EGFR truncating variants sensitize to EGFR inhibitors

Several base editing gRNAs targeting EGFR significantly increased sensitivity of PC9 cells to gefitinib and osimertinib but had minimal effect on cell growth in the absence of drug treatment (Fig. 5a,b). These gRNAs installed splice variants at residue E1091 of EGFR in the regulatory C-terminal domain. Using cell viability assays, we confirmed that these splice variants increased sensitivity to the EGFR inhibitors gefitinib and

osimertinib (Fig. 6a), as well as other chemically distinct EGFR inhibitors including erlotinib, lapatinib and the antibody cetuximab (Extended Data Fig. 6a). Sensitivity of the edited cells to the chemotherapeutics cisplatin and paclitaxel was largely unchanged, indicating an EGFR-specific mechanism of sensitization (Extended Data Fig. 6a). Sequencing confirmed that editing disrupted the splice donor site (Extended Data Fig. 6b), causing retention of intronic sequence after *EGFR* exon 27 (Fig. 6b) by the use of an alternative splice donor in the downstream intron (Extended Data Fig. 6c). This led to a frameshift and the introduction of a premature stop codon after residue E1091. WT and mutant EGFR proteins were expressed

at similar levels, but the migration of the mutant EGFR was consistent with a smaller protein lacking C-terminal epitopes and phospho-epitopes (Fig. 6c). MAPK signaling in EGFR-mutant cell lines was not altered relative to WT controls; however, reduction of p-ERK levels by EGFR inhibitors was more profound in mutant cells (Fig. 6c), consistent with increased sensitivity to EGFR inhibition. There was a significant increase in the surface expression of the EGFR C-terminal truncated protein relative to WT protein, suggesting differences in inhibitor sensitivity could relate to EGFR localization (Fig. 6d and Extended Data Fig. 6d), consistent with the C-terminal cytoplasmic tail having a regulatory role for receptor internalization[66]. Nonsmall cell lung cancer patients harboring rare tumor EGFR C-terminal truncations have responded to EGFR inhibition[67], warranting further investigation. These data highlight a mechanism of drug sensitization to EGFR inhibitors that could inform patient stratification and further our understanding of the mechanism of action of EGFR inhibitors.

## Perturb-seq functionally defines drug-resistant cell states

To further understand the signaling networks underpinning cancer drug resistance mechanisms, we examined the transcriptional programmes driven by drug-resistant variants (Fig. 7a). We recently described sc-SNV-seq[68]—a modified version of perturb-seq[69,70] that enables the systematic investigation of transcriptional changes in single cells harboring different endogenous variants installed with base editing. Using this methodology, we introduced a barcoded validation gRNA library ($n = 451$; Supplementary Tables 5 and 6) into HT-29 CBE and ABE cells and treated them acutely with the combination of dabrafenib and cetuximab, obtaining 27,823 cells with confident gRNA assignments (Fig. 7a, Extended Data Fig. 7a,b and Supplementary Note 3).

We compared the transcriptional response of groups of cells harboring gRNAs that conferred drug resistance to dabrafenib and cetuximab in base editing proliferation screens with those with control and NT gRNAs, revealing distinct transcriptional programmes (Fig. 7b). Of the 35 gRNAs tested that conferred resistance to dabrafenib and cetuximab in base editing proliferation screens, 22 (62.86%) elicited a significant transcriptional response versus NT gRNA-harboring cells (Supplementary Tables 7–10 and Supplementary Methods). Differential gene expression analysis of each targeting gRNA relative to the nontargeting controls revealed four clusters of gRNAs, which segregated by variant class and the target gene (Extended Data Fig. 7c), indicating that drug-resistant variant classes drive transcriptionally distinct cell states. This was more striking in the ABE dataset than the CBE dataset (Extended Data Fig. 7d) as the CBE dataset lacked drug addiction variants, which had the strongest transcriptional effect. Drug resistance genotypes had higher expression of transcripts involved in cell-cycle progression, such as *CDC20*, *BUB3* and *CDC6* (Extended Data Fig. 7e), consistent with an increase in S phase and a reduction in G1 occupancy in drug-resistant cells (Fig. 7c). Pathway analyses of differentially expressed genes in drug-resistant cells indicated significant enrichment in E2F target and G2M checkpoint control genes (Extended Data Fig. 8a and Supplementary Tables 11–14).

We directly compared the transcriptional programmes driven by different drug resistance variant classes. Differentially expressed genes between drug resistance classes were strongly associated with EGFR and MAPK pathway signaling (Fig. 7d,e). Drug addiction variants elicited activation of similar signaling pathways to canonical drug resistance variants, but activation was greater in drug addiction variants (Extended Data Fig. 8b). Ranking variants based on their impact on gene expression further highlighted the greater impact of drug addiction variants (Extended Data Fig. 8c,d), with some exceptional examples of canonical drug resistance variants that were closer to drug addiction in their transcriptional impact (for example, KRAS E63K/E62K and KRAS K117R/E/D119G). These data highlight that perturb-seq approaches can be used to assess the level of oncogenic signaling induced by different endogenous drug resistance variants.

## Drug resistance variants drive signatures of immune evasion

Some drug resistance variants caused a reduced expression of genes pertaining to the JAK–STAT signaling pathway (Fig. 7d); this was particularly apparent in drug addiction variants (Fig. 7e and Extended Data Fig. 8e). Transcripts critical for antigen presentation (for example *B2M*) were significantly downregulated by drug addiction variants in *MAP2K1* (Extended Data Fig. 8e). *B2M* and *HLA-A* were also downregulated by driver variants in *PIK3CA* (Fig. 7f). This is consistent with elevated RAS pathway signaling eliciting an immunosuppressive state[28,71–73], and implies that drug-resistant cells can adopt immune-resistant signatures[74]. We further verified the effects of drug resistance variants on B2M and HLA protein expression in HT-29 cells and a primary CRC organoid, CRC-9 (refs. 8,75,76) (Supplementary Note 4 and Extended Data Fig. 9a–d). In co-culture experiments with patient-derived, autologous anti-tumor T cells, pretreatment with MEK inhibitor enhanced cancer cell killing by T cells relative to controls treated with drug alone, except in tumor organoids harboring the MAP2K1 S194P drug addiction variant (Extended Data Fig. 9e).

To test whether the transcriptional cell states induced by drug resistance variants were predictive of patient treatment outcomes, we compared the transcriptional profiles associated with drug resistance variants with those identified in single-cell RNA sequencing of patient tumors ($n = 23$) from a recent phase-two clinical study of a BRAF, MEK and PD-1 inhibitor combination in patients with BRAF V600E CRC[28]. We generated a progression-free survival (PFS) outcome score based on the Spearman rank correlation of differentially expressed genes in drug-resistant cells from perturb-seq data and gene expression changes following treatment in responders (Supplementary Note 5). Drug-resistant variants had a significantly lower PFS outcome score compared with control and NT gRNAs (Fig. 7g). Drug addiction variants had the lowest PFS outcome score, consistent with signatures of higher MAPK pathway activation and reduced JAK–STAT signaling.

## A variant map indicates potential second-line therapies

Base editing installs edits within a specific activity window, which is focused roughly on nucleotides 4–9 of the gRNA target sequence[16,17,77]. Therefore, we determined the precise genomic variants installed for top-scoring drug resistance gRNAs from CBE and ABE base editing screens. We generated 46 isogenic HT-29 base-edited cell lines and performed next-generation sequencing of endogenous genomic loci before and after editing (Extended Data Fig. 10a). Amplicon sequencing demonstrated efficient installation of nonsynonymous edits exceeding 10% variant allele frequency (VAF) for 43 of 45 gRNAs, with a median VAF of 91% within the activity window (Extended Data Fig. 10b, Supplementary Note 6 and Supplementary Table 15).

We benchmarked our resistance variant map against COSMIC-curated drug response data[45] and literature[41,49] (Supplementary Note 7) revealing 35.29% concordance with screening data (Supplementary Table 16) and 252 edits at amino acid positions that had not been associated previously with altered drug response (for example, PARP1 Y889C, MAP2K1 S194P, BRAF K499E/R; Supplementary Table 17). Of the genotyped drug resistance variants, only 39% were within drug targets themselves, highlighting the potential of inhibiting orthogonal or downstream signaling pathways. Therefore, we explored whether verified drug resistance variants remained sensitive to other inhibitors tested in our screening dataset, representing possible alternative lines of therapy (Fig. 8). For example, the PIK3CA driver variants caused cross-resistance to several inhibitors, including EGFR and MEK inhibitors, but remained sensitive to pictilisib. Furthermore, activating drug addiction variants conferred resistance to several inhibitors (for example, MEK2 Y134H; resistance to six inhibitors), but could be sensitive to drug holidays in the context of BRAF V600E (Fig. 2d). Conversely, canonical drug resistance variants conferred resistance to

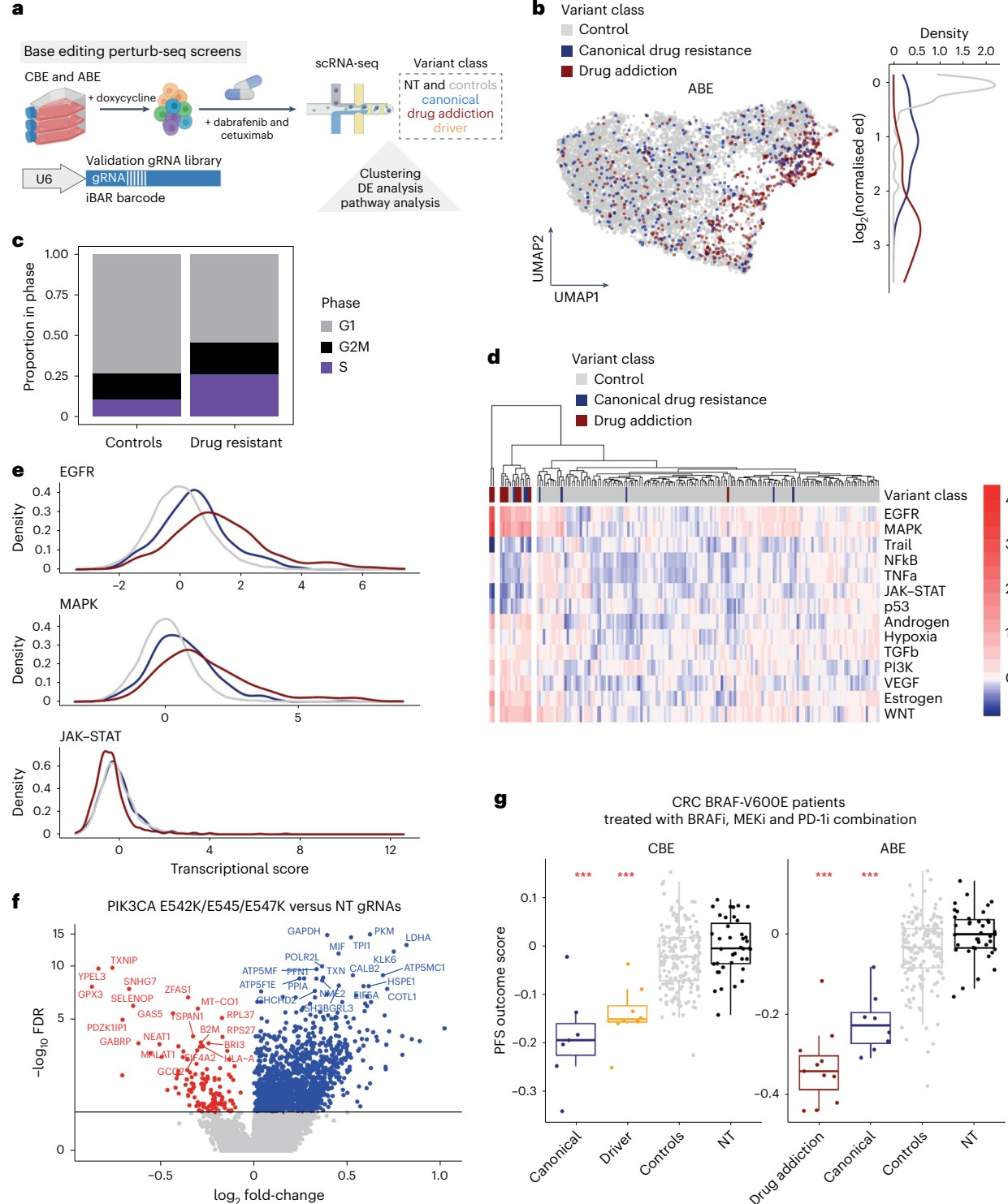

**Fig. 7 | Perturb-seq functionally defines drug-resistant cell states.**
**a**, Schematic of perturb-seq screening to investigate the transcriptomic effects of variants conferring resistance to dabrafenib and cetuximab in HT-29 cells using base editing. DE, differential expression analysis. **b**, Uniform manifold approximation and projection colored by variant class and normalized energy distances (ed) between NT gRNA cells and drug-resistant cells in ABE HT-29 cells treated with the combination of dabrafenib (80 nM) and cetuximab (1 μg ml⁻¹) for 16 h. **c**, Cell-cycle phase occupancy differences between cells with drug resistance conferring gRNAs and control gRNAs in the ABE perturb-seq dataset. $P < 2.2 \times 10^{-16}$ for ABE dataset, $P < 1.5 \times 10^{-14}$ for CBE dataset, chi-squared test, comparing control gRNAs with drug resistance gRNAs. **d**, Heatmap and hierarchical clustering of PROGENy pathway activity scores for each gRNA in the ABE dataset. **e**, Density plot of differences in PROGENy pathway scores between

the variant groups for the ABE dataset. **f**, Volcano plot of differentially expressed genes between NT gRNA control cells and cells with the PI3K p110α driver variant. Red, significantly downregulated transcripts (including *B2M* and *HLA-A*); blue, upregulated transcripts. **g**, Boxplot of PFS outcome score for each variant class, derived from CRC patients treated with BRAF, MEK and PD-1 inhibitor (PD-1i) combination therapy[28]. CBE and ABE perturb-seq scores are shown. ***$P < 0.01$, two-sided Wilcoxon rank-sum test compared with NT gRNAs. CBE; canonical drug resistance $n = 7$, driver $n = 8$, control $n = 172$, NT $n = 39$. ABE; drug addiction $n = 11$, canonical drug resistance $n = 8$, control $n = 115$, NT $n = 39$. Boxplots represent the median, IQR and whiskers are the lowest and highest values within 1.5× IQR. Control gRNAs are those that did not confer drug resistance in proliferation screens. See also Extended Data Figs. 7–9.

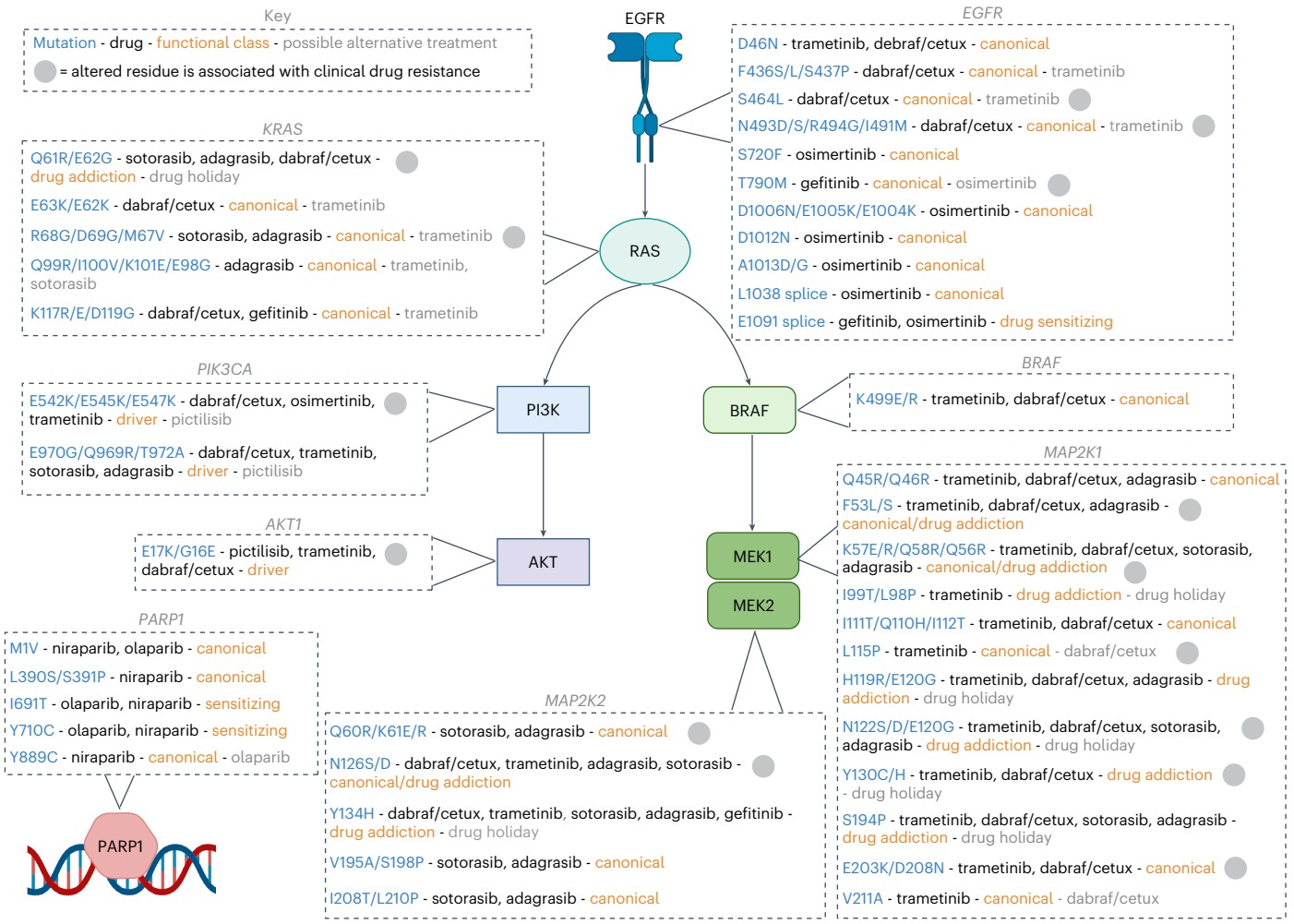

**Fig. 8 | A variant map indicates potential second-line therapies.** A variant function map highlights variants modulating drug sensitivity in cancer cells. Potential alternative treatments tested in this study are listed. Genotypes are from next-generation sequencing or Sanger sequencing of hits from base editing screens. See also Extended Data Fig. 10 and Supplementary Tables 15–17. Created with BioRender.com.

only a single inhibitor (for example, MEK1 L115P; trametinib), reflecting that the mechanism of action is related to drug binding, and thus remained sensitive to dabrafenib and cetuximab. KRAS inhibitors had broadly overlapping canonical drug resistance variants except for Q99R-region alterations, which were associated with resistance to adagrasib but not sotorasib, highlighting the subtly different inhibitor binding modes[49].

We identified rare canonical drug resistance variants (for example, BRAF K499E/R, KRAS E63K/E62K, KRAS K117R/E/D119G) that conferred resistance to multiple inhibitors, even though they did not affect cell growth in the absence of drug treatment. Perturb-seq showed these variants had an intermediate transcriptional impact (Extended Data Fig. 8d), consistent with a 'Goldilocks' level of oncogenic signaling that is well tolerated, even in the absence of pathway inhibition. In summary, we systematically categorize variants that confer resistance to several inhibitors and highlight possible alternative inhibitors or treatment schedules that could be effective in treating drug-resistant cancers (Fig. 8).

## Discussion

We report a prospective genetic landscape of drug resistance mechanisms in cancer, one of the most comprehensive functional investigations of genetic drug resistance mechanisms to date, comprising ten drugs and profiling 11 cancer genes spanning common drug targets and oncogenic pathways. We identify known mechanisms of drug resistance (for example, EGFR T790M–gefitinib[2,61]; MEK1 K57–cetuximab[39–41]) and previously unreported VUS causing drug resistance in vitro (Supplementary Table 15). We establish a framework for the functional classification of variants modulating drug sensitivity, which could inform clinical management (Fig. 8).

Single-cell transcriptomics enabled functional classification of drug resistance variants based on their mechanisms of action and transcriptional impact. The reduction in JAK–STAT pathway activity and reduced expression of antigen presentation machinery induced by drug resistance variants implies an overlap between transcriptional programmes driving drug resistance and immune evasion[28,71–74]. These data further support the rationale of combining targeted therapies with immune checkpoint blockade in cancer[71], and suggest that immunotherapies could be more effective in treatment-naïve patients.

Drug addiction variants displayed signatures of elevated MAPK pathway signaling in perturb-seq, and slower growth in the absence of drug treatment in screens and proliferation assays. Drug holidays have been proposed to mitigate the emergence of these drug-resistant cancer cell clones[42–44]. However, the clinical use of intermittent dosing has been limited so far, due partly to the lack of success of these trials in the absence of testing for the presence of such drug resistance variants[78]. The database of drug addiction

variants presented here, combined with the increasingly routine use of longitudinal tracking of variants with circulating tumor DNA profiling, could enable the accurate stratification of patients for drug holidays[79].

Extensions of this work could include the investigation of epigenetic or tumor-extrinsic drivers of drug resistance[1,80,81], and the in vivo assessment of how these variants affect drug sensitivity in the context of a physiological tumor microenvironment. Advancements in base editing and prime editing technology could increase the editing saturation and accuracy, and improve the interpretation of negative results, including the identification of benign variants without the need for extensive genotyping[68]. Nonetheless, we anticipate that our variant-to-function map will be useful to inform the interpretation of cancer genomics data in the clinical management of drug-resistant cancers.

High-throughput endogenous gene editing accurately mapped drug binding sites of several inhibitors with different modalities without previous structural information. Therefore, base editing screens could be useful to inform the design of new inhibitors targeting drug-resistant proteins (for example, EGFR T790M, EGFR S464L, EGFR C797S and MEK1 L115P), and to verify drug mechanism of action. Our prospective and systematic approach could be important for understanding genetic mechanisms of acquired resistance to new molecules in the future, even before the emergence of resistance in the clinic, thereby providing early insights to improve cancer treatment efficacy.

## Online content

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

[1]Translational Cancer Genomics, Wellcome Sanger Institute, Hinxton, UK. [2]Cancer Genome Editing, Wellcome Sanger Institute, Hinxton, UK. [3]Open Targets, Cambridge, UK. [4]EMBL-European Bioinformatics Institute, Cambridge, UK. [5]Cancer Research UK, Cambridge Institute, University of Cambridge, Cambridge, UK. [6]Gene Editing and Cellular Research and Development, Wellcome Sanger Institute, Hinxton, UK. [7]Bioscience, Oncology R&D, AstraZeneca, Cambridge, UK. [8]Department of Oncology, University of Turin, Turin, Italy. [9]Candiolo Cancer Institute, FPO-IRCCS, Candiolo, Italy. [10]Generative and Synthetic Genomics, Wellcome Sanger Institute, Hinxton, UK. [11]Department of Immunology and Molecular Oncology, Netherlands Cancer Institute, Amsterdam, the Netherlands. [12]Oncode Institute, Utrecht, the Netherlands. [13]Present address: Department of Mathematics and Statistics, University of Exeter, Exeter, UK. [14]Present address: Experimental Hematology Unit, IRCCS San Raffaele Scientific Institute, Milan, Italy. [15]Present address: Department of Pathology, Stanford University, Stanford, CA, USA. [16]Present address: Genentech, South San Francisco, CA, USA. ✉e-mail: matthew.coelho@sanger.ac.uk; mathew.garnett@sanger.ac.uk

## Methods

This research was conducted in accordance with institutional guidelines at the Wellcome Sanger Institute as outlined in the Good Research Practice Guidelines (v.4, 2021). This study complies with ethical regulations. Relevant ethical approval was obtained by Antoni van Leeuwenhoek, Netherlands Cancer Institute. We support inclusive, diverse and equitable research.

### Cell lines

All cell models used in this study (H23, PC9, HT-29, MHH-ES-1, HEK293T) were verified as mycoplasma-free and STR profiled in accordance with authentication guidelines[85]. Cells were cultured in RPMI medium with 10% FCS and 1× penicillin-streptomycin (ThermoFisher Scientific) at 37 °C in a humidified incubator with 95% air and 5% $CO_2$.

### gRNA library design and generation

To generate base editing gRNA controls we used SpliceR[21] and ranked the top three gRNAs using the following metric [cDNA disruption score × (ABEscore + CBEscore)]. For exonic gRNA designs we used BEstimate (manuscript in preparation; https://github.com/CansuDincer/BEstimate). gRNAs were designed against the reference genome GRCh38, and their design did not integrate cell line-specific SNVs. Oligo pools (Twist Biosciences) were PCR amplified and inserted into a *Bbs*I digested pKLV2-BFP-T2A-puroR lentiviral backbone (Addgene, plasmid no. 67974) using Gibson assembly (NEB). Libraries from Gibson assembly were ethanol precipitated before delivery into electrocompetent cells (Endura, Lucigen) by electroporation with several parallel transformations to maintain library representation, before propagation in LB supplemented with 100 µg ml⁻¹ ampicillin, shaking at 30 °C overnight. For virus packaging, HEK293T cells were cotransfected with the library plasmid pool, psPAX2 and pMD2.G plasmids using FuGene HD (Promega) in Opti-MEM (ThermoFisher Scientific). Viral particles were collected in media supernatant 72 h post-transfection, filtered and frozen.

For the perturb-seq library, we selected gRNAs that were top resistance hits for different drugs in our base editing screens. We introduced an iBAR barcode into the gRNA stem-loop[86], by using a primer with a hexanucleotide degenerate sequence (Sigma Aldrich) to amplify the oligo library (Twist Biosciences). This enables the identification of genetically identical clonal populations of daughter cells expressing the same gRNA.

### Base editing screens

Base editing cell lines were generated as described previously[8]. Briefly, we integrated doxycycline-inducible CBE or ABE NGN base editors into the *CLYBL* safe-harbor locus using CRISPR–Cas9 and homology-directed repair, and used blasticidin and mApple (fluorescence-activated cell sorting (FACS)) to select for base editing cell populations. We used BE-FLARE[87] (CBE) or a green fluorescent protein (GFP) stop codon reporter[88] (ABE) to measure overall base editing activity. For MHH-ES-1, we performed only ABE mutagenesis screens due to the poor performance of CBE in this cell model.

We transduced cells with the gRNA virus (plus 8 µg ml⁻¹ polybrene; ThermoFisher Scientific) to achieve an infection rate of ~30%, as measured by blue fluorescent protein (BFP) fluorescence with flow cytometry. We performed all base editing screens at ~1,000× coverage (estimated cells per gRNA). Cells were selected with puromycin (ThermoFisher Scientific) for 4 days before taking a time zero cell pellet and then induction of base editing with doxycycline (1 µg ml⁻¹; Sigma Aldrich) for 3 days. Screens were split into drug treatment arms or control arms for a further 11 days, with passaging or refreshing drug every 3–4 days. Drug concentrations were based on IC₅₀ from GDSC[19] and confirmed empirically using CellTitre-Glo (CTG) proliferation drug titration assays. Drug concentrations used in screens and validation experiments were as follows: trametinib 10 nM, pictilisib 1 µM,

dabrafenib 80 nM, cetuximab 1 µg ml⁻¹, sotorasib 2 µM, adagrasib 1 µM, gefitinib 75 nM, osimertinib 5 nM, olaparib 510 nM and niraparib 330 nM (Selleckchem). Each screen was repeated independently at least twice on separate weeks.

DNA was extracted from cell pellets (Qiagen) and the gRNA cassette was PCR amplified (PCR1, 28 cycles, 3 µg per reaction) with several reactions in parallel to maintain the complexity of the library. After column purification of PCR1 products (Qiagen), PCR products were indexed by PCR (PCR2, eight cycles, 0.2 ng per reaction). Libraries were sequenced on the HiSeq2500 (Illumina) using 19 bp SE sequencing on Rapid Run mode with a custom primer. All primers are listed in Supplementary Table 18.

### Screen analysis

For analysis of predicted edits and annotation of gRNAs, we used BEstimate (https://github.com/CansuDincer/BEstimate). To infer the effect of edits, we assumed a WT genome and did not consider SNPs or SNVs specific to the four cancer cell lines. To score gRNAs, we first normalized read counts by generating reads per million (RPM) (reads per total reads for sample × 1,000,000 + 1 pseudocount]. For replicate screens, we averaged the read count values of the same gRNAs to generate an average RPM. We then generated average log₂ fold change (L2FC) values as (log₂(RPM condition/RPM control)). The z-scores were generated as ((L2FC − mean L2FC)/s.d.), where s.d. represents the s.d. of the L2FC values for that comparison. We excluded gRNAs that had more than two perfect matches in the GRCh38 human genome (*n* = 134), and gRNAs with <100 read counts in the plasmid or any time zero sample in the screens (*n* = 27). For the few hit gRNAs with two perfect matches in the genome, we confirmed there was only one exonic target. We also excluded gRNAs specifically in the MHH-ES-1 screen that had a tenfold read count difference between replicates (*n* = 118) from downstream analysis of the MHH-ES-1 screens.

We assigned drug resistance hits as gRNAs with a z-score >2 in the presence of drug, and >1 in the presence of drug for each independent replicate. For variant classes based on proliferation screens, we assigned driver variants to gRNAs with a z-score >2 in the absence of drug and >1 in each independent replicate. Drug addiction variants were assigned to gRNAs with a z-score <−2 in the absence of drug (and <−1 in each independent replicate), and >2 in the presence of drug (and >1 in each independent replicate). Canonical drug resistance variants were assigned to gRNAs with a z-score >2 in the presence of drug (and >1 in each independent replicate) and did not fulfill driver or drug addiction phenotypic criteria. gRNAs with average z-scores <−2 in the presence of drug (and <−1 in each replicate), and >−2 in the absence of drug (and >−1 in each replicate), were assigned as drug-sensitizing hits. Control gRNAs did not satisfy any of the above criteria. Boolean classification of drug resistance irrespective of variant class was defined as z-score >2 in the presence of drug (and >1 in each independent replicate). For osimertinib and gefitinib resistance hits, the PC9 single-cell validation library screens were also used to further verify resistance (further threshold of z-score of >2 in these screens). We used MAGeCK[89] to generate statistics for base editing and prime editing screens. For base editing screens, we used the control gRNAs (NT, intergenic and nonessential) to generate a control distribution and implemented MAGeCK (RRA) to compare against the test gRNAs. For prime editing screens, we used the same approach but the control distribution was generated from pegRNAs designed to install synonymous variants. All gRNA and pegRNAs hits had MAGeCK *P* < 0.05 and a false discovery rate (FDR) < 0.1. For COSMIC variants, we downloaded nonsynonymous variants in our 11 target genes (February 2024) that pertained to resistance to inhibitors used in this study or similar drugs targeting the same oncogenes. This included patient samples and patient-derived xenograft in vivo samples with post-treatment biopsy genotyping of variants and an associated DRUG_RESPONSE field.

## Prime editing

*MLH1* was knocked-out (KO) in PC9 cells with CRISPR–Cas9 using a Cas9-T2A-EGFP (Addgene, plasmid no. 48140) plasmid using a gRNA listed in Supplementary Table 18. Cells were transfected (FuGene HD, Promega), sorted on EGFP by FACS as single cells into 96-well plates (BD Influx, BD Biosciences) and clones were tested for KO by western blotting (see below) with an MLH1 antibody (cat. no. 3515, Cell Signaling Technologies, 1:1,000). *MLH1* KO PC9 clone 1, or parental PC9 cells were cotransfected with a PiggyBac doxycycline-inducible PE2 plasmid with a constitutive GFP selection marker and PiggyBac transposon plasmid[90] at a 1:1 ratio. GFP positive cells were sorted by FACS as a pooled population of PE-expressing cells. EGFR C797S pegRNAs are listed in Supplementary Table 18.

PegRNAs were designed by taking the top three scoring pegRNAs as predicted by a recent pegRNA efficiency prediction algorithm, PRIDICT[91]. We used an end-to-end batching algorithm to call the PRIDICT tool for several query amino acids (https://github.com/mariemoullet/PRIDICT). We generated lentiviral pegRNAs (Twist Biosciences or IDT) as described above for base editing libraries, except we used a puromycinR plasmid using a *Bsm*BI entry site for the gRNA[90] (modified from Addgene, plasmid no. 84752). After selection with puromycin, prime editing was performed by 5–7 days exposure to doxycycline (1 μg ml$^{-1}$).

## Prime editing screens

For prime editing, we selected coding positions of EGFR that scored in the base editing screens for EGFR inhibitors and selected all possible amino acid mutations possible with an SNV for the following residues: A1013, C797, T790, D1006, D1012 and V1011. We designed the pegRNAs using PRIDICT as above and filtered for pegRNA inserts that were <200 bp in length after the addition of Gibson homology arms for cost-effective DNA oligo synthesis (Twist Biosciences). A library of 162 vectors was cloned using Gibson assembly (NEB), to generate a pool of plasmids for lentivirus production, as described above for base editing screens. PC9 *MLH1* KO PE2-GFP-expressing cells were infected such that approximately 15–30% of cells were infected and to achieve a library coverage of approximately 10,000×, and cells were subsequently selected with puromycin for 3 days (0.75 μg ml$^{-1}$). Prime editing was induced for 7 days with doxycycline (1 μg ml−1), before selection with osimertinib (75 nM) for 10 days, or grown in dimethylsulfoxide (DMSO) (control). Cells were pelleted, washed in PBS and DNA was extracted (Qiagen) for pegRNA amplification using primers listed in Supplementary Table 18. Amplicons were sequenced with paired-end sequencing on a MiSeq v.2 micro kit (Illumina).

For validation studies, we used an epegRNA design of pegRNAs that had scored in the prime editing screens (Supplementary Table 18). An NT gRNA vector expressing GFP was used as a WT control in competition assays against epegRNAs expressing BFP (including a EGFR C797C synonymous control), which were analyzed by flow cytometry 5 days after seeding. Ratios of BFP to GFP were normalized to the day zero timepoint.

## Proliferation assays

For validation assays we used the incucyte (Sartorius) or CTG assay (Promega). gRNAs were ordered as oligonucleotides (Sigma Aldrich), annealed and assembled by Golden Gate and sequence verified (Eurofins) as described[8]. This allowed us to validate gRNAs at scale in an arrayed format. gRNA sequences are listed in Supplementary Tables 18 and 19.

## Senescence assays

Oncogene-induced senescence was assessed by staining with the β-galactosidase staining kit 48 h after drug treatment in 96-well plates following manufacturer's instructions (Cell Signaling Technology, cat. no. 9860) and images were acquired on the EVOS XL Core microscope (ThermoFisher Scientific).

## Western blotting

PC9 cells were lysed in sample loading buffer (8% SDS, 20% β-mercaptoethanol, 40% glycerol, 0.01% bromophenol blue, 0.2 M Tris-HCl pH 6.8) and boiled for 5 min before loading onto a NuPAGE 4–12% Bis-Tris gel (ThermoFisher Scientific). Proteins were transferred to a polyvinylidenedifluoride membrane before blotting with the following primary antibodies: EGFR total (1068 epitope, cat. no. 2232, 1:1,000 dilution), p-EGFR (1148 region, cat. no. 4404, 1:1,000 dilution), β-actin (cat. no. 4970, 1:1,000 dilution), p-ERK (cat. no. 9101, 1:1,000 dilution), ERK total (cat. no. 9102, 1:1,000 dilution) (Cell Signaling Technology), EGFR epitope 1020-1046 (cat. no. 610017 BD Biosciences, 1:1,000 dilution). Secondary antibodies (anti-mouse and anti-rabbit) were conjugated to horseradish peroxidase (cat. nos. 31460 and 31430, ThermoFisher Scientific, 1:5,000 dilution).

## Flow cytometry

PC9 cells were harvested by trypsinization and washed in FACS buffer (0.5% FCS < 2 mM EDTA in PBS) before staining with anti-EGFR-FITC antibody (cat. no. MA5-28104, clone ICR10, ThermoFisher Scientific, 1:100 dilution) for 25 min on ice in the dark. For B2M (cat. no. 395805, BioLegend) and HLA staining (cat. no. MA5-44095, ThermoFisher Scientific), cells were treated the indicated drug for 48 h before analysis. Cells were washed twice in FACS buffer, incubated with 4,6-diamidino-2-phenylindole (1 μg ml$^{-1}$, ThermoFisher Scientific) before filtering through a nylon mesh cell strainer, and analysis on an LSRFortsessa (BD Biosciences). Flow cytometry data were acquired using FACSDiva software v.9 (BD Biosciences) and analyzed using FLowJo v.10 or FCS Express v.7.

## RNA analysis

RNA was extracted (RNeasy, Qiagen), DNA removed with DNaseI digest (Qiagen), and reverse transcription performed using poly dT priming (SuperScript IV, ThermoFisher Scientific). PCR was performed on cDNA from WT and mutant cells and intron retention was verified by the size of the resulting PCR product as assessed by DNA gel electrophoresis. PCR primers are listed in Supplementary Table 18.

## Amplicon sequencing

From our drug resistance hits from base editing screens (z-score of >2 and >1 in each replicate with FDR < 0.1 and $P < 0.05$), we filtered for nonsynonymous coding mutations in target genes that were nonredundant and had not already been genotyped by Sanger sequencing in validation studies (for example, EGFR splice variants and PARP1 drug-sensitizing variants). Given the large number of variants, we preferentially selected proximal variants such that we could cover several variants in a single amplicon. In total, we analyzed 46 gRNAs targeting seven genes over 21 amplicons in two separate experiments. HT-29 cells were directly lysed with a direct PCR lysis buffer supplemented with 100 μg ml$^{-1}$ proteinase K following manufacturer's instructions (Viagen Biotech); 2 μl of DNA lysate was used as input in 25 μl PCR reactions (KAPA HiFi HotStart ReadyMix, Roche) using amplicon sequencing primers listed in Supplemental Table 7. PCR cycle number for PCR1 was determined empirically. PCR products were SPRI purified (AMPure XP, Beckman Coulter), before indexing PCR2 (ten cycles), SPRI purification, quantification (Bioanalyzer, Agilent), equimolar pooling and sequencing on a MiSeq v.2 150 bp PE kit (Illumina). Similarly, Sanger sequencing (Eurofins) of base edits were performed by PCR amplification of endogenous edited loci using primers listed in Supplementary Tables 18 and 19. For analysis, we used BCFtools (v.1.20) and vafCorrect (v.5.4.0), with a read-depth cut-off of >1,000 reads, and a VAF > 0.1 for variants. We report the editing outcomes from amplicon sequencing in Supplementary Table 15. A total of 43 gRNAs produced nonsynonymous variants with >10% variant allele frequency and were considered in downstream analysis. For the annotation of MAP2K2 Y134H, we deduced this was the variant driving drug resistance as we observed two different gRNAs

making more complex edits with Y134 as the common major allele (97% median allele frequency for one gRNA). Finally, we removed the MAP2K2 F213C variant from downstream analysis as it was already detected in unedited DNA samples.

## PARP trapping assays

For PARP trapping immunofluorescence assays, MHH-ES-1 ABE cells were plated in 96-well optical plates (Greiner, cat. no. GN655906). The following day, PARP inhibitors were added in a serial dilution for 4 h in the presence of the DNA damaging agent, methyl methanesulfonate (MMS, Sigma Aldrich, cat. no. 129925) at a final concentration of 0.01%, as previously described[92]. Cells were permeabilized with ice-cold cytoskeleton buffer for 6 min on ice (10 mM PIPES pH 6.8, 200 M sucrose, 200 mM NaCl, 3 mM MgCl$_2$ and 0.3% Triton X-100) to remove soluble, nonchromatin-bound PARP. Nuclei were washed on the plate with ice-cold-PBS, fixed for 20 min with 4% PFA, washed again with ice-cold PBS before fixation with ice-cold methanol at −20 °C overnight. After a 1 h incubation in blocking buffer at room temperature (PBS supplemented with 1% BSA), staining of PARP1 was performed 4 °C overnight (Sigma, cat. no. WH0000142M1). Secondary goat anti-mouse AlexaFlour 488 secondary antibody was used for imaging, and 4,6-diamidino-2-phenylindole was used as a nuclear counterstain. The mean nuclear fluorescence intensity was measured using the CellInsight CX5 HCS Platform (ThermoFisher Scientific).

For cell fractionation assays, cells were treated with 0.01% MMS in the presence of 3 µM olaparib or niraparib for 4 h, washed in PBS and pelleted, then snap frozen on dry ice. Cell pellets were processed by subcellular fractionation using the Subcellular Protein Fractionation Kit for Cultured Cells (ThermoFisher Scientific, cat. no. 78840) according to manufacturer's instructions. The soluble nuclear and the chromatin-bound fractions were further analyzed by western blot to measure PARP1 trapping. Equal volumes were loaded on 4–12% Bis-Tris NuPAGE gels and analyzed by standard immunoblotting using the following primary antibodies; PARP1 (cat. no. 9532, clone 46D11, 1:1,000 dilution), lamin A/C (cat. no. 2032, 1:1,000 dilution), histone H3 (cat. no. 3638, clone 96C10, 1:1,000 dilution) (all from Cell Signaling Technology). Visualization of crystal structures was performed using PyMOL v.2.4.1.

## Perturb-seq

After infection with the barcoded validation gRNA library virus to achieve an infection rate of ~20%, HT-29 cells were selected with puromycin and then the cell population was bottlenecked to 50,000 cells to increase the numbers of genetic clones. CBE and ABE base editing was induced for 48 h with the addition of doxycycline (1 µg ml$^{-1}$). Base-edited cells were then maintained in puromycin (0.5 µg ml$^{-1}$) to maintain gRNA expression, and acutely treated with doxycycline (to express base editor and help stabilize gRNAs) and dabrafenib and cetuximab for 16 h before harvesting the cells for transcriptomics. Single-cell suspensions of 60,000 cells per reaction were prepared for superloading on the Chromium X according to manufacturer's instructions using Chip N and the Chromium Next GEM 5′ HT v.2 kit (10x Genomics). We spiked in 2.2 µl of a 10 µM solution of a primer for direct capture of the gRNA in the reverse transcription reaction. The gRNA libraries were prepared separately using a nested PCR. All primers are listed in Supplementary Table 18. The cDNA gene expression libraries were prepared according to manufacturer's instructions (10x Genomics), before pooling at 1:10 molar ratio with the gRNA libraries (gRNA:cDNA) and sequencing on one lane of a NovaSeq 6000 S4 (Illumina). Perturb-seq analysis is described in the Supplementary Methods.

## Tumor organoid T cell co-culture assays

Tumor organoid co-culture assays were performed as described[8,75,76], except the tumor cells were not pretreated with interferon (IFN)-gamma

as the proficiency of antigen presentation was being tested. For competition assays, BFP gRNA tumor cells were co-cultured with NT gRNA GFP-expressing cells in a competition assay. Briefly, PMBCs were cultured in 96-well plates coated with anti-CD28 antibody for 24 h with IL-2 (150 U ml$^{-1}$; ThermoFisher Scientific). Autologous CRC-9 tumor organoids were maintained in 80% basement membrane extract (R&D Systems), pretreated for 48 h in the MEK inhibitor trametinib (25 nM), and plated (5,000 cells of BFP gRNA and 5,000 cells of GFP gRNA) in suspension at a 3:1 E:T ratio for 72 h before FACS analysis. T cell killing assays were performed in the presence of the anti-PD-1 antibody nivolumab (20 µg ml$^{-1}$; Selleckchem) in RPMI medium supplemented with human serum and primocin (Invivogen). Then, 123count eBead counting beads (ThermoFisher Scientific) were used to calculate absolute cell counts from flow cytometry analysis. Relative T cell killing was quantified by normalizing the number of cancer cells in comparable conditions in the absence of T cells.

## Statistics and reproducibility

Statistical tests, exact value and description of $n$, definition of center, dispersion and precision measures are described in the figure legends. For base editing screen analysis with MAGeCK, $P < 0.05$ and an FDR of 0.1 were used as significance thresholds. All screens were performed at least twice on separate days. For Student's $t$-test, significance was defined as $P < 0.05$. No statistical method was used to predetermine sample size and the experiments were not randomized. The investigators were not blinded to allocation during experiments and outcome assessment. Data exclusion is specified in the base editing 'Screen analysis' section of Methods, based on the gRNA with off-targets or based on replicate correlation in MHH-ES-1 screens.

## Reporting summary

Further information on research design is available in the Nature Portfolio Reporting Summary linked to this article.

## Data availability

Sequencing data are deposited within the European Nucleotide Archive (ENA) and European Genome-phenome Archive (EGA) and accessions are described in Supplementary Table 20 (ENA: ERP146490, ERP148732, ERP144241, ERP141719, ERP156437; EGA: EGAS00001006683, EGAS00001006170, EGAS00001006169, EGAS00001006093, EGAS00001006092, EGAS00001006091) and can be found at https://www.ebi.ac.uk/ena/browser/home and https://ega-archive.org/studies/. All genomic indexing is relative to GRCh38 genome assembly https://www.ensembl.org/Homo_sapiens/Info/Index. COSMIC variants were downloaded in February 2024 v.99 (https://cancer.sanger.ac.uk/cosmic/download/cosmic). Screening data are available in Supplementary Tables 2 and 4. Screen $z$-scores are available on the MAVE database[93] (urn:mavedb:00001204; https://www.mavedb.org/experiments/urn:mavedb:00001204-a). Source data are provided with this paper.

## Code availability

Code used to analyze base editing screens can be found on GitHub: https://github.com/MatthewACoelho/Res1_analysis. Code used to analyse the single-cell screens can be found at https://github.com/MarioniLab/BE_perturb_seq_drug_resistance.

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

## Acknowledgements

We wish to acknowledge the contribution of the Cancer Ageing and Somatic Mutation Support team at the Wellcome Sanger Institute. This research was funded in whole, or in part, by the Wellcome Trust Grant 206194 (M.J.G.). M.E.S. is supported by the Wellcome Trust (220442/Z/20/Z). Figures 1a, 4b and 8 were created with BioRender.com.

## Author contributions

M.A.C. and M.J.G. devised the study, M.A.C. carried out experiments and M.A.C., M.E.S., E.K., C.D. and S.B. performed data analysis. S.C., S.F.V. and A.W. assisted with experiments and C.D., M.M., K.M. and G.P. assisted with gRNA and pegRNA library design. J.K., L.P., S.C., J.C.M., D.C., M.S. and A.B. developed and provided essential reagents and conceptual advice. G.I. assisted with PARP trapping experiments, data analysis and provided essential reagents. J.V.F. provided conceptual advice on PARP variants and PARP inhibition and trapping. E.E.V., V.V. and C.M.C. provided essential reagents relating to organoid co-cultures. M.A.C. and M.J.G. wrote, and all authors reviewed, the manuscript.

## Competing interests

M.J.G. has received research grants from AstraZeneca, GSK and Astex Pharmaceuticals, and is a founder and advisor for Mosaic Therapeutics. J.C.M. has been an employee of Genentech since September 2022. A.B. is a founder and consultant for EnsoCell Therapeutics. M.A.C. and M.J.G. are inventors on patent applications that encompass work described in this paper. J.V.F. and G.I. are employees and shareholders of AstraZeneca. E.E.V. is a founder and advisor for Mosaic Therapeutics. The other authors declare no competing interests.

## Additional information

**Extended data** is available for this paper at https://doi.org/10.1038/s41588-024-01948-8.

**Correspondence and requests for materials** should be addressed to Matthew A. Coelho or Mathew J. Garnett.

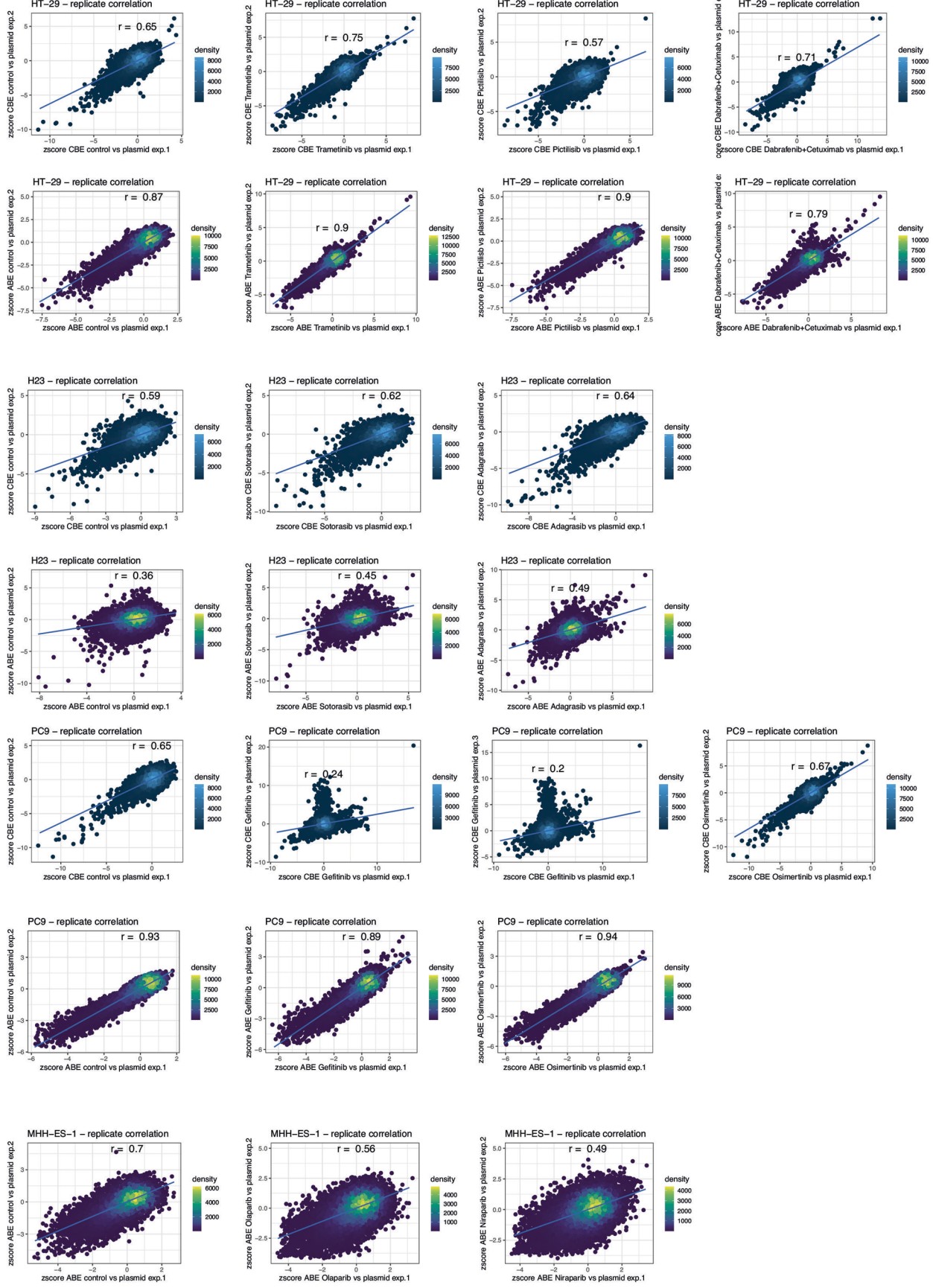

**Extended Data Fig. 1 | Base editing screens map functional domains in driving oncogenes.** Replicate correlation for CBE and ABE screens across four cancer cell models; HT-29, H23, PC9 and MHH-ES-1. Pearson correlation coefficient values (r) between independent replicate screens are shown. Low correlation was observed for replicates of PC9 screens with gefitinib, which may relate to a high degree of enrichment of resistant, EGFR T790M base edit harbouring cells.

**a**

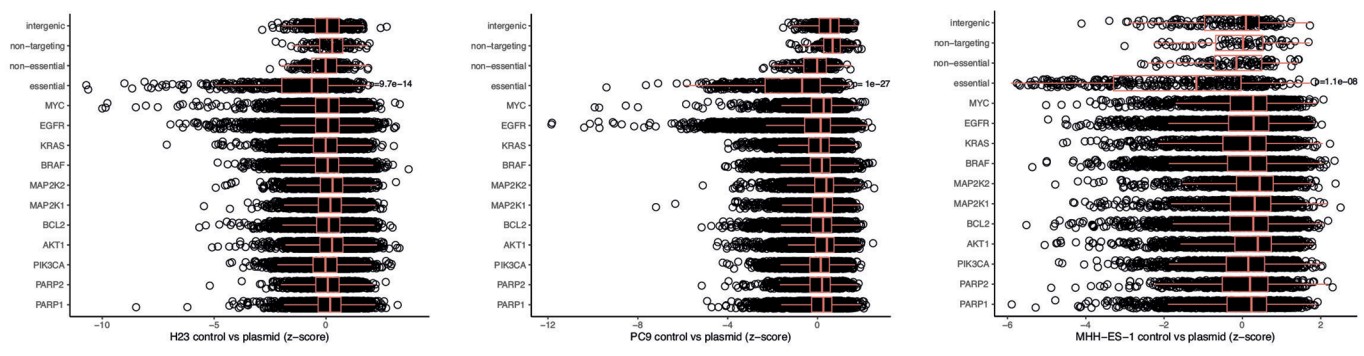

**b**

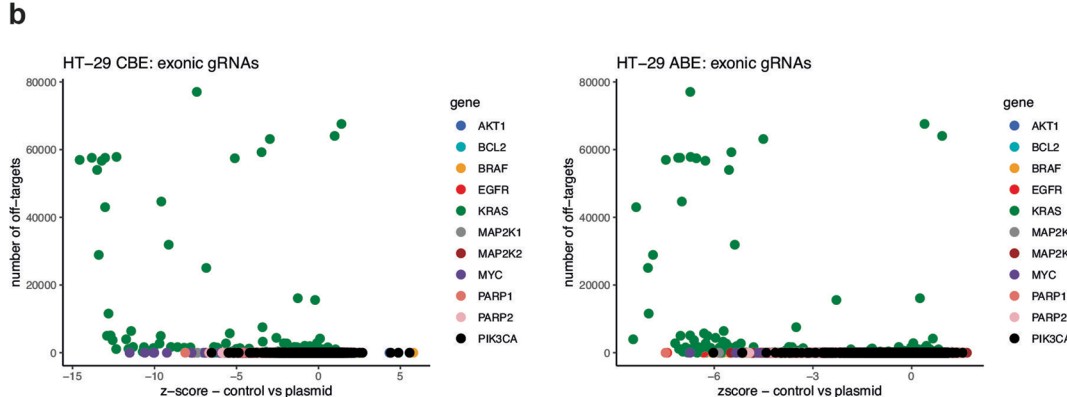

**c**

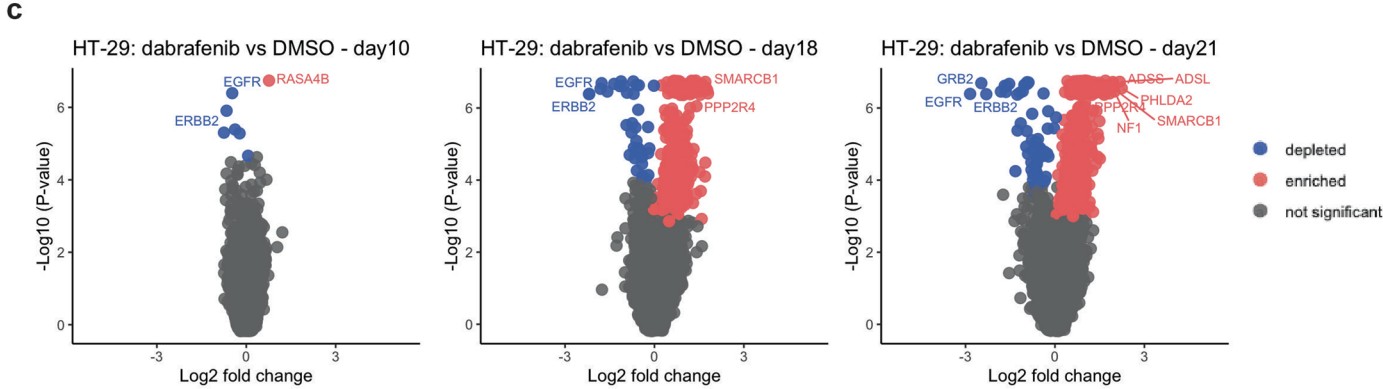

**d**

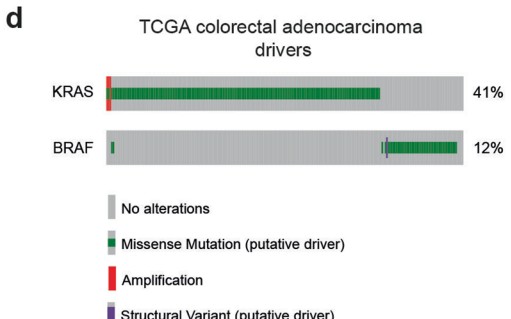

**Extended Data Fig. 2 | See next page for caption.**

**Extended Data Fig. 2 | Variants modulating drug sensitivity cluster into four functional classes. a)** Base editor screens in H23, PC9 and MHH-ES-1 cancer cells targeting 11 cancer genes show depletion of gRNAs targeting essential genes demonstrating base editing activity. Unpaired, two-tailed Student's t-test comparing non-targeting gRNAs ($n = 114$) to gRNAs targeting essential gene splice sites ($n = 632$) in CBE and ABE screens. For MHH-ES-1, ABE screens are shown (NT; $n = 57$; essential-targeting, $n = 306$). Boxplots represent the median and interquartile range (IQR), and whiskers represent the lowest and highest values within 1.5 x the IQR. **b)** Number of off-target sites plotted against the z-score for base editing gRNAs. A high number of off-targets for a small number of *KRAS UTR*-targeting gRNAs is associated with severe gRNA depletion. These were filtered out of downstream analysis. **c)** Our previously reported whole-genome CRISPR-Cas9 KO screen in HT-29 cells in the presence of dabrafenib (0.1 μM) across three time-points. Volcano plot showing *EGFR* KO as the top sensitising hit. Data are the average of two independent screens and significance was determined with MAGeCK, with a threshold of p-value < 0.05 and FDR < 0.05. **d)** TCGA oncoprint (pan-cancer cohort, $n = 526$) of colorectal adenocarcinomas with alterations in *KRAS* and *BRAF*. Mutual exclusivity p-value < 0.001 derived from two-sided Fisher exact test, q-value < 0.001 derived from Benjamini-Hochberg FDR correction procedure for multiple hypothesis testing.

**a**

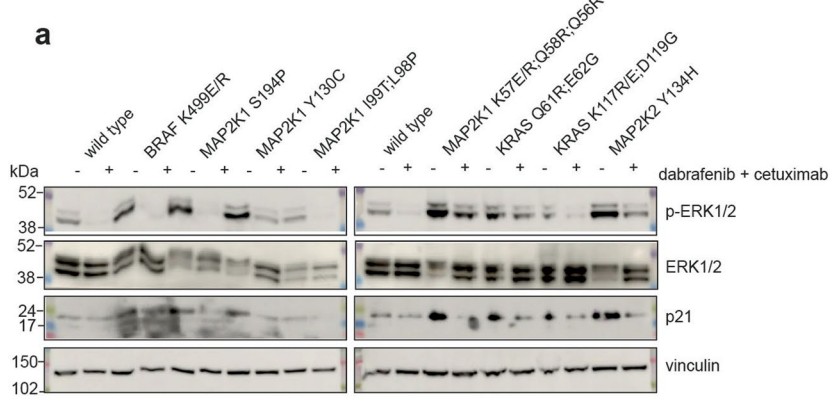

**b**

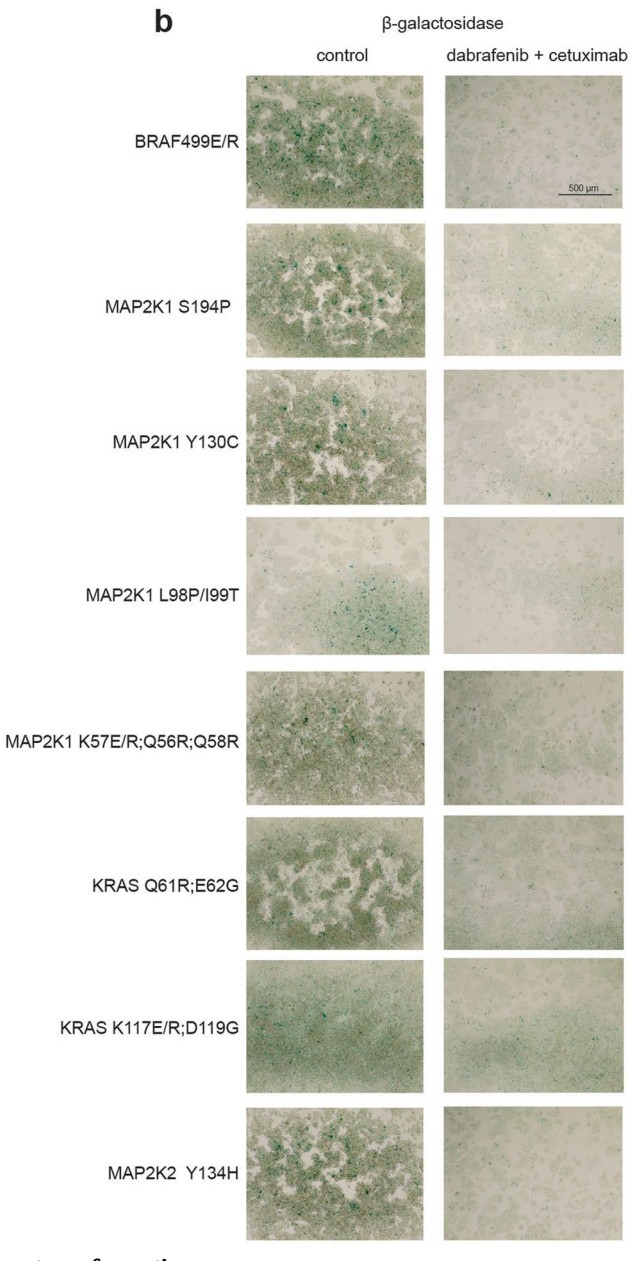

**Extended Data Fig. 3 | See next page for caption.**

**Extended Data Fig. 3 | Validation of cancer drug addiction variant phenotypes. a)** Western blotting of drug resistance variants from base editing screens in HT-29 cells conferring resistance to dabrafenib and cetuximab combination therapy. HT-29 cells harbouring the indicated variants were treated with dabrafenib (80 nM) and cetuximab (1 µg/ml) or DMSO (control) for 24 h before analysis. Data are representative of two independent experiments (see also Fig. 2). **b)** Microscopy images of ß-galactosidase assays performed to measure the induction of senescence. HT-29 cells harbouring the indicated variants were treated with dabrafenib (80 nM) and cetuximab (1 µg/ml) or DMSO (control) for 48 h before analysis. Representative images from two independent experiments. Scale bar indicates 500 µm. Genotyped variants are shown.

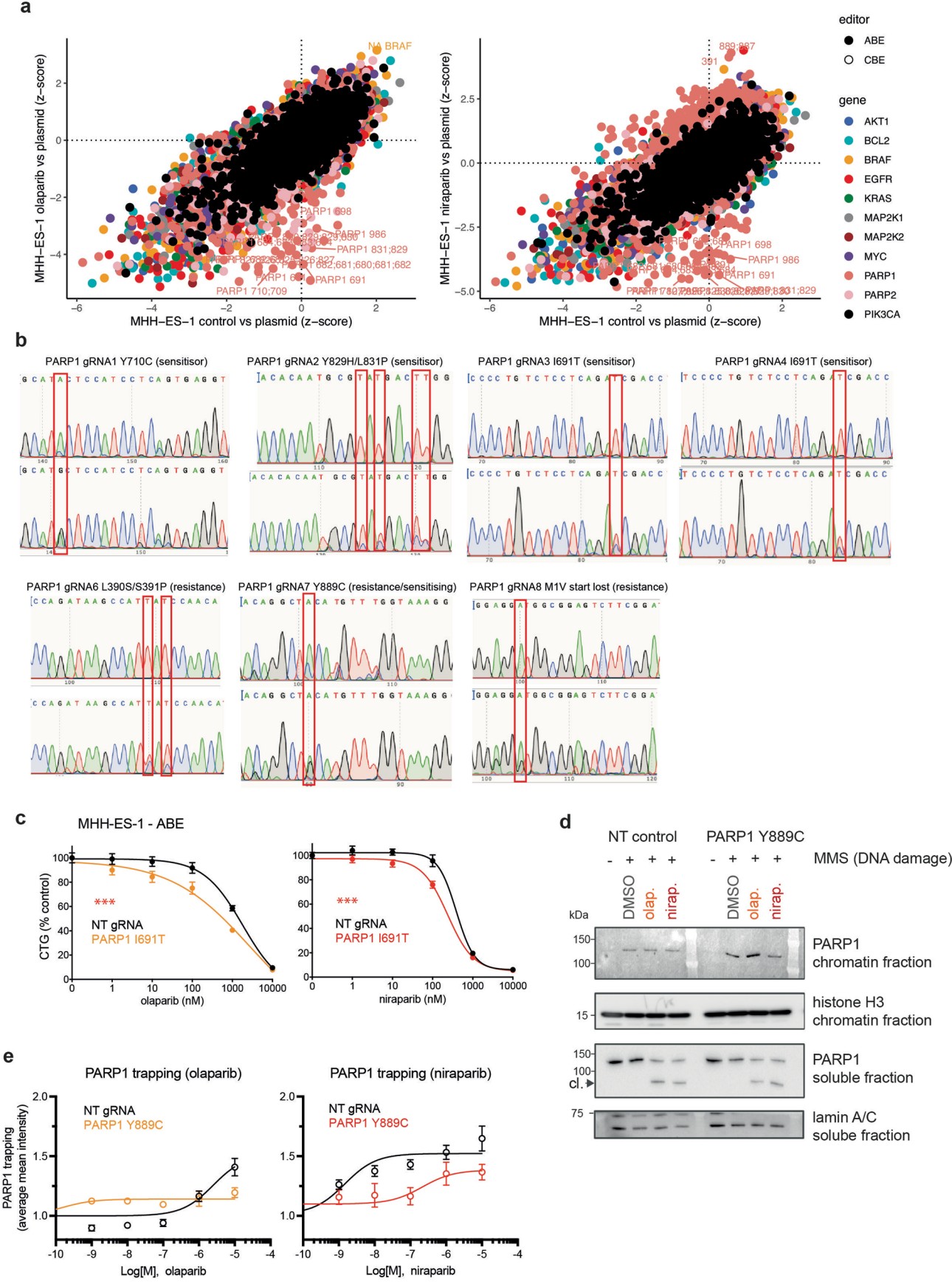

**Extended Data Fig. 4 | See next page for caption.**

**Extended Data Fig. 4 | Drug-sensitising variants. a)** Variants modulating sensitivity to PARP1/2 inhibitors olaparib or niraparib in MHH-ES-1 cells in CBE or ABE screens. Comparison of gRNA z-scores for the drug-treated arm vs plasmid library against the z-scores from the untreated control vs the plasmid library. Predicted edited amino acid positions are labelled. **b)** Sanger Sequencing of base edits in *PARP1* from validation experiments using individual gRNAs that caused drug resistance or sensitisation to PARP1/2 inhibitors. Data are representative of two independent experiments. **c)** Proliferation assays measuring drug response to olaparib and niraparib PARP inhibitors in MHH-ES-1 ABE cells harbouring the genotyped drug-sensitising variant, PARP1 I691T. Data represent the mean ± SD of two independent experiments performed on separate days, each in biological triplicate (CTG; CellTiter-Glo). 2-way ANOVA; ***p-value < 0.0001. **d)** Western blotting assessment of PARP trapping on DNA in MHH-ES-1 ABE cells harbouring

the PARP1 Y889C variant or a non-targeting (NT) control gRNA. Cells were treated with a DNA damaging agent (MMS, 0.01 %) and the PARP inhibitor with olaparib or niraparib (both at 3 µM) for 4 h before analysis. Nuclei were fractionated into a chromatin-bound and soluble fractions prior to immunoblotting. Cl. denotes cleaved PARP in response to DNA damage and PARP inhibition. Lamin A/C and histone H3 serve as loading controls for chromatin-bound and soluble fractions, respectively. **e)** Immunofluorescence microscopy assessment of PARP trapping on DNA in MHH-ES-1 ABE cells harbouring the PARP1 Y889C variant or a non-targeting (NT) control gRNA. Cells were treated with a DNA damaging agent (MMS, 0.01 %) and the PARP inhibitor with olaparib or niraparib (dose titration) for 4 h. PARP protein not bound to chromatin was removed before fixation and staining. Data represent the mean ± SD fluorescence nuclear intensity of PARP1 from biological triplicates.

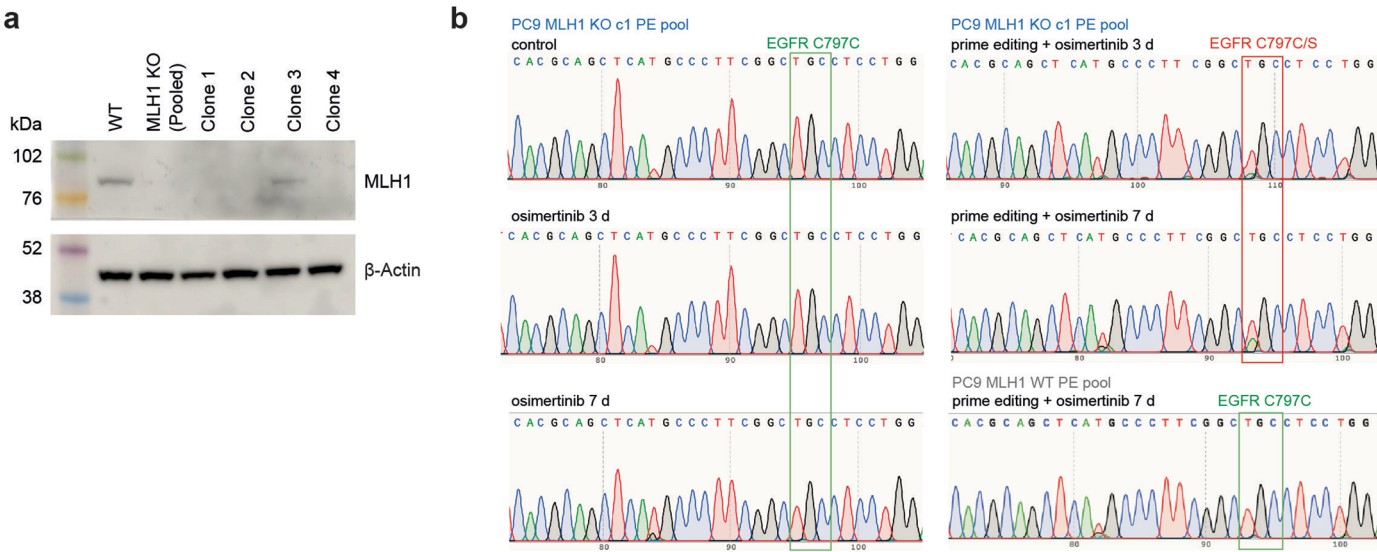

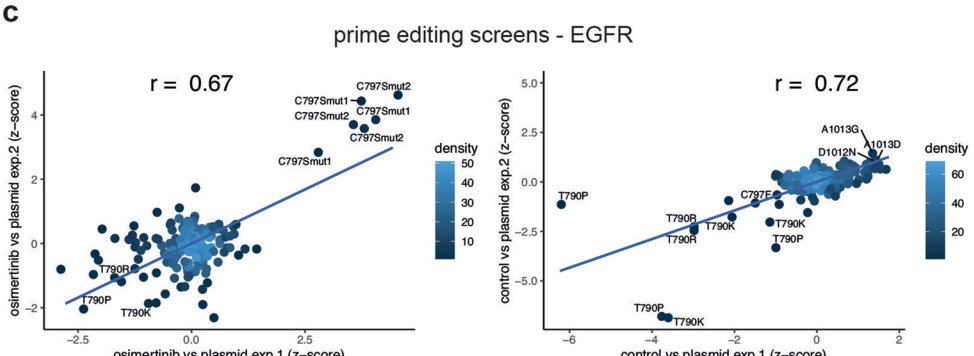

**Extended Data Fig. 5 | Prime editing screening of EGFR variants. a)** Western blot for MLH1 verifies KO of MLH1 in PC9 cells. PC9 cells were transfected with a Cas9-GFP plasmid encoding an *MLH1* targeting gRNA. FACS of GFP positive single cells gave clonal populations, or a pooled population ("pool"). Cells were expanded before analysis by Western blotting. Actin serves as a loading control. Data are representative of two independent experiments. **b)** Sanger sequencing of prime editing of EGFR C797S in PC9 cells. PC9-PE MLH1 KO (clone 1 from above), or MLH1 WT PC9-PE cells were infected with a pegRNA encoding the C797S edit, puromycin selected and prime editing was initiated with the addition of doxycycline for 5 days. Control (untreated) or osimertinib selected cells (5 nM) are shown. The EGFR C797 locus was PCR amplified and then analysed with Sanger sequencing. **c)** Replicate correlation between pegRNA z-scores from EGFR prime editing mutagenesis screens performed in PC9 MLH1 KO PE2 cells. Data are from two independent screens performed on different days. Labelled are predicted mutations in EGFR installed by the pegRNAs. Pearson correlation coefficient values (r) between independent replicate screens are shown. pegRNA, prime editing gRNA.

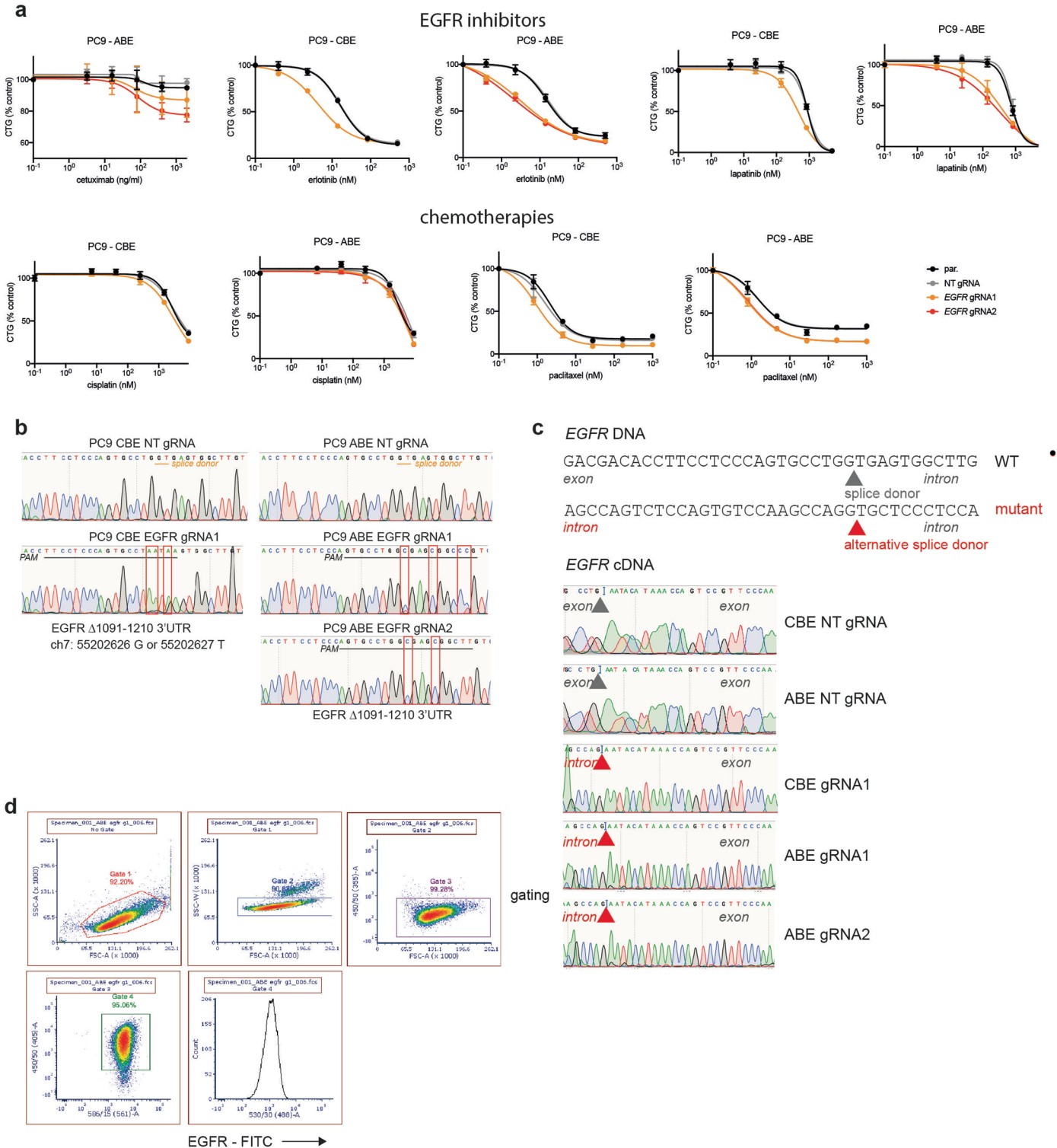

**Extended Data Fig. 6 | EGFR C-terminal truncating variants sensitise to EGFR inhibitors. a)** Drug titration experiments in PC9 CBE and ABE cells using Cell-titre Glo to measure cell proliferation in the presence of EGFR inhibitors (cetuximab, erlotinib, lapatinib), or chemotherapy agents (cisplatin, paclitaxel). Data represent the mean ± SD of two independent experiments performed on separate days, each in biological triplicate. **b)** Sanger sequencing of DNA from WT or base edited PC9 cells harbouring the EGFR-inhibitor sensitising splice variant. CBE editing and ABE editing of a known (GT) splice donor is shown.

The position of each gRNA is indicated. **c)** Sanger sequencing cDNA from WT or base edited PC9 cells harbouring the EGFR-inhibitor sensitising splice variant. WT cells display exon-exon splicing as expected, whereas mutant cells display intron retention by utilising an alternative splice donor in the downstream intron. **d)** Gating strategy for flow cytometry analysis of EGFR expression on PC9 cells (FITC). Gating was performed on cells, singlets, viable cells, BFP+ cells (gRNA expression).

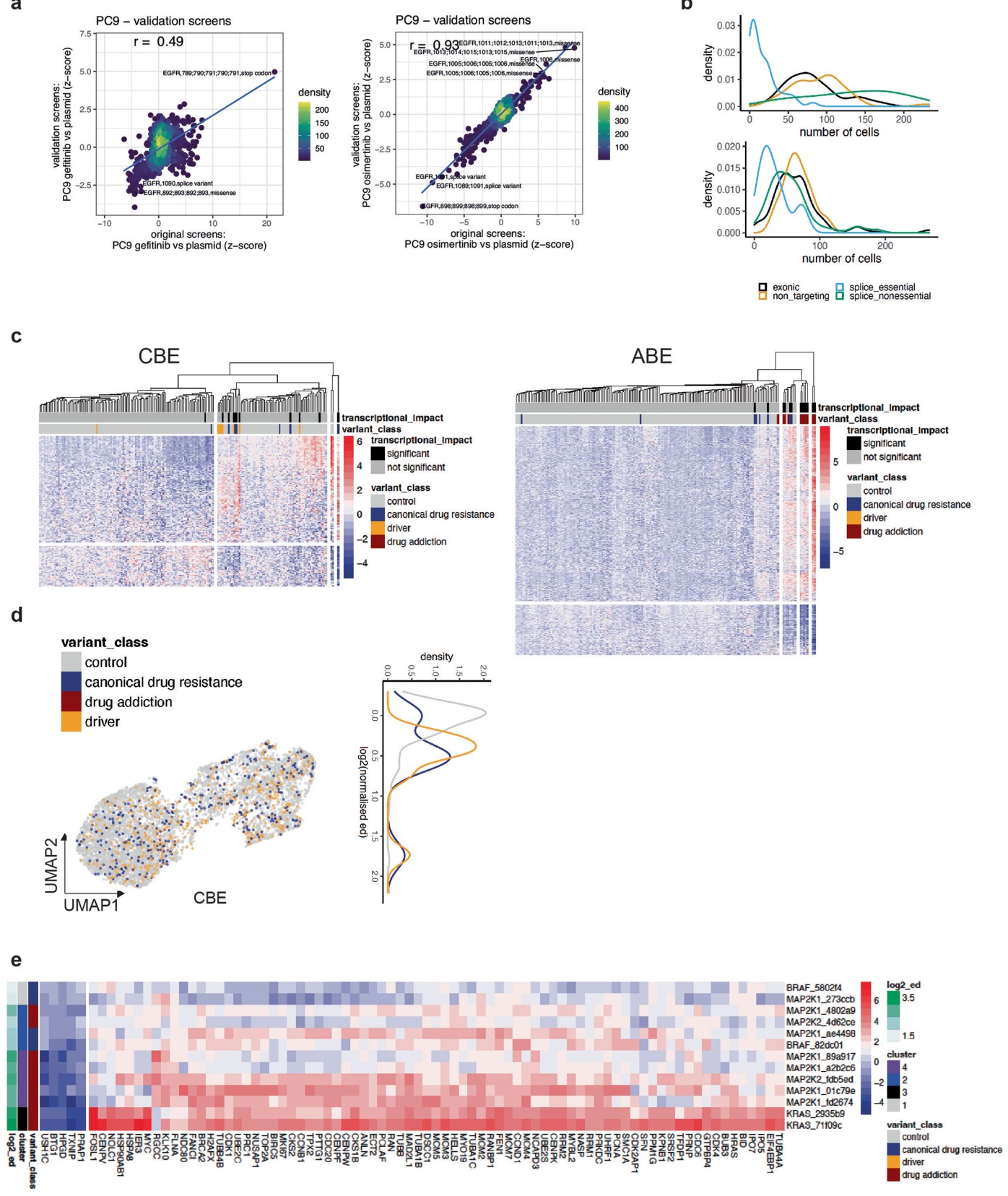

**Extended Data Fig. 7 | See next page for caption.**

**Extended Data Fig. 7 | Perturb-seq quality control and pathway analysis.**
**a)** Correlation between large-scale base editing screens (PC9 CBE and ABE) and a small-scale validation base editing screen designed for perturb-seq. Pearson correlation coefficients (r) are shown for gefitinib and osimertinib screens.
**b)** Density plot of gRNA classes against cell numbers in single-cell sequencing for HT-29 CBE and ABE experiments after quality control. Cells with gRNAs targeting splice sites in essential genes are depleted, indicating efficient editing.
**c)** Heatmap of scaled expression levels (mean=0, SD = 1, average across gRNA) of genes differentially expressed for at least one resistance gRNA with an absolute log2-fold change > 0.5 at FDR < 0.1, when comparing against cells with NT gRNAs

in HT-29 CBE or ABE perturb-seq screens. The dendrogram was cut at 4 clusters to show the varying gene expression levels and their association with variant class. **d)** UMAPs coloured by variant class and normalised energy distances (ed) between NT gRNA cells and drug resistant cells in CBE HT-29 cells treated with the combination of dabrafenib (80 nM) and cetuximab (1 μg/ml) for 16 h. **e)** Heatmap of scaled expression levels (mean=0, SD = 1, average across gRNA) of cell-cycle related genes (GO.0007049) that are differentially expressed for at least one resistance gRNA with absolute log2-fold change > 0.75 and FDR < 0.001 for the HT-29 ABE perturb-seq screen for at least one gRNA.

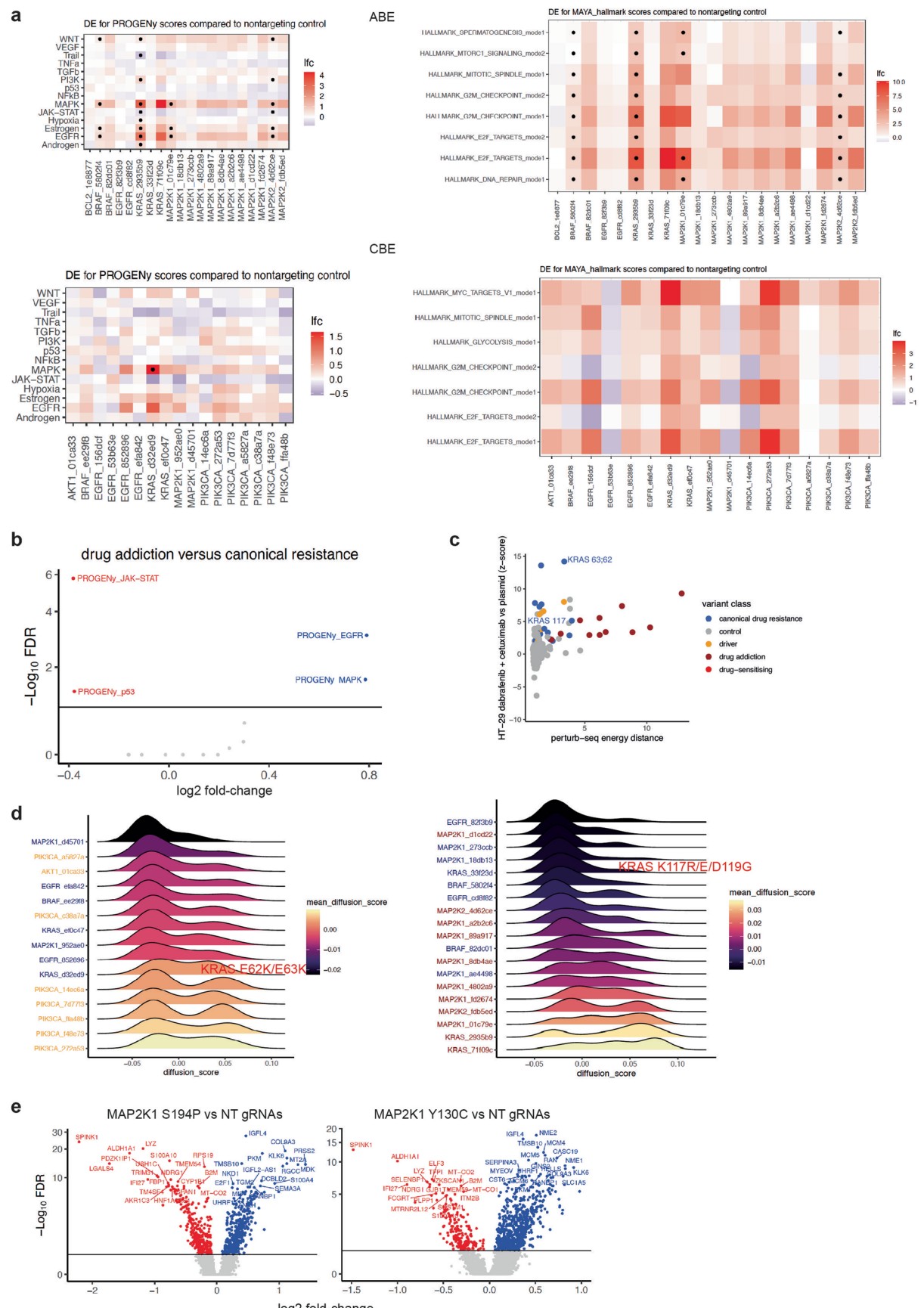

**Extended Data Fig. 8 | See next page for caption.**

**Extended Data Fig. 8 | Perturb-seq functionally defines drug resistant cell states. a)** Differential expression analysis of pathways from MAYA or PROGENy for HT-29 CBE and ABE perturb-seq screens. Heatmaps display log-fold changes for a given pathway-gRNA comparison, and statistical significance is denoted with a dot (significance at FDR < 0.1). **b)** Differential expression at the level of PROGENy pathway scores for drug addiction versus canonical drug resistance. For each gRNA the same number of iBARs was sampled to avoid biases resulting from an over-representation of individual gRNAs. **c)** Comparison of z-scores from proliferation read-out base editing screens to energy distance scores derived from perturb-seq screens. Variant classes based on the HT-29 proliferation screens in dabrafenib and cetuximab are indicated. Intermediate variants discussed in the text are labelled. **d)** Diffusion scores illustrate progressive levels of mutational impact for the CBE and ABE data set, with drug addiction variants having the highest scores and a range of different impact levels across the gRNAs conferring drug resistance. The intermediate variants KRAS E62K/E63K and KRAS K117R/E/D119G are highlighted. **e)** Volcano plots of significantly differentially expressed genes (vs NT control gRNA cells) from representative drug resistance gRNAs. *B2M* is downregulated by both variants. Significant down- and upregulation at FDR < 0.1 (Benjamini-Bogomolov correction) are indicated in blue and red respectively.

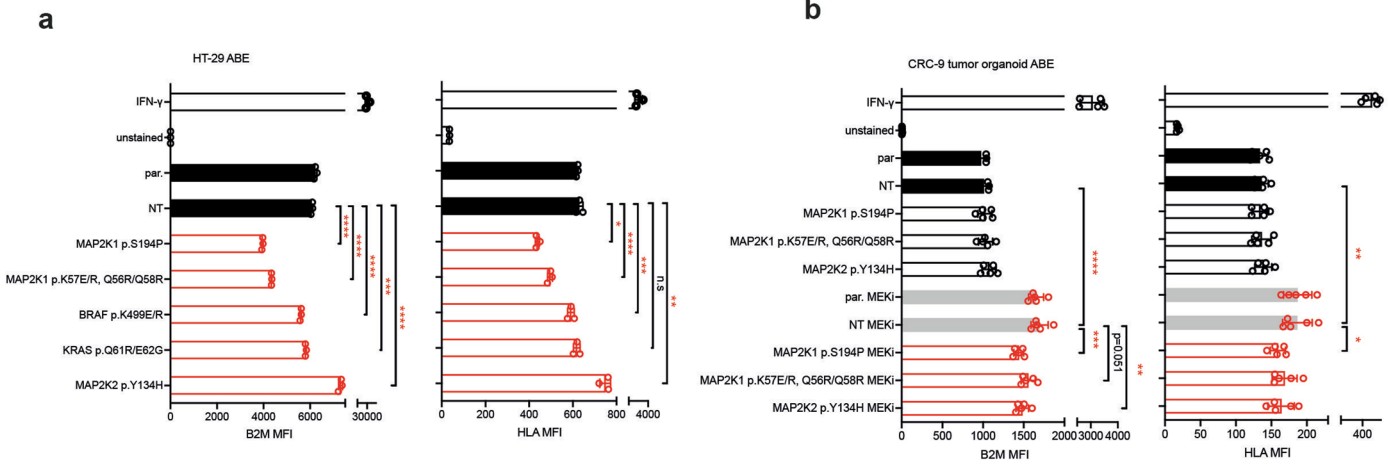

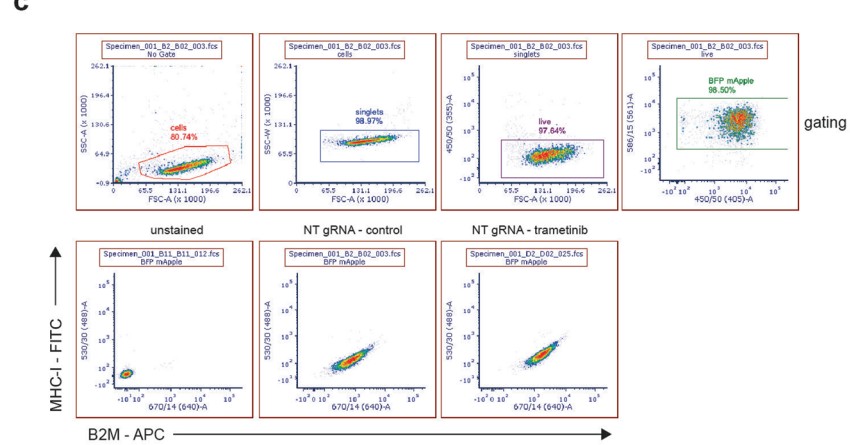

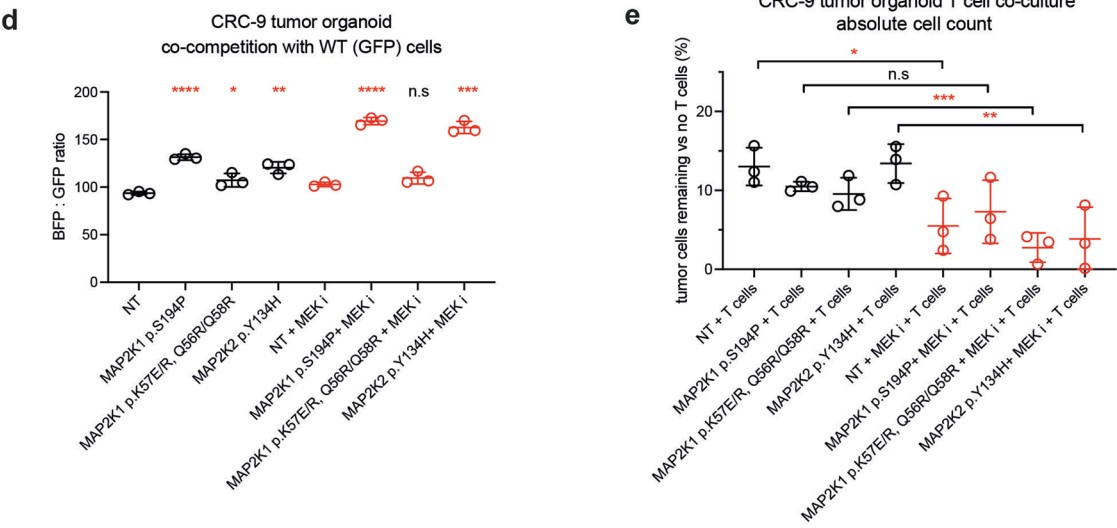

**Extended Data Fig. 9 | See next page for caption.**

**Extended Data Fig. 9 | Effect of MAPK signalling and drug addiction variants on antigen presentation and sensitivity to T cell killing. a)** Flow cytometry assessment of B2M and HLA-A,B,C expression in HT-29 ABE cells harbouring drug addiction variants. Data represent the mean ± SD of biological triplicates. IFN-gamma treatment serves as a positive control (48 h, 400 U/ml). ****P-value < 0.0001; ***P-value = 0.0003 (HLA) or 0.0005 (B2M); **P-value = 0.002; *P-value = 0.037; unpaired, two-tailed Student's t-test comparing to non-targeting gRNA (NT) condition. Genotyped variants are shown. **b)** Flow cytometry assessment of B2M and HLA-A,B,C expression in CRC-9 ABE tumour organoid cells harbouring drug addiction variants. Cells were treated with DMSO (control) or the MEK inhibitor trametinib (25 nM) for 48 h before analysis. Data represent the mean ± SD of two independent experiments. IFN-g treatment serves as a positive control (48 h, 400 U/ml). ****P-value < 0.0001; ***P-value = 0.0015; **P-value = 0.0066 (B2M or 0.0012 (HLA); *P-value = 0.031; unpaired, two-tailed Student's t-test comparing non-targeting gRNA (NT) condition. Genotyped variants are shown. **c)** Representative flow cytometry gating used for CRC-9 tumour organoids to assess HLA-A,B,C

and B2M cell surface protein expression. Single, live cells with mApple (ABE) and BFP (gRNA) expression were gated for analysis. **d)** Co-competition flow cytometry assays of WT (GFP – NT gRNA expressing cells) and drug resistant CRC-9 tumour organoids (BFP – gRNA expressing) at 72 h. Data represent the mean ± SD of biological triplicates. ****P-value < 0.0001; ***P-value = 0.0001; **P-value = 0.0018; *P-value = 0.031; unpaired, two-tailed Student's t-test comparing to non-targeting gRNA (NT) condition. Genotyped variants are shown. **e)** Co-culture assay of primary, autologous, anti-tumour T cells with CRC-9 tumour organoids harbouring different drug addiction variants. Cancer cells were pre-treated with the MEK inhibitor trametinib (25 nM) for 48 h before washing and plating the co-culture assay plate. Flow cytometry assessment of absolute cell numbers (measured by counting beads) following 72 h co-culture. Data are expressed as the percentage of live cells remaining as compared to the relevant condition in the absence of T cells and represent the mean ± SD of biological triplicates. ***P-value = 0.013, **P-value = 0.025, *P-value = 0.037; unpaired, two-tailed Student's t-test. Genotyped variants are shown.

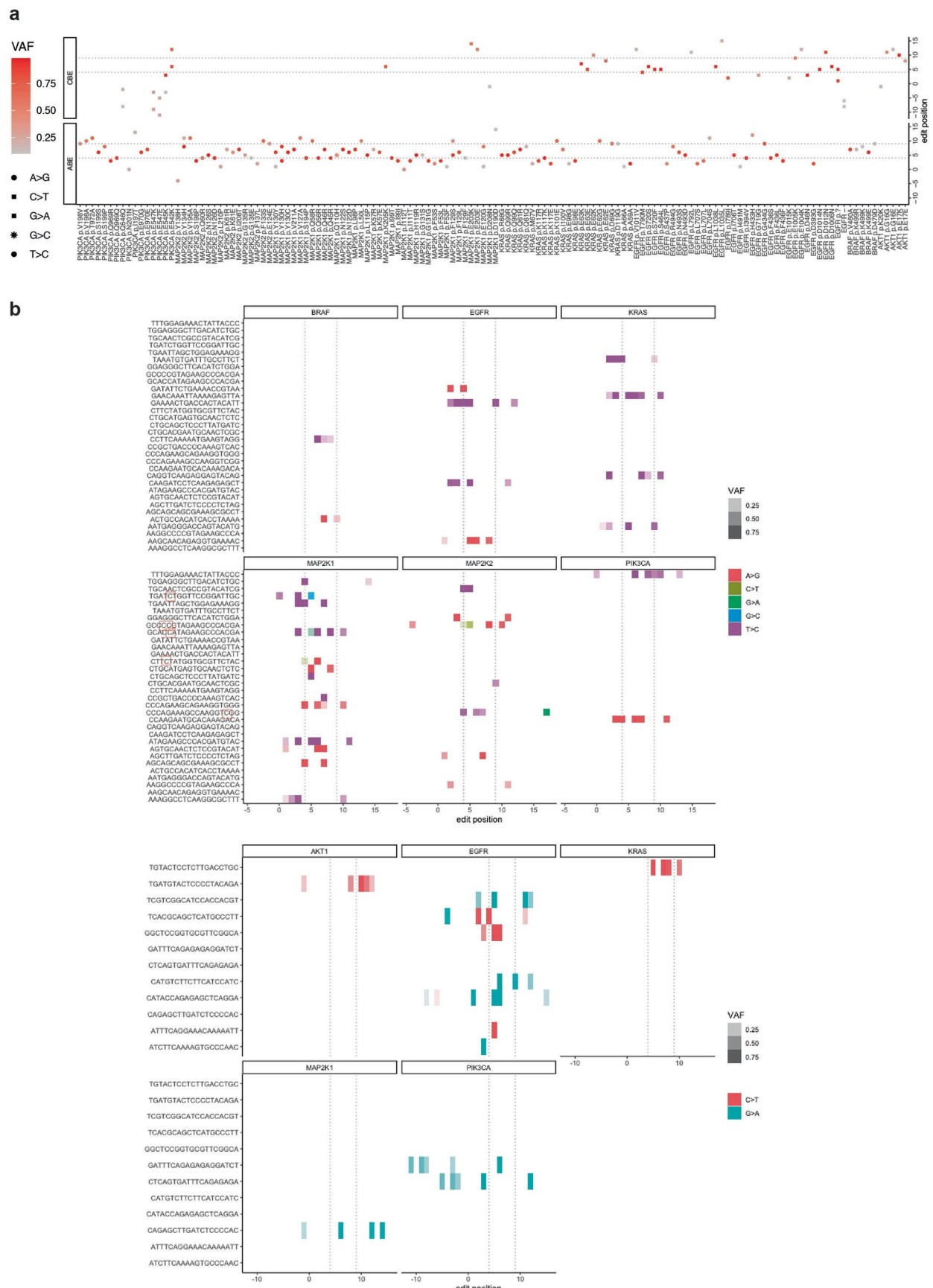

**Extended Data Fig. 10 | See next page for caption.**

**Extended Data Fig. 10 | Next-generation sequencing of base edits across 45 variants modulating drug sensitivity. a)** Base editing efficiency and precision mapped across 45 endogenous loci in HT-29 CBE and ABE cells. Average VAFs for exact edits for hit gRNAs are shown for each variant from amplicon sequencing data that were absent in unedited samples. Dashed lines represent the predicted base editing activity window. Data represent the mean of two independent experiments performed on separate days. VAF, variant allele frequency. **b)** Editing efficiency and precision of CBE and ABE base editors are shown by amplicon sequencing of endogenous DNA loci. Base editing was performed by doxycycline-induced expression of ABE (top panel) or CBE (bottom panel) for three days. Rare transversion mutations and their sequence context within the gRNA are highlighted by a red box. VAF, variant allele frequency from amplicon sequencing and represent the mean of two independent experiments.

# Reporting Summary

## Statistics

For all statistical analyses, confirm that the following items are present in the figure legend, table legend, main text, or Methods section.

| n/a | Confirmed | |
|---|---|---|
| ☐ | ☒ | The exact sample size (*n*) for each experimental group/condition, given as a discrete number and unit of measurement |
| ☐ | ☒ | A statement on whether measurements were taken from distinct samples or whether the same sample was measured repeatedly |
| ☐ | ☒ | The statistical test(s) used AND whether they are one- or two-sided<br>*Only common tests should be described solely by name; describe more complex techniques in the Methods section.* |
| ☐ | ☒ | A description of all covariates tested |
| ☐ | ☒ | A description of any assumptions or corrections, such as tests of normality and adjustment for multiple comparisons |
| ☐ | ☒ | A full description of the statistical parameters including central tendency (e.g. means) or other basic estimates (e.g. regression coefficient) AND variation (e.g. standard deviation) or associated estimates of uncertainty (e.g. confidence intervals) |
| ☐ | ☒ | For null hypothesis testing, the test statistic (e.g. *F*, *t*, *r*) with confidence intervals, effect sizes, degrees of freedom and *P* value noted<br>*Give P values as exact values whenever suitable.* |
| ☒ | ☐ | For Bayesian analysis, information on the choice of priors and Markov chain Monte Carlo settings |
| ☐ | ☒ | For hierarchical and complex designs, identification of the appropriate level for tests and full reporting of outcomes |
| ☐ | ☒ | Estimates of effect sizes (e.g. Cohen's *d*, Pearson's *r*), indicating how they were calculated |

*Our web collection on statistics for biologists contains articles on many of the points above.*

## Software and code

Policy information about availability of computer code

| Data collection | For flow cytometry, we used FACSDiva software (version 9; BD Biosciences). |
|---|---|
| Data analysis | Code used to analyse base editing screens can be found on GitHub here: https://github.com/MatthewACoelho/Res1_analysis. Code used to analyse the single-cell screens can be found here: https://github.com/MarioniLab/BE_perturb_seq_drug_resistance.<br>In amplicon sequencing analysis we used BCFtools (version 1.20) and vaf correct (version 5.4.0).<br>In perturb-seq analysis we used scater (version 1.20.1) and Seurat (version 4.0.6) , scuttle (version 1.2.1) and mclust R (version 6.1.1).<br>For visualization of crystal structures, we used PyMOL (version 2.4.1), for graphs we used GraphPad Prism (version 8) or R ggplot2 (version 3.3.0).<br>For flow cytometry analysis we used FlowJo (version 10) or FCS Express (version 7). |

For manuscripts utilizing custom algorithms or software that are central to the research but not yet described in published literature, software must be made available to editors and reviewers. We strongly encourage code deposition in a community repository (e.g. GitHub). See the Nature Portfolio guidelines for submitting code & software for further information.

## Data

Policy information about availability of data

All manuscripts must include a data availability statement. This statement should provide the following information, where applicable:

- Accession codes, unique identifiers, or web links for publicly available datasets
- A description of any restrictions on data availability
- For clinical datasets or third party data, please ensure that the statement adheres to our policy

SSequencing data are deposited on ENA and EGA and accessions are described in Supplementary Table 8 (ERP146490, ERP148732, ERP144241, ERP141719, ERP156437, EGAS00001006683, EGAS00001006170, EGAS00001006169, EGAS00001006093, EGAS00001006092, EGAS00001006091) and can be found here: https://www.ebi.ac.uk/ena/browser/home and here, https://ega-archive.org/studies/. All genomic indexing is relative to GRCh38 genome assembly https://www.ensembl.org/Homo_sapiens/Info/Index. COSMIC variants were downloaded in February 2024 version 99 https://cancer.sanger.ac.uk/cosmic/download/cosmic. Unprocessed Western blots are available as Source Data and screening data are available in Supplementary Table 2. Screen z-scores are available on the MAVE database103 (urn:mavedb:00001204).

## Research involving human participants, their data, or biological material

Policy information about studies with human participants or human data. See also policy information about sex, gender (identity/presentation), and sexual orientation and race, ethnicity and racism.

| | |
|---|---|
| Reporting on sex and gender | N/A |
| Reporting on race, ethnicity, or other socially relevant groupings | N/A |
| Population characteristics | N/A |
| Recruitment | N/A |
| Ethics oversight | N/A |

Note that full information on the approval of the study protocol must also be provided in the manuscript.

# Field-specific reporting

Please select the one below that is the best fit for your research. If you are not sure, read the appropriate sections before making your selection.

☒ Life sciences　　　☐ Behavioural & social sciences　　　☐ Ecological, evolutionary & environmental sciences

For a reference copy of the document with all sections, see nature.com/documents/nr-reporting-summary-flat.pdf

# Life sciences study design

All studies must disclose on these points even when the disclosure is negative.

| | |
|---|---|
| Sample size | No statistical methods were used for sample size determination. Base editing screens gave significant effect sizes for gRNAs that were independently validated as functional. We analysed the effects of all possible variants that could be installed with CBE and ABE in the 11 cancer genes studied, which constituted a library of 22,816 gRNAs. |
| Data exclusions | We excluded gRNAs that had more than two perfect matches in the GRCh38 human genome (n=134), and gRNAs with < 100 read counts in the plasmid or any time 0 sample in the screens (n=27). For the few hit gRNAs with two perfect matches in the genome, we confirmed there was only one exonic target. We also excluded gRNAs specifically in the MHH-ES-1 screen that had a 10-fold read count difference between replicates (n=118) from downstream analysis of the MHH-ES-1 screens. |
| Replication | All experiments were performed independently on a separate day at least twice, including CRISPR and base editing screens, as stated in the figure legends. All attempts to repeat experiments were successful. |
| Randomization | No randomisation was performed. Base editing screens are unbiased, pooled experiments where sample preparation is identical. For other experiments, randomisation was not performed as all cell culture experiments were performed under identical conditions apart from the experimental perturbation. |
| Blinding | No blinding was performed. Base editing screens are unbiased, pooled experiments where sample preparation is identical. The investigators where unbiased towards the screening data analysis, as this was a hypothesis-generating experiment without prior assumptions of variant effect. For other experiments, blinding was not performed as all cell culture experiments were performed under identical conditions apart from the experimental perturbation. |

# Reporting for specific materials, systems and methods

We require information from authors about some types of materials, experimental systems and methods used in many studies. Here, indicate whether each material, system or method listed is relevant to your study. If you are not sure if a list item applies to your research, read the appropriate section before selecting a response.

## Materials & experimental systems

| n/a | Involved in the study |
|-----|----------------------|
| ☐ | ☒ Antibodies |
| ☐ | ☒ Eukaryotic cell lines |
| ☒ | ☐ Palaeontology and archaeology |
| ☒ | ☐ Animals and other organisms |
| ☒ | ☐ Clinical data |
| ☒ | ☐ Dual use research of concern |
| ☒ | ☐ Plants |

## Methods

| n/a | Involved in the study |
|-----|----------------------|
| ☒ | ☐ ChIP-seq |
| ☐ | ☒ Flow cytometry |
| ☒ | ☐ MRI-based neuroimaging |

## Antibodies

| | |
|---|---|
| Antibodies used | Western blotting primary antibodies: EGFR total (1068 epitope, #2232, 1:1,000 dilution), p-EGFR (1148 region, #4404, 1:1,000 dilution), b-actin (#4970, 1:1,000 dilution), p-ERK (#9101, 1:1,000 dilution), ERK total (#9102, 1:1,000 dilution) (Cell Signaling Technology), EGFR epitope 1020-1046 (#610017 BD Biosciences, 1:1,000 dilution), PARP1 (#9532, clone 46D11, 1:1,000 dilution), lamin A/C (#2032, 1:1,000 dilution), histone H3 (#3638, clone 96C10, 1:1,000 dilution) (all from Cell Signaling Technology). <br><br> Secondary antibodies (anti-mouse and ant-rabbit) were conjugated to horseradish peroxidase (#31460 and #31430, Thermo Fisher Scientific, 1:5,000 dilution). <br><br> Flow cytometry: anti-EGFR-FITC antibody (#MA5-28104, clone ICR10, Thermo Fisher Scientific, 1:100 dilution). |
| Validation | No additional validation was performed for these commercially available antibodies but they have been validated by the vendors. Datasheets including validation, citations and application notes can be found here: <br> https://www.cellsignal.com <br> https://www.thermofisher.com/uk/en/home/life-science/antibodies/primary-antibodies.html?icid=ab-search-primary-icons |

## Eukaryotic cell lines

Policy information about cell lines and Sex and Gender in Research

| | |
|---|---|
| Cell line source(s) | H23 (NCI), PC9 (RIKEN), HT-29 (NCI), MHH-ES-1 (DSMZ), HEK293T (ATCC) |
| Authentication | All cell models used in this study (H23, PC9, HT-29, MHH-ES-1, HEK293T) were STR profiled in accordance with authentication guidelines. |
| Mycoplasma contamination | All cell models used in this study were routinely verified as mycoplasma-free. |
| Commonly misidentified lines (See ICLAC register) | None used in this study. |

## Plants

| | |
|---|---|
| Seed stocks | *Report on the source of all seed stocks or other plant material used. If applicable, state the seed stock centre and catalogue number. If plant specimens were collected from the field, describe the collection location, date and sampling procedures.* |
| Novel plant genotypes | *Describe the methods by which all novel plant genotypes were produced. This includes those generated by transgenic approaches, gene editing, chemical/radiation-based mutagenesis and hybridization. For transgenic lines, describe the transformation method, the number of independent lines analyzed and the generation upon which experiments were performed. For gene-edited lines, describe the editor used, the endogenous sequence targeted for editing, the targeting guide RNA sequence (if applicable) and how the editor was applied.* |
| Authentication | *Describe any authentication procedures for each seed stock used or novel genotype generated. Describe any experiments used to assess the effect of a mutation and, where applicable, how potential secondary effects (e.g. second site T-DNA insertions, mosiacism, off-target gene editing) were examined.* |

# Flow Cytometry

## Plots

Confirm that:

☒ The axis labels state the marker and fluorochrome used (e.g. CD4-FITC).

☒ The axis scales are clearly visible. Include numbers along axes only for bottom left plot of group (a 'group' is an analysis of identical markers).

☒ All plots are contour plots with outliers or pseudocolor plots.

☒ A numerical value for number of cells or percentage (with statistics) is provided.

## Methodology

| | |
|---|---|
| Sample preparation | PC9 cells were harvested by trypsinisation and washed in FACS buffer (0.5 % FCS< 2 mM EDTA in PBS) before staining with anti-EGFR-FITC antibody (#MA5-28104, Thermo Fisher Scientific) for 25 min on ice in the dark. Cells were washed twice in FACS buffer, incubated with DAPI (1 µg/ml, Thermo Fisher Scientific) before filtering through a nylon mesh cell strainer, and analysis on an LSRFortsessa (BD Biosciences). |
| Instrument | LSRFortessa; BD Biosciences |
| Software | Flow cytometry data were acquired using FACSDiva software (version 9, BD Biosciences), and analysed using FlowJo (version 10) or FCS Express (version 7). |
| Cell population abundance | No verification of post-sort abundance was performed. |
| Gating strategy | Gating of PC9 cells was based on FSC-A vs SSC-A (cells), FSC-A vs SSC-W (singlets), DAPI vs FSC-A (viable), mApple vs BFP (base editor construct and gRNA construct). A histogram of EGFR-FITC expression in this cell population is shown in Figure 6d. Similarly, Supplementary 9c shows the gating strategy for CRC9 cells stained for MHC-I and B2M. |

☒ Tick this box to confirm that a figure exemplifying the gating strategy is provided in the Supplementary Information.

