## [Peer Review File · Nature Genetics]

Peer Review Information

Manuscript Title: Base editing screens define the genetic landscape of cancer drug resistance mechanisms

Corresponding author name(s): Dr Mathew (J) Garnett, Dr Matthew (A) Coelho

Reviewer Comments & Decisions:

Decision Letter, initial version:

5th Feb 2024

Dear Dr Garnett,

First, please allow me to apologise for the delay in returning this decision to you. Thank you for your patience.

Your Article entitled "Genetic landscape of cancer drug resistance mechanisms from base editing screens" has now been seen by 3 referees, whose comments are attached. While they find your work of potential interest, they have raised serious concerns which in our view are sufficiently important that they preclude publication of the work in Nature Genetics, at least in its present form.

While the referees find your work of some interest, they raise concerns about the strength of the novel conclusions that can be drawn at this stage.

Should further experimental data allow you to fully address these criticisms we would be willing to consider an appeal of our decision (unless, of course, something similar has by then been accepted at Nature Genetics or appeared elsewhere). This includes submission or publication of a portion of this work someplace else.

The required new experiments and data include, but are not limited to those detailed here. We hope you understand that until we have read the revised manuscript in its entirety we cannot promise that it will be sent back for peer review.

If you are interested in attempting to revise this manuscript for submission to Nature Genetics in the future, please contact me to discuss a potential appeal. Otherwise, we hope that you find our referees' comments helpful when preparing your manuscript for resubmission elsewhere.

Although we cannot publish your paper, it may be appropriate for another journal in the Nature

Portfolio. If you wish to explore the journals and transfer your manuscript please use our manuscript transfer portal. You will not have to re-supply manuscript metadata and files, unless you wish to make modifications. For more information, please see our manuscript transfer FAQ page.

Sincerely,

Safia Danovi
Editor
Nature Genetics

Referee expertise:

Referee #1: gene editing screens, cancer

Referee #2: base editing

Referee #3: functional genomics, cancer

Reviewers' Comments:

Reviewer #1:

Remarks to the Author:

Here Coelho et al. have performed CRISPR base editing mutagenesis screens targeting 11 cancer genes in 4 cancer cells looking at the genetic resistance mechanisms to 10 oncology drugs. They have identified four classes of protein variants modulating drug sensitivity, including variants that are: drug addiction variants, canonical drug resistance variants, driver variants and drug sensitizing variants. Furthermore they have used single cell transcriptomics to investigate how drug resistant variants operate. Ultimately they propose using CRISPR base editing at scale to generate variant function maps to help with therapy decisions including patient stratification, drug combinations and scheduling of drug treatments.

However, base editor screens have been done previously multi times (see Xu 2021 NBT; Lue 2023 Cell, Martin-Rufino 2023 Cell; among others), thus the technique is not new.

Biologically, the cancer variant found from this screen here and the validations are rudimentary.

Data-wise, the study fell far short of major advancement and current data did not fully support the novelty or insight claims at NG level.

Main criticisms:

The main criticism is that although the authors present several novel variants based on the screening analysis, there is insufficient subsequent experiments performed to show that these variants have functional significance in terms of cancer cell survival.

Furthermore the authors present a number of hypotheses on how particular variants could potentially affect survival through downstream pathways but there are no mechanistic studies presented. The

authors propose that variant mapping can be used to guide treatment but there is no in-vivo evidence that tumors harboring particular variants respond as would be expected based on the results of the screen and perturb-seq. This severely limits the significance of their findings. However, there is no mechanistic and in-vivo studies for a particular set of variants to show that the predictions based on the screen and perturb-seq have functional significance and can provide information useful for treatment decisions.

- The authors describe when the screen identifies previously known resistant variants. Can they also put a description for each gene of previously known resistant variants that were missed. This could be presented as a supplementary table. As the authors mention base editing screens are not saturating and cannot install all possible variants, but it would be helpful to know the sensitivity of these screens in identifying functionally significant variants.
- Could the authors present the known and unknown variants identified for each gene in a supplementary table so that is easy to see and review.
- The authors use perturb-seq to present potential hypotheses of potential mechanisms of action for these resistant variants and explain the difference in behavior between drug addition and canonical drug resistance variants. They attributed this to different levels of oncogenic signaling. Can the authors verify this using protein analysis? Can they also perform further mechanistic studies to verify their hypotheses based on the perturb-seq data.
- The authors hypothesize that reduction in JAK-STAT pathway activity and reduced expression of antigen presentation machinery induced by drug resistance variants may lead to immune evasion. Can the authors verify reduced antigen presentation by flow cytometry in cells with these variants? Can the authors also perform in-vitro and in-vivo studies showing reduced immune cell killing of the cancer cells and enhanced cell growth.
- The authors present a drug resistant variant map that they propose can guide treatment. Their argument would be much stronger if they could provide in-vivo studies showing that tumors harboring the particular variants respond as would be expected based on this map and that the strategies they propose (for example drug holidays) enhance survival.

Other:

1. Figure 3a-b, is there a way to quantify the trend instead of highlighting selected points? I would suggest adding structure annotation/needle-plot at the x-axis if applicable in 3a.

2. Fig. 6f Why is the author trying to use AUROC as the x-axis instead of log fold change? ROC analysis will be biased if there are many more negative points than positive ones, this may be the case for PI3K drive variant cells, then a PRAUC curve is recommended.

3. Refer to Fig. 6g, please add more details on calculating the PFS score, "the metric based on Spearman rank correlation of differentially expressed genes in drug-resistant cells from perturb-seq data" (vs non-perturb?), and patients with PFS <6 months vs > 6 months? And can I interpret as canonical/drive variant will lead to long/favorable PFS? Spearman correlation here might not be sensitive to reflect the global gene alteration similarity, SubMap like algorithm would be better here. For calculating PFS score using Tian study [ref23], the author use all (Pre + On treatment)? The pre only or on only can be processed separately.

Reviewer #2:

Remarks to the Author:

In this study, Coelho et al. employed CRISPR base editing mutagenesis screens in four cancer cell lines to prospectively identify genetic resistance mechanisms to 10 oncology drugs. Through these screens the authors identified four classes of protein variants influencing drug sensitivity, offering insights into potential strategies for overcoming drug resistance. The authors coupled these approaches with single-cell transcriptomics to reveal the effects of protein variants on gene expression.

Overall, the findings of the study are well presented and provide a valuable map of genetic drug resistance mechanisms with implications for patient stratification and optimizing cancer treatment approaches. We recommend the authors to address the points below to strengthen their observations.

- 1) It would be important to provide additional experimental evidence on the mechanisms of action of the different classes of variants identified. For example, how do drug addiction mutants in MAPK signaling factors cause detrimental effects in the absence of treatment? What are the mechanisms by which PARP1 variants sensitize to PARP inhibitors?
- 2) Figure 6. The authors show that sc-SNV-seq can be employed to examine the transcriptional changes induced by variants that cause drug resistance. Can the authors use the same approach to investigate the effects induced by variants that cause drug sensitivity? Would this approach be informative for understanding the mechanism of action of these variants?
- 3) Figure 6d. The authors identify drug resistance variants with immune evasion signatures. Do these variants reduce the ability of immune cells to kill cancer cells in co-culture systems?
- 4) Figure 7b. The genotype of several of the variants introduced is complex. To define which amino acid changes are responsible for the phenotype observed, the authors should use other genome editing approaches, such as prime editing or homology-directed repair, for introducing precise single amino acid changes and studying their effects.
- 5) It would be helpful to highlight all variants identified through the screens that have also been associated to altered drug response in cancer patient datasets.

Reviewer #3:

Remarks to the Author:

Coelho et al. used base-editing screens with or without drugs to identify drug resistance and sensitization variants across 11 cancer genes. In total, they identified 45 genotyped variants that confer resistance and sensitivity. The authors validate a subset of variants in different types of follow-up experiments but also perform a base-editing perturb-seq screen with a focused library to identify signatures and measure the transcriptional impact of each variant. As the authors mentioned, this is one of the largest base-editing screens dataset and identifies many valuable drug-resistant and sensitizing variants. The authors provide important ideas about how this information can be used to identify personalized therapeutic strategies.

Main point

The manuscript's main concern is that the screens are very noisy, and the authors don't provide sufficient information on the false discovery rate of the hits. The quality of the screens differs some screens have better signal, and there are clearly some outlier hits. The metrics they used to call hits are similar across the screens, independently of their noise. The authors must provide additional quality control metrics typically used on these types of screens beyond the experiment correlation. They must also include more information about how the controls score compared to experimental sgRNAs and statistical or false discovery rate estimates.

The authors focus in the sections on highlighting specific examples, which is valuable but also need to provide clear quantitative information on how many hits are scoring in the experimental vs. control sgRNAs, which would provide information on the false discovery rate. It would be helpful if they could start the section by providing this information before moving to describe specific hits.

The authors use perturb-seq to assess the level of transcriptional signal as a way to validate the resistant variants, which seems to be a nice way to validate at scale. They should provide more information on how many of the expected variants tested elicited a transcriptional response, even if they provide a ranking in the supplemental table and figures of the patterns.

In Figure 7A, the authors provide a nice summary illustration of their top hits, which are also genotyped. Since this is their final hit list, can they more precisely describe how they selected the top-scoring?

Minor points

The classification of the functional variants is important. However, structuring the section by functional variants makes it hard to follow. The authors might want to consider separating the sections by drug classes (e.g. Osimertinib, Gefinitif) or make the sections more consistent to help the reader follow the results.

The authors should consider better visualizing the data for the proliferation base-editing screens. For example, figure 1b and supplemental figure 2a could be shown as distributions and figure 2. In Figure 2a they might consider not plotting all the genes in the same plot. The results of the perturb-seq screens have clear visualizations.

The authors use MYC throughout the paper as an essential gene control, but they have many more sgRNA against essential genes that they could use in aggregation. Is there a reason to use only MYC?

In supplemental fig 5a, the authors use PC9 with gefinitif and osimerib to validate the focused library and HT29 with the dafratinib and cetuximax to do the perturb-seq profiling. Can the authors make this clear in the main text?

When the authors describe the validation experiments, it would be important to describe how many are identified and how many are validated. For example, in the second paragraph of page 10, they

could write that among the X identified, we validated Y. They might consider a summary table that describes which variants were validated.

In the Drug-sensitizing variants section, the authors start by describing the resistance, not the sensitizing hits, which makes it confusing to the reader. They could consider an introductory sentence to include this information in this section.

Decision Letter, Appeal:

13th Jun 2024

Dear Dr Garnett,

Thank you for asking us to reconsider our decision on your manuscript "Genetic landscape of cancer drug resistance mechanisms from base editing screens". I have now discussed your appeal with my colleagues, and we think that you have some valid points. We therefore invite you to revise your manuscript along the lines that you propose.

When preparing a revision, please ensure that it fully complies with our editorial requirements for format and style; details can be found in the Guide to Authors on our website (<http://www.nature.com/ng/>).

Please be sure that your manuscript is accompanied by a separate letter detailing the changes you have made and your response to the points raised. At this stage we will need you to upload:

1) a copy of the manuscript in MS Word .docx format.

2) The Editorial Policy Checklist:

<https://www.nature.com/documents/nr-editorial-policy-checklist.pdf>

3) The Reporting Summary:

(Here you can read about the role of the Reporting Summary in reproducible science:

<https://www.nature.com/news/announcement-towards-greater-reproducibility-for-life-sciences-research-in-nature-1.22062>)

Please use the link below to be taken directly to the site and view and revise your manuscript:

[redacted]

With kind wishes,

Safia Danovi, PhD

Senior Editor, Nature Genetics

ORCID: 0009-0007-7822-5479

Author Rebuttal to Initial comments

Point-by-point response to peer review

Reviewers' Comments:

Reviewer #1:

Remarks to the Author:

Here Coelho et al. have performed CRISPR base editing mutagenesis screens targeting 11 cancer genes in 4 cancer cells looking at the genetic resistance mechanisms to 10 oncology drugs. They have identified four classes of protein variants modulating drug sensitivity, including variants that are: drug addiction variants, canonical drug resistance variants, driver variants and drug sensitizing variants. Furthermore they have used single cell transcriptomics to investigate how drug resistant variants operate. Ultimately they propose using CRISPR base editing at scale to generate variant function maps to help with therapy decisions including patient stratification, drug combinations and scheduling of drug treatments.

We thank the reviewer for their thoughtful and helpful comments on our manuscript.

However, base editor screens have been done previously multi times (see Xu 2021 NBT; Lue 2023 Cell, Martin-Rufino 2023 Cell; among others), thus the technique is not new.

Thank you for your comment. We appreciate that base editing screens have been reported before. The aim of our study is to apply this approach to understand mechanisms of drug resistance in cancer at scale, using state-of-the-art CBE and ABE base editors with flexible PAM recognition (NGN). To our knowledge, this is the most comprehensive functional analysis of drug resistance variants to date. We have cited the additional important reports of base editing screens mentioned in the revised text to better place our findings in this context.

Biologically, the cancer variant found from this screen here and the validations are rudimentary.

Due to the scale of our study, we are unable to deeply characterise all drug resistant variants on an individual basis. Nonetheless, in our revision, we have added extensive validation experiments, including prime editing of variants, functional characterisation of drug addiction phenotypes, antigen presentation, and investigation of PARP inhibitor sensitising variants (please refer to later sections of this document for more detail). In addition, we performed in-depth mechanistic studies on EGFR drug-sensitising variants (revised Fig. 6). We used perturb-seq to validate hundreds of variants and provide further mechanistic insights into the cancer cell signalling underpinning drug resistance (revised Fig. 7).

Data-wise, the study fell far short of major advancement and current data did not fully support the novelty or insight claims at NG level.

We feel that our functional characterisation of drug resistance variants supported by additional validation in our revision has significantly strengthened our manuscript. Through systematically highlighting new drug resistant variants and their mechanism of action, potential second-line therapies, and biomarkers of drug sensitivity, we have generated a resource that we feel will be broadly useful.

Main criticisms:

The main criticism is that although the authors present several novel variants based on the screening analysis, there is insufficient subsequent experiments performed to show that these variants have functional significance in terms of cancer cell survival.

Furthermore the authors present a number of hypotheses on how particular variants could potentially affect survival through downstream pathways but there are no mechanistic studies presented. The authors propose that variant mapping can be used to guide treatment but there is no in-vivo evidence that tumors harboring particular variants respond as would be expected based on the results of the screen and perturb-seq. This severely limits the significance of their findings. However, there is no mechanistic and in-vivo studies for a particular set of variants to show that the predictions based on the screen and perturb-seq have functional significance and can provide information useful for treatment decisions. Thank you for your comment. Recovery of known drug resistance and known cancer driver mutations in our base editing screens demonstrates the physiological relevance of our data. We discuss the overlap with clinical data further below. In addition, we highlight the specific additional validation experiments we have performed to address these concerns in our responses below.

- The authors describe when the screen identifies previously known resistant variants. Can they also put a description for each gene of previously known resistant variants that were missed. This could be presented as a supplementary table. As the authors mention base editing screens are not saturating and cannot install all possible variants, but it would be helpful to know the sensitivity of these screens in identifying functionally significant variants.

Thank you for this important suggestion. To our knowledge, there are limited databases of drug resistance mutations in cancer and this was, in part, a motivation for this study. One of the most comprehensive efforts to compile curated clinical drug resistance mutations is from COSMIC. Therefore, we have included COSMIC drug resistance variants and variants from selected cited articles relating to drugs that we screen in this study (e.g. Awad et al, *NEJM*, 2021, and Brammell et al, *Genome Res*, 2017) in revised Supplemental Table 5. We highlight variants that were recovered in our base editing screens and those that were missed due to the editing capabilities of cytidine and adenine base editors. Drug resistance variants that we detect in this study and do not have previous drug resistance annotations can be found in revised Supplemental Table 6. For our validated, genotyped variants, this information is also labelled in revised Figure 8. Furthermore, we include the following text in the results section of the manuscript to provide more context for our findings:

“To benchmark our resistance variant map, we surveyed COSMIC-curated drug response data⁴⁶ and literature relating to clinical incidence of resistance to the drugs analysed in this study^{42,50}. This identified 88 amino acid positions in the screened target proteins associated with drug response (Supplementary Table 5, Methods). 85 of these had at least one gRNA predicted to target the amino acid position, 30 of which (35.29 %) had a concordant drug resistance phenotype in our screening dataset specific to the reported drug and gene target. In addition, we observed 252 edits at amino acid positions that had not been previously associated with altered drug response (e.g. PARP1 Y889C, MAP2K1 S194P, BRAF K499E/R; Supplementary Table 6).”

- Could the authors present the known and unknown variants identified for each gene in a supplementary table so that is easy to see and review.

Thank you for your comment. We have included this information in revised Supplemental Table 5 and revised Supplementary Table 6.

- The authors use perturb-seq to present potential hypotheses of potential mechanisms of action for these resistant variants and explain the difference in behavior between drug addition and canonical drug resistance variants. They attributed this to different levels of oncogenic signaling. Can the authors verify this using protein analysis? Can they also perform further mechanistic studies to verify their hypotheses based on the perturb-seq data.

We have now performed additional validation experiments to further investigate the behaviour of drug addiction variants.

“HT-29 cells with drug addiction variants had elevated basal MAPK signalling as measured by p-ERK1/2, and elevated p21 protein expression (Fig.2e and Supplementary Fig. 3a). Addition of dabrafenib and cetuximab reduced MAPK activity and p21 expression to near wild-type levels. HT-29 cells harbouring drug addiction variants showed an altered cell morphology and increased β -galactosidase staining in the absence of drug, indicating increased senescence, which was reversed with drug treatment (Fig. 2f and Supplementary Fig. 3b).”

This is consistent with our findings from perturb-seq screens, which revealed increases in EGFR and MAPK signalling pathways through differential gene expression and pathway analyses (revised Fig. 7). Collectively, these data are consistent with oncogene-induced senescence, and are in line with cited reports of this phenotype for individual cancer variants observed in patients (e.g. Das Thakur, *Nature*, 2013; Sun et al, *Nature*, 2014; Mariceau, *Cell*, 2015).

Revised Fig. 2e and 2f.

2e) Western blotting of wild-type HT-29 ABE cells and cells harbouring drug resistance mutations activating the MAPK signalling pathway. Cells were treated with the combination of dabrafenib (80 nM) and cetuximab (1 μ g/ml) or DMSO as a control for 24 h before analysis.

2f) *β-galactosidase staining for senescent cells. β-galactosidase positive senescent foci (blue) are indicated with arrows. Cells were treated with the combination of dabrafenib (80 nM) and cetuximab (1 μg/ml) or DMSO as a control for 48 h before analysis. Representative images are shown for the drug addiction variant MAP2K1 Y130C/H. Predicted amino acid editing consequences are labelled for drug resistance screens and genotyped edits are shown in d, e and f. Data are the average of two independent experiments performed on separate days, or representative of two independent experiments (e-f). See also Supplementary Fig. 2 and 3.*

- The authors hypothesize that reduction in JAK-STAT pathway activity and reduced expression of antigen presentation machinery induced by drug resistance variants may lead to immune evasion. Can the authors verify reduced antigen presentation by flow cytometry in cells with these variants? Can the authors also perform in-vitro and in-vivo studies showing reduced immune cell killing of the cancer cells and enhanced cell growth.

Thank you for your comment. We have shown in the revised manuscript that drug addiction variants can reduce the expression of antigen presentation machinery (HLA and B2M), and conversely, MAPK pathway inhibition with trametinib increases expression, in line with findings from our perturb-seq screens and cited literature. As expected, this effect is reduced in tumour organoids harbouring drug-addiction variants. We also performed co-culture experiments of primary tumour organoids in the presence of autologous anti-tumour T cells, revealing that MEK inhibition significantly increases T cell-mediated killing of tumour cells in this context. This effect was modestly reduced for some drug addiction variants. Overall, these new co-culture studies verify the relevance of modulating MAPK signalling in affecting antigen presentation and anti-tumour immunity in translational, primary cell models, adding valuable supporting evidence for our findings from perturb-seq screens.

“In line with perturb-seq data, HT-29 cells with engineered drug resistance variants had significantly reduced B2M and HLA protein expression, with the exception of the MAP2K2 Y134H drug addiction variant (1/5 variants; Supplementary Fig. 9a). We also engineered three drug addiction variants with base editing into a primary colorectal cancer organoid^{8,83,84}, CRC-9, harbouring FBXW7 and TP53 driver mutations. Although drug addiction variants alone did not reduce B2M and HLA expression in this model, we observed a consistent increase in B2M and HLA protein expression following MAPK pathway inhibition with trametinib, in line with previous reports⁸⁵⁻⁸⁹ (Supplementary Fig. 9b). Induction of B2M and HLA expression with MEK inhibition was significantly attenuated in tumour organoids harbouring drug addiction variants conferring drug resistance (Supplementary Fig. 9b and 9c). Competition assays with GFP⁺ WT control organoids showed a proliferation advantage in MAP2K1 S194P and MAP2K2 Y134H harbouring CRC-9 cells, which was enhanced in the presence of MEK inhibition (Supplementary Fig. 9d). We next tested whether modulation of MAPK signalling would directly affect tumour cell killing by patient-derived autologous, anti-tumour T cells. In co-culture experiments, we observed robust and comparable levels of T-cell mediated killing in all tumour organoid lines (Supplementary Fig. 9e). However, pre-treatment with MEK inhibitor significantly enhanced T cell-mediated cancer cell killing relative to controls without T cells, except in tumour organoids harbouring the MAP2K1 S194P drug addiction variant (Supplementary Fig. 9e).”

Revised Supplementary Figure 9.

Supplementary Figure 9

Supplementary Figure 9. Effect of MAPK signalling and drug addiction variants on antigen presentation and sensitivity to T cell killing

- a) Flow cytometry assessment of B2M and HLA-A,B,C expression in HT-29 ABE cells harbouring drug addiction variants. Data represent the mean \pm SD of biological triplicates. IFN-gamma treatment serves as a positive control (48 h, 400 U/ml). ****P-value <0.0001; ***P-value <0.001; **P-value <0.01; *P-value <0.05; unpaired, two-tailed Student's t-test comparing to non-targeting gRNA (NT) condition. Genotyped variants are shown.
- b) Flow cytometry assessment of B2M and HLA-A,B,C expression in CRC-9 ABE tumour organoid cells harbouring drug addiction variants. Cells were treated with DMSO (control) or the MEK inhibitor trametinib (25 nM) for 48 h before analysis. Data represent the mean \pm SD of two independent experiments, each with two-three replicates. IFN-g treatment serves as a positive control (48 h, 400 U/ml). ****P-value <0.0001; ***P-value <0.001; **P-value <0.01; *P-value <0.05; unpaired, two-tailed Student's t-test comparing non-targeting gRNA (NT) condition. Genotyped variants are shown.
- c) Representative flow cytometry gating used for CRC-9 tumour organoids to assess HLA-A,B,C and B2M cell surface protein expression. Single, live cells with mApple (ABE) and BFP (gRNA) expression were gated for analysis.
- d) Co-competition flow cytometry assays of WT (GFP – NT gRNA expressing cells) and drug resistant CRC-9 tumour organoids (BFP – gRNA expressing) at 72 h. Data represent the mean \pm SD of biological triplicates. ****P-value <0.0001; ***P-value <0.001; **P-value <0.01; *P-value <0.05; unpaired, two-tailed Student's t-test comparing to non-targeting gRNA (NT) condition. Genotyped variants are shown.
- e) Co-culture assay of primary, autologous, anti-tumour T cells with CRC-9 tumour organoids harbouring different drug addiction variants. Cancer cells were pre-treated with the MEK inhibitor trametinib (25 nM) for 48 h before washing and plating the co-culture assay plate. Flow cytometry assessment of absolute cell numbers (measured by counting beads) following 72 h co-culture. Data are expressed as the percentage of live cells remaining as compared to the relevant condition in the absence of T cells and represent the mean \pm SD of biological triplicates. *P-value <0.05; unpaired, two-tailed Student's t-test. Genotyped variants are shown.

• The authors present a drug resistant variant map that they propose can guide treatment. Their argument would be much stronger if they could provide in-vivo studies showing that tumors harboring the particular variants respond as would be expected based on this map and that the strategies they propose (for example drug holidays) enhance survival.

Thank you for this suggestion. Due to the scale of our approach, we took a reductionist in vitro approach to systematically analyse variant function. Although we expect the drug resistance variants discovered here to have predominantly cell-autonomous effects on drug response, we agree that further in vivo validation of variants could be useful to further understand non-cell autonomous effects in the context of a physiological tumour microenvironment. We have added the following remark in the revised Discussion to reflect this point.

“We note that our in vitro screening approach is limited to measuring the cell-autonomous function of variants. Although we gained further mechanistic insights into a subset of these

variants using perturb-seq, in vivo assessment of how these variants affect drug sensitivity in the context of an immune system and physiological tumour microenvironment would be required for a more comprehensive understanding of their mechanism of action in patients.”

Overall, we feel that undertaking in vivo screening of the large number of drug resistant variants identified here would not be feasible. Furthermore, we cite published reports of drug holidays being beneficial in mice harbouring tumours with drug addiction variants (e.g. Das Thakur, *Nature*, 2013). In cancer patients, pulsatile dosing has also been successful for treating cancers with drug addiction variants (Sartore-Bianchi, A. *et al*, *Nat Med*, 2022). Taken together, we feel these studies further validate the physiological relevance of our variant function database.

Other:

1. Figure 3a-b, is there a way to quantify the trend instead of highlighting selected points? I would suggest adding structure annotation/needle-plot at the x-axis if applicable in 3a. Thank you for your suggestion. We have updated the figure accordingly, demonstrating hot-spot mutations found in patients, and loss-of-function mutations clustering around important functional protein regions, such as the PIK3CA kinase domain.

Revised Fig. 3a.

3a) Base editing screens reveal clinically apparent hotspot mutations in oncogenes. Comparison of total COSMIC mutation counts for amino acid positions were compared to the z-score of gRNAs tiling across AKT1 and PIK3CA in HT-29 cells.

We also include a figure showing the correlation between COSMIC mutation counts and the z-score for each gRNA in the screen, relating to Fig. 3a. We recovered the main hot-spot mutations in AKT1 (E17) and PIK3CA (E542 and E545) in our screens with at least one gRNA, but PIK3CA H1047 was not detected as a driver. We mention in the Discussion that we cannot assess all variants here due to the editing capabilities of CBE and ABE NGN base editors, and cannot guarantee the editing of all gRNAs used. We have clarified that PIK3CA H1047 is not detected in the revised Results section relating to Fig. 3a.

“These sites resided in known mutation hot-spots in cancer; AKT1 E17K and PIK3CA E545K/E542K⁴⁵. We also detected rarer driver variants at residues AKT1 D323 and PIK3CA

C378 and E365⁴⁵, although known driver variants of residue H1047 of PIK3CA were not detected due to the editing and saturation constraints of the CBE and ABE NGN base editors.”

Reviewer only Fig 1.

Relationship between gRNA z-scores in HT-29 base editing screens and COSMIC mutation counts. Amino acid positions are labelled for amino acid positions with a total mutation count across all samples >200, or for a z-score of >2.

2. Fig. 6f Why is the author trying to use AUROC as the x-axis instead of log fold change? ROC analysis will be biased if there are many more negative points than positive ones, this may be the case for PI3K drive variant cells, then a PRAUC curve is recommended.

We thank the reviewer for their helpful suggestion. We have updated all of the graphs using AUC to log-2 fold-change, as recommended. We agree this is clearer, and has not changed the interpretation of our findings.

Revised Fig. 7f

Volcano plot of differentially expressed genes between NT gRNA control cells and cells with the PI3K p110a driver variant. FDR, false discovery rate. Significantly downregulated transcripts are in red (including B2M and HLA-A), and upregulated transcripts are in blue.

Revised Supplementary Fig. 8e

Volcano plots of significantly differentially expressed genes (vs NT control gRNA cells) from representative drug resistance gRNAs. B2M is downregulated by both variants. Significant down- and upregulation at FDR<0.1 (Benjamini-Bogomolov correction) are indicated in blue and red respectively.

3. Refer to Fig. 6g, please add more details on calculating the PFS score, "the metric based on Spearman rank correlation of differentially expressed genes in drug-resistant cells from perturb-seq data" (vs non-perturb?), and patients with PFS <6 months vs > 6 months? And can I interpret as canonical/driver variant will lead to long/favorable PFS? Spearman correlation here might not be sensitive to reflect the global gene alteration similarity, SubMap like algorithm would be better here. For calculating PFS score using Tian study [ref23], the author use all (Pre + On treatment)? The pre only or on only can be processed separately.

Thank you for your suggestion. We have now described the computation of the PFS score in more detail in the Methods section, which we include below. In brief, the PFS score is based on the differential expressed results for on- versus pre-treatment provided by Tian et al. as part of their publication, rather than on separate gene expression for pre- or on-treatment. This is because our aim was to compare the change of expression as a result of treatment for patients with longer PFS with the differential expression from our single-cell data, which for each gRNA shows the response of on treatment compared to no treatment. Therefore, the log₂-fold change between on- versus pre-treatment from the Tian study is the corresponding quantity to which to compare our log₂-fold-change data from the single-cell study. A higher (less negative) PFS score indicates favourable PFS compared to other variants. Canonical and driver variants tend to have a lower PFS score compared to drug controls, which indicates a more negative outlook in terms of PFS. Drug addiction variants have the lowest PFS. In order to increase sensitivity of the Spearman correlation to the relevant genes, we computed the correlations based on the subset of genes with adjusted p-values < 0.1 and abs(log₂-fold change) > 0.25 for differential expression at 15 days versus pre-treatment for PFS<6m or PFS>6m. We were not able to run the SubMap algorithm from the GenePattern platform. We think that this is due to the nature of the comparison, with only one column of data for PFS<6m and one for PFS>6m, as we did not have patient-specific data from Tian et al.

"We used published differential gene expression results for BRAF-V600E colorectal cancer patients treated with a combination of PD-1, BRAF and MEK inhibition²⁷. Tian et al. reported log₂-fold gene expression changes at 15 days versus pre-treatment bulked across patients with progression free survival (PFS) > 6 months, and bulked across those with PFS < 6 months. To perform correlation analysis to our data, we identified the genes with an absolute log₂-fold expression change of more than 0.25 and adjusted p-value of less than

0.1 for either the data for PFS > 6 months or PFS < 6 months (2,148 genes). Then, based on this subset of genes, we used Spearman rank correlation to correlate the log2-fold expression changes for PFS > 6 months reported in Tian et al with our log2-fold changes from differential expression testing results for individual targeting gRNAs compared to non-targeting controls (PFS outcome scores)."

Reviewer #2:

Remarks to the Author:

In this study, Coelho et al. employed CRISPR base editing mutagenesis screens in four cancer cell lines to prospectively identify genetic resistance mechanisms to 10 oncology drugs. Through these screens the authors identified four classes of protein variants influencing drug sensitivity, offering insights into potential strategies for overcoming drug resistance. The authors coupled these approaches with single-cell transcriptomics to reveal the effects of protein variants on gene expression.

Overall, the findings of the study are well presented and provide a valuable map of genetic drug resistance mechanisms with implications for patient stratification and optimizing cancer treatment approaches. We recommend the authors to address the points below to strengthen their observations.

We thank the reviewer for their thoughtful and helpful comments on our manuscript.

1) It would be important to provide additional experimental evidence on the mechanisms of action of the different classes of variants identified. For example, how do drug addiction mutants in MAPK signaling factors cause detrimental effects in the absence of treatment? We have now performed additional validation experiments to further investigate the behaviour of drug addiction variants.

“HT-29 cells with drug addiction variants had elevated basal MAPK signalling as measured by p-ERK1/2, and elevated p21 protein expression (Fig.2e and Supplementary Fig. 3a). Addition of dabrafenib and cetuximab reduced MAPK activity and p21 expression to near wild-type levels. HT-29 cells harbouring drug addiction variants showed an altered cell morphology and increased β -galactosidase staining in the absence of drug, indicating increased senescence, which was reversed with drug treatment (Fig. 2f and Supplementary Fig. 3b).”

This is consistent with our findings from perturb-seq screens, which revealed increases in EGFR and MAPK signalling pathways through differential gene expression and pathway analyses (revised Fig. 7). Collectively, these data are consistent with oncogene-induced senescence, and are in line with cited reports of this phenotype for individual cancer variants observed in patients (e.g. Das Thakur, *Nature*, 2013; Sun et al, *Nature*, 2014; Mariceau, *Cell*, 2015).

Revised Fig. 2e and 2f

2e) Western blotting of wild-type HT-29 ABE cells and cells harbouring drug resistance mutations activating the MAPK signalling pathway. Cells were treated with the combination of dabrafenib (80 nM) and cetuximab (1 µg/ml) or DMSO as a control for 24 h before analysis.

2f) β-galactosidase staining for senescent cells. β-galactosidase positive senescent foci (blue) are indicated with arrows. Cells were treated with the combination of dabrafenib (80 nM) and cetuximab (1 µg/ml) or DMSO as a control for 48 h before analysis. Representative images are shown for the drug addiction variant MAP2K1 Y130C/H.

Predicted amino acid editing consequences are labelled for drug resistance screens and genotyped edits are shown in d, e and f. Data are the average of two independent experiments performed on separate days, or representative of two independent experiments (e-f). See also Supplementary Fig. 2 and 3.

What are the mechanisms by which PARP1 variants sensitize to PARP inhibitors?

Thank you for your comment. We have now performed multiple additional analyses of the drug-sensitising mutations in PARP1, including PARP-trapping assays, proliferation assays and analysis of the PARP1 structure.

“These drug-sensitising variants were exclusively within the catalytic domain (10/10), and predominantly within the helical subdomain (6/10; Fig. 4a), which has an autoinhibitory role through blocking NAD⁺ binding⁶¹. For example, the helical subdomain missense variant PARP1 I691T, sensitised to both PARP inhibitors in base editing screens (Fig. 4a), in validation experiments using cell competition assays (Fig. 4b), and in proliferation assays (Supplementary Fig. 4c). In contrast, PARP1 Y889C within the ART (ADP-ribosyl transferase fold) subdomain, conferred resistance to niraparib, but sensitised to olaparib (Fig. 4a-c). Consistently, PARP1 Y889C increased cytotoxic PARP trapping on DNA following DNA damage in the presence of low doses of olaparib, and reduced PARP trapping in the presence of niraparib relative to non-targeting gRNA control cells in immunofluorescence and chromatin fractionation assays (Supplementary Fig. 4d and 4e). PARP1 Y889 is within the drug-binding pocket within the catalytic domain (Fig. 4d), and has been shown to be a critical residue for determining inhibitor specificity over related proteins such as tankyase 1 through van der Waals and pi-interactions with PARP inhibitors⁶². This provides a potential explanation for the apparent disparate effects of the Y889C variant on the sensitivity to different PARP inhibitors. Overall, these data imply that olaparib could be effective in treating niraparib-resistant tumour cells harbouring this variant.”

Revised Fig. 4a.

4a) Base editing screens of PARP1 and PARP2 in the presence of olaparib or niraparib reveal drug-sensitising and drug resistance variants. Comparison of gRNA z-scores for the control treated arm, olaparib-treated or niraparib-treated arm vs plasmid library is plotted against the amino acid position. Predicted amino acid editing consequences are labelled for drug resistance variants. The position of the catalytic domain of PARP1 and PARP2 is shown. Screening data are the average of two independent screens performed on separate days. HD; helical subdomain. ART; ADP-ribosyl transferase fold subdomain.

Revised Fig. 4c and 4d.

4c) Dose response proliferation assay comparing the growth of MHH-ES-1 ABE cells harbouring a NT control gRNA or a gRNA installing the PARP1 Y889C variant. Data are the average of two-independent experiments performed, each performed in triplicate. CTG; CellTiter-Glo. 2-way ANOVA; ***p-value < 0.0001.

4d) Crystal structures of PARP1 bound to olaparib (PDB 7AAD)⁶¹ or niraparib (PDB 7KK5)⁶² comparing the two binding modes of the inhibitors with respect to the Y889 residue. NT gRNA; non-targeting gRNA. See also Supplementary Fig. 4.

Revised Supplementary Fig. 4c.

4c) Proliferation assays measuring drug response to olaparib and niraparib PARP inhibitors in MHH-ES-1 ABE cells harbouring the genotyped drug-sensitising variant, PARP1 I691T.

Data represent the mean \pm SD of two independent experiments performed on separate days, each in biological triplicate (CTG; CellTiter-Glo). 2-way ANOVA; ***p-value < 0.0001.

Revised Supplementary Fig. 4d.

4d) Western blotting assessment of PARP trapping on DNA in MHH-ES-1 ABE cells harbouring the PARP1 Y889C variant or a non-targeting (NT) control gRNA. Cells were treated with a DNA damaging agent (MMS, 0.01 %) and the PARP inhibitor with olaparib or niraparib (both at 3 μ M) for 4 h before analysis. Nuclei were fractionated into a chromatin-bound and soluble fractions prior to immunoblotting. Cl. denotes cleaved PARP in response to DNA damage and PARP inhibition. Lamin A/C and histone H3 serve as loading controls for chromatin-bound and soluble fractions, respectively.

Revised Supplementary Fig. 4e.

4e) Immunofluorescence microscopy assessment of PARP trapping on DNA in MHH-ES-1 ABE cells harbouring the PARP1 Y889C variant or a non-targeting (NT) control gRNA. Cells were treated with a DNA damaging agent (MMS, 0.01 %) and the PARP inhibitor with olaparib or niraparib (dose titration) for 4 h. PARP protein not bound to chromatin was removed before fixation and staining. Data represent the mean \pm SD fluorescence nuclear intensity of PARP1 from biological triplicates.

2) Figure 6. The authors show that sc-SNV-seq can be employed to examine the transcriptional changes induced by variants that cause drug resistance. Can the authors use the same approach to investigate the effects induced by variants that cause drug sensitivity?

Would this approach be informative for understanding the mechanism of action of these variants?

Thank you for your suggestion. We discovered drug sensitising variants in EGFR, which we analyse in depth (revised Fig. 6) and find to be associated with more profound inhibition of EGFR signalling and increased levels of mutant EGFR protein on the cell surface. For drug sensitising variants to dabrafenib and cetuximab, these variants were associated with reduced EGFR signalling. Finally, for drug-sensitising variants in PARP1, we have provided additional mechanistic understanding of these variants in our revisions and suggest this phenotype is related to modulation of inhibitor binding and increased PARP trapping. Given these considerations, we suggest that additional perturb-seq screens focused on this limited number of drug-sensitising variants and their associated transcriptional signatures would not add significant further mechanistic insights.

3) Figure 6d. The authors identify drug resistance variants with immune evasion signatures. Do these variants reduce the ability of immune cells to kill cancer cells in co-culture systems?

Thank you for your comment. We have shown in the revised manuscript that drug addiction variants can reduce the expression of antigen presentation machinery (HLA and B2M), and conversely, MAPK pathway inhibition with trametinib increases expression, in line with findings from our perturb-seq screens and cited literature. As expected, this effect is reduced in tumour organoids harbouring drug-addiction variants. We also performed co-culture experiments of primary tumour organoids in the presence of autologous anti-tumour T cells, revealing that MEK inhibition significantly increases T cell-mediated killing of tumour cells in this context. This effect was modestly reduced for some drug addiction variants. Overall, these new co-culture studies verify the relevance of modulating MAPK signalling in affecting antigen presentation and anti-tumour immunity in translational, primary cell models, adding valuable supporting evidence for our findings from perturb-seq screens.

“In line with perturb-seq data, HT-29 cells with engineered drug resistance variants had significantly reduced B2M and HLA protein expression, with the exception of the MAP2K2 Y134H drug addiction variant (1/5 variants; Supplementary Fig. 9a). We also engineered three drug addiction variants with base editing into a primary colorectal cancer organoid^{8,83,84}, CRC-9, harbouring FBXW7 and TP53 driver mutations. Although drug addiction variants alone did not reduce B2M and HLA expression in this model, we observed a consistent increase in B2M and HLA protein expression following MAPK pathway inhibition with trametinib, in line with previous reports⁸⁵⁻⁸⁹ (Supplementary Fig. 9b). Induction of B2M and HLA expression with MEK inhibition was significantly attenuated in tumour organoids harbouring drug addiction variants conferring drug resistance (Supplementary Fig. 9b and 9c). Competition assays with GFP⁺ WT control organoids showed a proliferation advantage in MAP2K1 S194P and MAP2K2 Y134H harbouring CRC-9 cells, which was enhanced in the presence of MEK inhibition (Supplementary Fig. 9d). We next tested whether modulation of MAPK signalling would directly affect tumour cell killing by patient-derived autologous, anti-tumour T cells. In co-culture experiments, we observed robust and comparable levels of T-cell mediated killing in all tumour organoid lines (Supplementary Fig. 9e). However, pre-treatment with MEK inhibitor significantly enhanced T cell-mediated cancer cell killing

relative to controls without T cells, except in tumour organoids harbouring the MAP2K1 S194P drug addiction variant (Supplementary Fig. 9e).”

Revised Supplementary Figure 9.

Supplementary Figure 9

Supplementary Figure 9. Effect of MAPK signalling and drug addiction variants on antigen presentation and sensitivity to T cell killing

- f) Flow cytometry assessment of B2M and HLA-A,B,C expression in HT-29 ABE cells harbouring drug addiction variants. Data represent the mean \pm SD of biological triplicates. IFN-gamma treatment serves as a positive control (48 h, 400 U/ml). ****P-value <0.0001; ***P-value <0.001; **P-value <0.01; *P-value <0.05; unpaired, two-tailed Student's t-test comparing to non-targeting gRNA (NT) condition. Genotyped variants are shown.*
- g) Flow cytometry assessment of B2M and HLA-A,B,C expression in CRC-9 ABE tumour organoid cells harbouring drug addiction variants. Cells were treated with DMSO (control) or the MEK inhibitor trametinib (25 nM) for 48 h before analysis. Data represent the mean \pm SD of two independent experiments, each with two-three replicates. IFN-g treatment serves as a positive control (48 h, 400 U/ml). ****P-value <0.0001; ***P-value <0.001; **P-value <0.01; *P-value <0.05; unpaired, two-tailed Student's t-test comparing non-targeting gRNA (NT) condition. Genotyped variants are shown.*
- h) Representative flow cytometry gating used for CRC-9 tumour organoids to assess HLA-A,B,C and B2M cell surface protein expression. Single, live cells with mApple (ABE) and BFP (gRNA) expression were gated for analysis.*
- i) Co-competition flow cytometry assays of WT (GFP – NT gRNA expressing cells) and drug resistant CRC-9 tumour organoids (BFP – gRNA expressing) at 72 h. Data represent the mean \pm SD of biological triplicates. ****P-value <0.0001; ***P-value <0.001; **P-value <0.01; *P-value <0.05; unpaired, two-tailed Student's t-test comparing to non-targeting gRNA (NT) condition. Genotyped variants are shown.*
- j) Co-culture assay of primary, autologous, anti-tumour T cells with CRC-9 tumour organoids harbouring different drug addiction variants. Cancer cells were pre-treated with the MEK inhibitor trametinib (25 nM) for 48 h before washing and plating the co-culture assay plate. Flow cytometry assessment of absolute cell numbers (measured by counting beads) following 72 h co-culture. Data are expressed as the percentage of live cells remaining as compared to the relevant condition in the absence of T cells and represent the mean \pm SD of biological triplicates. *P-value <0.05; unpaired, two-tailed Student's t-test. Genotyped variants are shown.*

4) Figure 7b. The genotype of several of the variants introduced is complex. To define which amino acid changes are responsible for the phenotype observed, the authors should use other genome editing approaches, such as prime editing or homology-directed repair, for introducing precise single amino acid changes and studying their effects.

We thank the reviewer for their comment. We have now performed a targeted prime editing screen in PC9 cells to introduce specific variants, and to introduce variants that could not be installed with base editing (such as EGFR C797S for osimertinib resistance). This has also highlighted several variants that are deleterious for EGFR-addicted cells (such as EGFR T790K, below). We have also performed validation experiments where we discover new drug resistance variants in the regulatory region of EGFR.

“Firstly, we confirmed the installation of EGFR C797S in PC9 MLH1 KO doxycycline-inducible prime editor cells in the presence of osimertinib (Supplementary Fig. 5a). We then

created a focused prime editing gRNA (pegRNA) library of 162 pegRNAs⁷¹ (three pegRNAs per variant) designed to install all amino acid substitutions achievable with SNVs at six residues of interest identified from base editing screens of EGFR and including C797 (Supplementary Table 1). pegRNA sequencing confirmed a strong correlation between independent biological replicates, both with and without selection with osimertinib (Supplementary Fig. 5c; Supplementary Table 2). 6/6 pegRNAs installing C797S were significantly enriched in the presence of osimertinib (Fig. 5c). Interestingly, EGFR C797 and T790 substitutions to chemically distinct residues were significantly depleted (Fig. 5c and Supplementary Fig. 5c), including substitutions to lysine and arginine (charged), or proline and phenylalanine (steric effects). This suggests that particular variants of these key drug resistance residues can disrupt EGFR function, potentially explaining why they are not observed as drug resistance alleles. Although we achieved the sensitivity required to detect loss-of-function effects, pegRNAs designed to install stop codons in EGFR were not depleted (0/3; Fig. 5c), indicating low prime editing efficiencies for some pegRNAs, and potentially relating to the high copy number of EGFR in PC9 cells²⁰. Overall, this highlights the relative merits of prime editing and base editing technologies¹⁵.

We set out to validate drug resistance variants in the C-terminal regulatory region of EGFR in arrayed validation experiments using a stabilised, engineered prime editing gRNA (epegRNA) design⁷² in competition assays with WT cells. EGFR D1012N, A1013D and A1013G prime edited variants had a modest growth advantage over WT cells and C797C synonymous variant harbouring cells, but this was significantly enhanced in the presence of osimertinib, suggesting these variants confer resistance, albeit less effectively than EGFR C797S (Fig. 5d). This highlights the utility of base editing screens to capture sites functionally involved in drug sensitivity across entire proteins, distal to drug-binding and active sites, and for directing subsequent mutational scans at higher resolution using prime editing.”

Revised Fig. 5c and d.

5c) Prime editing mutagenesis screens of EGFR in the presence and absence of osimertinib. PC9 MLH1 KO cells were prime edited for 7 days with doxycycline (1 μ g/ml) before growth for 10 days in DMSO (control), or osimertinib (75 nM). Data represent the z-score for each pegRNA derived from the average of two independent screens performed on separate days. Samples were compared to the plasmid library.

5d) Competition flow cytometry assays in PC9 MLH1 KO cells comparing the growth of NT gRNA GFP cells to epegRNA BFP cells harbouring different EGFR variants in the presence and absence of osimertinib (75 nM) for 5 d. Data are normalised to day 0 ratios and represent

the mean \pm SD of biological triplicates. Unpaired, two-tailed Student's t-test comparing to the EGFR C797C synonymous variant control; *p-value <0.0005.

Revised Supplementary Fig. 5c

5c) Replicate correlation between pegRNA z-scores from EGFR prime editing mutagenesis screens performed in PC9 MLH1 KO PE2 cells. Data are from two independent screens performed on different days. Labeled are predicted mutations in EGFR installed by the pegRNAs. Pearson correlation coefficient values (r) between independent replicate screens are shown. pegRNA, prime editing gRNA.

5) It would be helpful to highlight all variants identified through the screens that have also been associated to altered drug response in cancer patient datasets.

Thank you for this important suggestion. To our knowledge, there are limited databases of drug resistance mutations in cancer and this was, in part, a motivation for this study. One of the most comprehensive efforts to compile curated clinical drug resistance mutations is from COSMIC. Therefore, we have included COSMIC drug resistance variants and variants from selected cited articles relating to drugs that we screen in this study (e.g. Awad et al, *NEJM*, 2021, and Brammeld et al, *Genome Res*, 2017) in revised Supplementary Table 5. We highlight variants that were recovered in our base editing screens and those that were missed due to the editing capabilities of cytidine and adenine base editors. Drug resistance variants that we detect in this study and do not have previous drug resistance annotations can be found in revised Supplementary Table 6. For our validated, genotyped variants, this information is also labelled in revised Figure 8. Furthermore, we include the following text in the results section of the manuscript to provide more context for our findings:

“To benchmark our resistance variant map, we surveyed COSMIC-curated drug response data⁴⁶ and literature relating to clinical incidence of resistance to the drugs analysed in this study^{42,50}. This identified 88 amino acid positions in the screened target proteins associated with drug response (Supplementary Table 5, Methods). 85 of these had at least one gRNA predicted to target the amino acid position, 30 of which (35.29 %) had a concordant drug resistance phenotype in our screening dataset specific to the reported drug and gene target. In addition, we observed 252 edits at amino acid positions that had not been previously associated with altered drug response (e.g. PARP1 Y889C, MAP2K1 S194P, BRAF K499E/R; Supplementary Table 6).”

Reviewer #3:

Remarks to the Author:

Coelho et al. used base-editing screens with or without drugs to identify drug resistance and sensitization variants across 11 cancer genes. In total, they identified 45 genotyped variants that confer resistance and sensitivity. The authors validate a subset of variants in different types of follow-up experiments but also perform a base-editing perturb-seq screen with a focused library to identify signatures and measure the transcriptional impact of each variant. As the authors mentioned, this is one of the largest base-editing screens dataset and identifies many valuable drug-resistant and sensitizing variants. The authors provide important ideas about how this information can be used to identify personalized therapeutic strategies.

We thank the reviewer for their thoughtful and helpful comments on our manuscript.

Main point

The manuscript's main concern is that the screens are very noisy, and the authors don't provide sufficient information on the false discovery rate of the hits. The quality of the screens differs some screens have better signal, and there are clearly some outlier hits. The metrics they used to call hits are similar across the screens, independently of their noise. The authors must provide additional quality control metrics typically used on these types of screens beyond the experiment correlation. They must also include more information about how the controls score compared to experimental sgRNAs and statistical or false discovery rate estimates.

Thank you for this important suggestion. We have now included MAGeCK-generated statistics for all screens reported in the revised manuscript, including p-values and false discovery rates (FDRs). These can be found in the revised Supplementary Table 2, including the data for all control gRNAs, and verify high screening quality, our hit calling and signal to noise ratio - we discuss this in more detail in points below. Furthermore, in our revised figures, we display the relative distributions of control gRNAs (non-targeting) vs essential-targeting gRNAs – this helps the reader to assess screen quality and the distribution of control gRNAs. We note that z-scores, used throughout, also help to account for the degree of noise, as the standard deviation of each dataset is considered in the derivation of z-scores.

Revised Fig. 3b

3b) Drug resistance variants to the PI3K inhibitor, pictilisib, profiled with CBE and ABE base editors in HT-29 CRC cells. Comparison of gRNA z-scores for the control treated arm vs plasmid library, and the drug-treated arm vs plasmid library is shown.

We note that the FDR and p-values have been useful to detect rare gRNAs/pegRNAs where the correlation between replicates is poor, especially in prime editing screens (included in our revision), where editing rates are generally lower. As an example, in the analysis below, the synonymous variant V1011V has ostensibly modestly increased proliferation in the presence of osimertinib, but has a high FDR score due to a low replicate correlation (not significant; p-value >0.05 and FDR >0.1). We include MAGeCK-derived p-values and FDRs for base and prime editing screens in revised Supplementary Table 2.

Reviewer only Fig. 2

Prime editing mutagenesis screens of EGFR in the presence and absence of osimertinib. PC9 MLH1 KO cells were prime edited for 7 days with doxycycline (1 µg/ml) before growth for 10 days in DMSO (control), or osimertinib (75 nM). Data represent the z-score for each pegRNA derived from the average of two independent screens performed on separate days. Samples were compared to the plasmid library. P-values and FDR (false discovery rate) of each pegRNA are indicated by size and colour, respectively.

The authors focus in the sections on highlighting specific examples, which is valuable but also need to provide clear quantitative information on how many hits are scoring in the experimental vs. control sgRNAs, which would provide information on the false discovery rate. It would be helpful if they could start the section by providing this information before moving to describe specific hits.

Thank you for this suggestion. We now start each section with a brief summary of the hits proportion of hits that are control gRNAs. All hit gRNAs are based on robust effects sizes (z-score <-2 or z-score >2; Methods) and have MAGeCK FDR <0.1 and p-value < 0.05. From the descriptions below, a minority of hit gRNAs are controls (overall, 10/1,098, or, 0.91 %), verifying screen quality. We have summarised the numbers of hit gRNAs that were controls for the reviewer below, and have included these remarks in the revised text at the start of the appropriate sections, as suggested.

Trametinib 0/175; dabrafenib plus cetuximab 0/81; pictilisib 1/40 (2.5 %); sotorasib 3/198 (1.51 %); adagrasib 1/224 (0.45 %); olaparib 2/65 (3.08 %); niraparib 0/131; gefitinib 3/103 (2.91 %); osimertinib 0/81.

The authors use perturb-seq to assess the level of transcriptional signal as a way to validate the resistant variants, which seems to be a nice way to validate at scale. They should provide more information on how many of the expected variants tested elicited a transcriptional response, even if they provide a ranking in the supplemental table and figures of the patterns.

We thank the reviewer for this important suggestion. We identified gRNAs that elicited a significant response by testing for differential expression for all genes, and with PROGENY and MAYA pathway scores for those barcodes with at least 10 iBAR barcodes assigned to them. Out of these gRNAs, we selected those with at least one gene or pathway score differentially expressed with a p-value $< 10^{-6}$. The selection of gRNAs using this cut-off allowed us to use the Benjamini-Bogomolov approach (an extension of Benjamini-Hochberg), for selection of groups with multiple testing correction and principled control of the false discovery rate for the selected gRNAs. We have now added this information to the revised Supplementary Table 3, and the following text in the revised Results section.

“(Supplementary Table 3). Of the 35 gRNAs tested that conferred resistance to dabrafenib and cetuximab in base editing proliferation screens, 22 (62.86 %) elicited a significant transcriptional response vs non-targeting gRNA harbouring cells (Methods).”

The Revised Supplementary Fig. 8c further summarises the performance of gRNAs in the perturb-seq screen by variant class and base editing gRNA z-score.

Revised Supplementary Fig. 8c.

Comparison of z-scores from proliferation read-out base editing screens to energy distance scores derived from perturb-seq screens. Variant classes based on the HT-29 proliferation screens in dabrafenib and cetuximab are indicated. Intermediate variants discussed in the text are labelled.

In Figure 7A, the authors provide a nice summary illustration of their top hits, which are also genotyped. Since this is their final hit list, can they more precisely describe how they selected the top-scoring?

Thank you for your comment. We have added the following comment in the Methods section.

“From our drug resistance hits from base editing screens (z-score of >2 and >1 in each replicate with FDR <0.1 and p-value <0.05), we filtered for non-synonymous coding mutations in target genes that were non-redundant and had not already been genotyped by Sanger sequencing in validation studies (e.g. EGFR splice variants and PARP1 drug-sensitising variants). Given the large number of variants, we preferentially selected proximal variants such that we could cover multiple variants in a single amplicon. In total, we analysed 46 gRNAs targeting 7 genes over 21 amplicons in two separate experiments.”

Minor points

The classification of the functional variants is important. However, structuring the section by functional variants makes it hard to follow. The authors might want to consider separating the sections by drug classes (e.g. Osimertinib, Gefinitif) or make the sections more consistent to help the reader follow the results.

Thank you for this suggestion. In the revision, we have separated the EGFR inhibitor resistance section from the discussion of sensitising variants to increase clarity (revised Fig. 5 and Fig. 6).

The authors should consider better visualizing the data for the proliferation base-editing screens. For example, figure 1b and supplemental figure 2a could be shown as distributions and figure 2. In Figure 2a they might consider not plotting all the genes in the same plot. The results of the perturb-seq screens have clear visualizations.

We thank the reviewer for this recommendation. We wanted to capture rare gain-of-function mutations with positive L2FC in these plots (e.g. activating mutations in *BRAF*, *AKT1*, *PIK3CA*). These would not be captured in density or histogram plots. In the revision, we have used box-plots to more accurately convey the distribution of the data and highlight outlier gain-of-function variants. In Fig. 2, we wanted to show all of the genes in one plot to capture as much information as possible given space constraints. In the revision, we have added gene labels for important variants to make them more easily interpretable, although this is not feasible for all hit gRNAs in all graphs due to overlapping labels. Overall, we hope these alterations improve clarity.

Revised Fig. 1b and revised Fig. 3b

1b) Base editor screens in HT-29 cells across 11 cancer genes show depletion of gRNAs targeting essential genes demonstrating base editing activity. Unpaired, two-tailed Student's t-test comparing non-targeting gRNAs to gRNAs targeting essential gene splice sites. Boxplots represent the median, inter-quartile range (IQR) and whiskers are the lowest and highest values within 1.5 x IQR.

3c) Drug resistance variants to the PI3K inhibitor, pictilisib, profiled with CBE and ABE base editors in HT-29 CRC cells. Comparison of gRNA z-scores for the control treated arm vs plasmid library, and the drug-treated arm vs plasmid library is shown.

The authors use MYC throughout the paper as an essential gene control, but they have many more sgRNA against essential genes that they could use in aggregation. Is there a reason to use only MYC?

We included all of the essential genes in Fig. 1b and Supplementary Fig. 2a, and originally omitted them from the other screen figures for clarity. We have now included the aggregated essential-targeting control gRNAs in the revised figures.

Revised Fig. 3b

Drug resistance variants to the PI3K inhibitor, pictilisib, profiled with CBE and ABE base editors in HT-29 CRC cells. Comparison of gRNA z-scores for the control treated arm vs plasmid library, and the drug-treated arm vs plasmid library is shown.

In supplemental fig 5a, the authors use PC9 with gefinitif and osimerib to validate the focused library and HT29 with the dafratinib and cetuximax to do the perturb-seq profiling. Can the authors make this clear in the main text?

Thank you. We have changed the text in the revision accordingly.

“This focused gRNA library performed as expected in proliferation screens in PC9 cells, displaying a significant correlation in gRNA effect size with larger base editing screens and independently validating drug resistance hits for gefitinib and osimertinib (Supplementary Fig. 7a).”

When the authors describe the validation experiments, it would be important to describe how many are identified and how many are validated. For example, in the second paragraph of page 10, they could write that among the X identified, we validated Y. They might consider a summary table that describes which variants were validated.

Thank you for your suggestion. We have added these descriptions in the relevant sections of the revised text. For example:

“We validated the effect of BRAF and EGFR combination and MEK1/2 inhibitor drug resistance variants using arrayed proliferation assays and by analysing cell signalling. Overall, we set out to validate 4/13 drug addiction and 3/36 canonical drug resistance gRNAs for dabrafenib and cetuximab, and 4/10 drug addiction and 2/30 canonical drug resistance gRNAs for trametinib.”

In summary, we experimentally validated 20 gRNAs and four epegRNAs, of which 11 have amplicon sequencing genotyping data and eight have been verified with Sanger sequencing. This information is summarised in the revised Supplementary Table 2.

In the Drug-sensitizing variants section, the authors start by describing the resistance, not the sensitizing hits, which makes it confusing to the reader. They could consider an introductory sentence to include this information in this section.

Thank you for your suggestion. We have now split Fig. 5 into two figures (revised Fig. 5 and Fig. 6) to make these sections on drug resistance and drug-sensitising mutations distinct – we agree that this makes the revised text clearer.

Decision Letter, first revision:

2nd Aug 2024

Dear Dr Garnett,

Thank you for submitting your revised manuscript "Genetic landscape of cancer drug resistance mechanisms from base editing screens" (NG-A64099R1). It has now been seen by the original referees and their comments are below. The reviewers find that the paper has improved in revision, and therefore we'll be happy in principle to publish it in Nature Genetics, pending minor revisions to satisfy our editorial and formatting guidelines.

Sincerely,

Safia Danovi, PhD
Senior Editor, Nature Genetics
ORCID: 0009-0007-7822-5479

Reviewer #1 (Remarks to the Author):

The reviewer appreciate the additional effort/work the authors put into this study. However, the two major criticisms remain: (1) the novelty and impact of the base editing screen is limited, which is now standard; and (2) the characterization of the top hits remain superficial. While the screen provide a catalog of mutants' effect in these cell line is useful, little biological insight was provided, nor is therapeutically relevant data.

Reviewer #2 (Remarks to the Author):

The authors have satisfactorily addressed the concerns of this reviewer.

Reviewer #3 (Remarks to the Author):

The revised version is a significant improvement. The authors have addressed all my concerns.

Final Decision Letter:

13th Sep 2024

Dear Dr Garnett,

I am delighted to say that your manuscript "Base editing screens define the genetic landscape of cancer drug resistance mechanisms" has been accepted for publication in an upcoming issue of *Nature Genetics*.

Over the next few weeks, your paper will be copyedited to ensure that it conforms to *Nature Genetics* style. Once your paper is typeset, you will receive an email with a link to choose the appropriate publishing options for your paper and our Author Services team will be in touch regarding any additional information that may be required.

Your paper will be published online after we receive your corrections and will appear in print in the next available issue. You can find out your date of online publication by contacting the Nature Press Office (press@nature.com) after sending your e-proof corrections.

Before your paper is published online, we shall be distributing a press release to news organizations worldwide, which may very well include details of your work. We are happy for your institution or funding agency to prepare its own press release, but it must mention the embargo date and *Nature Genetics*. Our Press Office may contact you closer to the time of publication, but if you or your Press Office have any enquiries in the meantime, please contact press@nature.com.

Please note that *Nature Genetics* is a Transformative Journal (TJ). Authors may publish their research with us through the traditional subscription access route or make their paper immediately open access through payment of an article-processing charge (APC). Authors will not be required to make a final decision about access to their article until it has been accepted. Find out more about Transformative Journals

Authors may need to take specific actions to achieve compliance with funder and

institutional open access mandates. If your research is supported by a funder that requires immediate open access (e.g. according to [Plan S principles](https://www.nature.com/nature-portfolio/editorial-policies/self-archiving-and-license-to-publish)) then you should select the gold OA route, and we will direct you to the compliant route where possible. For authors selecting the subscription publication route, the journal's standard licensing terms will need to be accepted, including [a href="https://www.nature.com/nature-portfolio/editorial-policies/self-archiving-and-license-to-publish"](https://www.nature.com/nature-portfolio/editorial-policies/self-archiving-and-license-to-publish). Those licensing terms will supersede any other terms that the author or any third party may assert apply to any version of the manuscript.

If you have not already done so, we strongly recommend that you upload the step-by-step protocols used in this manuscript to protocols.io. protocols.io is an open online resource that allows researchers to share their detailed experimental know-how. All uploaded protocols are made freely available and are assigned DOIs for ease of citation. Protocols can be linked to any publications in which they are used and will be linked to from your article. You can also establish a dedicated workspace to collect all your lab Protocols. By uploading your Protocols to protocols.io, you are enabling researchers to more readily reproduce or adapt the methodology you use, as well as increasing the visibility of your protocols and papers. Upload your Protocols at <https://protocols.io>. Further information can be found at <https://www.protocols.io/help/publish-articles>.

Sincerely,

Safia Danovi, PhD
Senior Editor, Nature Genetics
ORCID: 0009-0007-7822-5479